# Selfish conflict underlies RNA-mediated parent-of-origin effects

Pinelopi Pliota[1], Hana Marvanova[1,2], Alevtina Koreshova[1,2], Yotam Kaufman[3], Polina Tikanova[1,2], Daniel Krogull[1,2], Andreas Hagmüller[1], Sonya A. Widen[1], Dominik Handler[1], Joseph Gokcezade[1], Peter Duchek[1], Julius Brennecke[1], Eyal Ben-David[3,4,5] & Alejandro Burga[1,5 ✉]

Genomic imprinting—the non-equivalence of maternal and paternal genomes—is a critical process that has evolved independently in many plant and mammalian species[1,2]. According to kinship theory, imprinting is the inevitable consequence of conflictive selective forces acting on differentially expressed parental alleles[3,4]. Yet, how these epigenetic differences evolve in the first place is poorly understood[3,5,6]. Here we report the identification and molecular dissection of a parent-of-origin effect on gene expression that might help to clarify this fundamental question. Toxin-antidote elements (TAs) are selfish elements that spread in populations by poisoning non-carrier individuals[7–9]. In reciprocal crosses between two *Caenorhabditis tropicalis* wild isolates, we found that the *slow-1/grow-1* TA is specifically inactive when paternally inherited. This parent-of-origin effect stems from transcriptional repression of the *slow-1* toxin by the PIWI-interacting RNA (piRNA) host defence pathway. The repression requires PIWI Argonaute and SET-32 histone methyltransferase activities and is transgenerationally inherited via small RNAs. Remarkably, when *slow-1/grow-1* is maternally inherited, *slow-1* repression is halted by a translation-independent role of its maternal mRNA. That is, *slow-1* transcripts loaded into eggs—but not SLOW-1 protein—are necessary and sufficient to counteract piRNA-mediated repression. Our findings show that parent-of-origin effects can evolve by co-option of the piRNA pathway and hinder the spread of selfish genes that require sex for their propagation.

Diploid organisms carry two copies of each gene: one inherited from their mother and the other one from their father. Typically, these copies are functionally interchangeable. Imprinted genes are the exception to this rule. They keep an epigenetic memory of their gametic provenance, making maternal and paternal genomes non-equivalent, which has a large effect on embryonic development, species hybridization and human disease[10]. Multiple theories have been put forward to explain the evolution of imprinting. The most accepted theory—kinship conflict—states that imprinting arises when there are conflicting interests between maternal and paternal genomes owing to differential investment in their offspring[3,4]. Notably, this theory presupposes the existence of mechanisms that establish differences in the expression of maternal and paternal alleles—otherwise, there would be nothing to select on[3]. This raises the critical question of how parent-of-origin effects on gene expression evolve in the first place.

The discovery of the first imprinted loci in mammals led to the hypothesis that imprinting evolved from host defence mechanisms that use DNA methylation to keep viruses and parasitic genes at bay[11,12]. This is in line with the close proximity of many imprinted loci to transposable elements in plants[13,14] and piRNA-induced DNA methylation of a retrotransposon being critical for the paternal imprinting of mouse

*Rasgrf1* (ref. 15). However, the evolutionary origins of imprinting remain poorly understood at the molecular level. More recently, histone modifications, such as H3K27me3, have been reported to act as imprinting marks independently of DNA methylation in mice[16]. These observations have raised the possibility that a link between parent-of-origin-dependent gene expression and host defence mechanisms can also be found in organisms that lack DNA methylation but are rich in small regulatory RNAs, such as *Caenorhabditis elegans* and related nematodes[17]. Here we dissect the mechanism behind a parent-of-origin effect on gene expression and provide a physiological context for the emergence of imprinting.

## A TA with a parent-of-origin effect

*C. tropicalis* is a hermaphroditic nematode that—unlike its more widely distributed relative *C. elegans*—inhabits exclusively equatorial regions[18]. While studying genetic incompatibilities between the two *C. tropicalis* wild isolates NIC203 (Guadeloupe, France) and EG6180 (Puerto Rico, USA), we uncovered a maternal-effect TA, which we named *slow-1/grow-1* (ref. 9). This selfish element is located in NIC203 chromosome III and comprises three tightly linked genes: a maternally expressed toxin,

[1]Institute of Molecular Biotechnology of the Austrian Academy of Sciences (IMBA), Vienna BioCenter (VBC), Vienna, Austria. [2]Vienna BioCenter PhD Program, Doctoral School of the University of Vienna and Medical University of Vienna, Vienna, Austria. [3]Department of Biochemistry and Molecular Biology, Institute for Medical Research Israel–Canada, The Hebrew University of Jerusalem, Jerusalem, Israel. [4]Present address: Illumina Artificial Intelligence Laboratory, Illumina, San Diego, CA, USA. [5]These authors jointly supervised this work: Eyal Ben-David, Alejandro Burga. ✉e-mail: alejandro.burga@imba.oeaw.ac.at

*slow-1*, and two identical and redundant antidotes, *grow-1.1* and *grow-1.2*, which are expressed zygotically. For simplicity, we will refer to the two antidotes collectively as *grow-1* unless specifically noted (Extended Data Fig. 1a and Supplementary Discussion). *Slow-1* transcripts are maternally loaded into eggs prior to fertilization and remain stable in embryos, at least until the 20-cell stage. However, from the comma stage until hatching, *slow-1* transcripts are found only in the germline precursor cells[9]. SLOW-1 is homologous to nuclear hormone receptors, whereas the antidote GROW-1 has no homology to known proteins. In crosses between TA carrier and non-carrier strains, heterozygous mothers poison all their eggs but only progeny that inherit the TA can counteract the toxin by zygotically expressing its antidote (Extended Data Fig. 1b). Whereas wild-type worms typically take two days to develop from the L1 stage to the onset of egg laying, embryos poisoned by maternal SLOW-1 take on average four days. This developmental delay imposes a high fitness cost and favours the spread of the selfish element in the population[9].

To study the inheritance of *slow-1/grow-1* TA, we previously generated a near-isogenic line strain (hereafter referred to as 'NIL') containing the *slow-1/grow-1* NIC203 chromosome III locus in an otherwise EG6180 background[9]. As expected, *slow-1* mRNA was detected in the NIL but not in EG6180 (Extended Data Fig. 1c). As previously reported, in crosses between NIL hermaphrodites and EG6180 males, the toxin induced developmental delay in all the $F_2$ homozygous non-carrier (EG/EG) individuals[9] (100% delay, $n = 34$; Fig. 1a). However, we noticed an unexpected pattern of inheritance when performing the reciprocal cross. If EG6180 hermaphrodites were mated to *slow-1/grow-1* NIL males, most of their $F_2$ EG/EG progeny were not developmentally delayed but phenotypically wild type (9.4% delay, $n = 53$; $P \leq 0.0001$; Fig. 1a,b). This was surprising, because known TAs—including *C. elegans peel-1/zeel-1* and *sup-35/pha-1*, the *Medea* locus in *Tribolium*, and the mouse homogeneously staining region (HSR) locus—affect non-carrier individuals regardless of whether the element is inherited from the maternal or paternal lineage[8,19–21] (Extended Data Fig. 1d).

We also investigated the inheritance pattern of two recently discovered maternal-effect TAs in *C. tropicalis* and *C. briggsae* that cause developmental delay[9,22]. However, we found no evidence of a parent-of-origin effect, indicating that this is not a general feature of non-lethal toxins (Fig. 1b and Extended Data Fig. 1e). Mito-nuclear incompatibilities could not explain the observed pattern because both parental lines carry the same mito-genotype (Extended Data Fig. 1f). Moreover, *C. tropicalis*, like all nematodes of the Rhabditida group, lacks de novo methyltransferases, making the involvement of mammalian-like epigenetic imprinting unlikely[23]. Because parent-of-origin effects are extremely rare in nematodes and all reported cases involve transgenic reporters, we set out to investigate this phenomenon[24,25].

## Reduced dosage of the SLOW-1 toxin

Maternally expressed *slow-1* causes the *slow-1/grow-1* TA delay phenotype. Thus, we reasoned that the parent-of-origin effect could stem from reduced expression of the paternally inherited toxin in the germline of $F_1$ heterozygous mothers. To test this idea, we performed reciprocal crosses between EG6180 and the *slow-1/grow-1* NIL strains, followed by RNA sequencing (RNA-seq) of $F_1$ heterozygous young adult hermaphrodites. In agreement with our hypothesis, *slow-1* mRNA levels were significantly lower in $F_1$ mothers when *slow-1/grow-1* was paternally inherited (2.4-fold decrease, $P = 0.0092$; Fig. 1c). The *slow-1* parent-of-origin effect was not exclusive to the recombinant NIL strain, as we observed the same difference in *slow-1* gene expression when performing reciprocal crosses between NIC203 and EG6180 parental strains (Extended Data Fig. 1g).

To independently validate the parent-of-origin effect on gene expression at the protein level, we first tagged the endogenous *slow-1* locus with mScarlet on its N terminus. In agreement with its maternal-effect, SLOW-1 was present in the gonads of hermaphrodites and loaded into

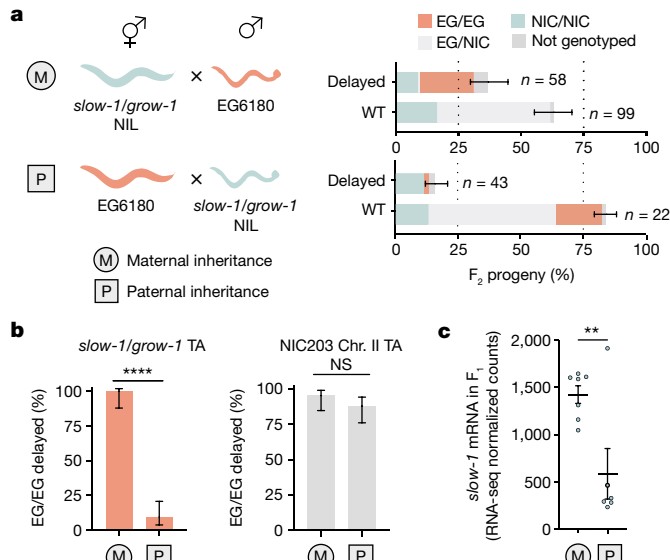

**a**

EG/EG  EG/NIC  NIC/NIC  Not genotyped

**Fig. 1 | *slow-1/grow-1*, a selfish element with a parent-of-origin effect.**
**a**, Reciprocal crosses between the *slow-1/grow-1* TA NIL and the EG6180 parental strain. Maternal (M) or paternal (P) inheritance refers to the *slow-1/grow-1* locus. Worms with a significant developmental delay or larval arrest were categorized as delayed, otherwise they were classified as wild type (WT). Sample sizes ($n$) are shown for each phenotypic class. Error bars indicate 95% binomial confidence intervals calculated with the Agresti–Coull method. Each cross was performed independently at least twice with identical results (see Supplementary Table 1 for raw data). **b**, Activity of the NIC203 chromosome II TA in reciprocal crosses. Penetrance of the toxin, the percentage of $F_2$ non-carrier individuals that are phenotypically affected, is used as a proxy for TA activity (*slow-1/grow-1* TA: M, $n = 34$; P, $n = 53$; $P < 0.0001$, NIC chromosome II TA: M, $n = 44$; P, $n = 50$; $P = 0.27$; two-sided Fisher's exact test; data are mean ± 95% confidence interval). Chr., chromosome; NS, not significant. **c**, Reciprocal crosses between the NIL and EG6180 followed by RNA-seq of their $F_1$ progeny indicate that *slow-1* transcripts are more abundant when maternally inherited (two-sided unpaired *t*-test; M, $n = 7$; P, $n = 6$; $P = 0.0092$; data are mean ± s.e.m.).

eggs prior to fertilization (Extended Data Fig. 1h,i). Next, we performed reciprocal crosses between *mScarlet::slow-1* in the NIL background and EG6180 strains and quantified the fluorescence signal in the germline of their $F_1$ progeny. In agreement with both our genetic crosses and RNA-seq experiments (Fig. 1b,c), when *slow-1/grow-1* was paternally inherited, SLOW-1 protein levels were significantly lower in the germline of $F_1$ individuals (Fig. 2a,b), as well as in $F_2$ 2-cell stage embryos (Fig. 2a,c). To test whether the SLOW-1 dosage correlated with the severity of the phenotype, we impaired the antidote function in the parental NIL strain, which expresses twice as much *slow-1* mRNA compared to heterozygous worms (Extended Data Fig. 1a). We found that *grow-1.1(+/−);grow-1.2(−/−)* worms were viable but we could not retrieve any viable *grow-1.1(−/−);grow-1.2(−/−)* individuals among their progeny. The double homozygous mutants arrested as larvae and died before laying eggs, indicating that *slow-1* is dosage-sensitive (Extended Data Fig. 1a). These results show that the lack of activity of *slow-1/grow-1* following its paternal inheritance stems from a reduction in *slow-1* mRNA levels in the germline of $F_1$ hermaphrodites and, consequently, a reduced dosage of the toxin in $F_2$ embryos.

## *slow-1* is transgenerationally repressed

In *C. elegans*, silencing of transgenes can result in the inheritance of the repressed state for multiple generations[26,27]. Typically, this transgenerational effect is mediated by small RNAs (sRNAs) in response to external or internal cues. To test whether sRNAs could underlie the impaired expression of the paternally inherited *slow-1/grow-1* allele, we explored

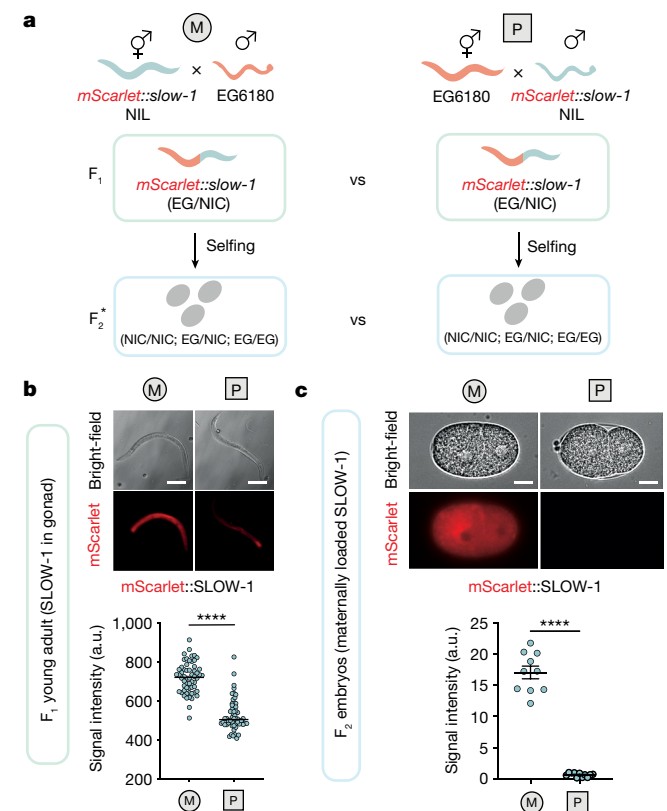

**Fig. 2 | Paternal inheritance of *slow-1/grow-1* leads to decreased SLOW-1 dosage. a**, Reciprocal crosses between *mScarlet::slow-1* NIL and EG6180 strains. The asterisk indicates that the dataset includes only F$_2$ embryos at the 2-cell stage (maternal SLOW-1 protein). **b**, Quantification of total body fluorescence of F$_1$ young adults includes signal from the germline and gut autofluorescence. Each dot represents a single individual. Reciprocal crosses were performed twice (two-sided unpaired *t*-test; M, *n* = 58; P, *n* = 58; *P* < 0.0001; data are mean ± s.e.m.). Representative images of F$_1$ young adults from the reciprocal crosses. Scale bars, 150 μm. a.u., arbitrary units. **c**, Total fluorescence of F$_2$ 2-cell stage embryos from reciprocal crosses between *mScarlet::slow-1* NIL and EG6180 strains. Only maternal SLOW-1 is quantified in early embryos (two-sided unpaired *t*-test; M, *n* = 10; P, *n* = 11; *P* < 0.0001; data are mean ± s.e.m.). Scale bars, 10 μm.

whether the inheritance of *slow-1/grow-1* could compromise its toxicity in subsequent generations. To test this, we first crossed EG6180 hermaphrodites to NIL males and singled their F$_2$ progeny. Then, we identified F$_2$ homozygous *slow-1/grow-1* hermaphrodites, allowed them to self-fertilize, collected their progeny (F$_3$), and crossed them back to EG6180 males. Finally, we collected the F$_4$ heterozygous offspring, allowed them to self-fertilize, and inspected their progeny (F$_5$) (Fig. 3a). In this way, the impaired *slow-1/grow-1* allele was reintroduced into the maternal lineage, which enabled us to probe whether *slow-1* could delay its progeny once again. We found that 97% (*n* = 34) of F$_5$ homozygous *slow-1/grow-1* (EG/EG) individuals were phenotypically wild type, indicating that *slow-1/grow-1* activity was largely impaired 3 generations after paternal inheritance (Fig. 3b). Additional crosses revealed that *slow-1/grow-1* regained its activity 9 generations after paternal inheritance (22.2% (*n* = 27) of EG/EG individuals were phenotypically wild type), indicating that the *slow-1* repressed state can be spontaneously reversed[26,28] (Fig. 3b).

## piRNAs target *slow-1*

Since the transgenerational repression of *slow-1/grow-1* does not stem from an external trigger, we reasoned that endogenous piRNAs could mediate this effect. PRG-1, the *C. elegans* orthologue of *Drosophila*

PIWI-clade proteins, binds piRNAs and is essential for their function[29,30]. To study the role of PIWI and other Argonaute proteins in *slow-1* repression, we first identified homologues and built a comprehensive Argonaute phylogeny (Extended Data Fig. 2). The *C. tropicalis* genome encodes two PRG-1 orthologues on chromosome I, which we named PRG-1.1 and PRG-1.2, both of which are maternally loaded into eggs (Fig. 3c). They share 87.7% protein sequence identity and are probably the result of a recent gene duplication event (Extended Data Figs. 2 and 3a–c). To test whether the repression of *slow-1/grow-1* was dependent on piRNA activity, we generated *prg-1.1* and *prg-1.2* null alleles in an EG6180 background. Both *prg-1.1* and *prg-1.2* mutant lines were viable and did not show any obvious signs of developmental delay or larval arrest (0%, *n* = 118 and 0%, *n* = 95, respectively). However, *prg-1.1;prg-1.2* double mutants were fully sterile indicating significant redundancy between these two genes (Extended Data Fig. 3d).

Next, we set up crosses between EG6180 hermaphrodites and NIL males, in which both parents carried null alleles of either *prg-1.1* or *prg-1.2*. Loss of *prg-1.2* impaired *slow-1/grow-1* repression when inherited through the paternal lineage: 69.2% (*n* = 78) of F$_2$ homozygous EG/EG individuals were developmentally delayed (Fig. 3d). By contrast, loss of *prg-1.1* had only a minor effect on the activity of the TA (6.25% of EG/EG were delayed, *n* = 16) (Fig. 3d). Additional crosses revealed that maternally provisioned *prg-1.2* is necessary for *slow-1* repression (Extended Data Fig. 3e) and that *prg-1.2* is necessary for the initiation but not the maintenance of the repression[31,32] (Extended Data Fig. 3f).

To identify the specific piRNAs responsible for *slow-1* repression, we first annotated 27,445 piRNAs in *C. tropicalis* with a mean abundance of 0.1 ppm or higher (Methods and Supplementary Data 1). As in *C. elegans* and *C. briggsae*, piRNAs were found almost exclusively in chromosome IV[33] (96.9%; Fig. 3e). Next, we leveraged known targeting rules, predicted piRNA–target binding energies and overall complementarity to define a list of top candidates (Methods and Supplementary Data 2). We observed that two of our top piRNA candidate loci were in tight genetic linkage: *Ctr-21ur-06949* and *Ctr-21ur-06917* were only 4.6 kb apart on chromosome IV and both were predicted to bind the 3′ untranslated region (UTR) of *slow-1* (Fig. 3f). To test their role in *slow-1* repression, we deleted these piRNAs and performed paternal crosses. Of note, whereas deletion of individual piRNAs had no effect on *slow-1/grow-1* repression, simultaneous loss of both piRNAs hindered *slow-1/grow-1* repression, phenocopying the *prg-1.2* loss of function mutation (Fig. 3d,f). As a control, only background levels of delay were observed in the parental single and double piRNA mutant strains (*Ctr*-21*ur-06949(Δ)*: 2%, *n* = 100; *Ctr*-21*ur-06917(Δ)*: 1.25%, *n* = 80; *Ctr*-21*ur-06949(Δ)*; *Ctr*-21*ur-06917(Δ)*: 2.2%, *n* = 180). These results show that piRNAs repress *slow-1* following its paternal inheritance and that their activity is epistatic.

## PRG-1s are redundant but non-equivalent

Given the epistatic nature of the *slow-1* piRNA-mediated repression (Fig. 3f) and the synthetic lethality observed in *prg-1.1* and *prg-1.2* double mutants (Extended Data Fig. 3d), we hypothesized that any role of *prg-1.1* in the repression of *slow-1* might be masked by genetic redundancy. To test this idea, we generated triple mutant worms carrying the *prg-1.1* mutant allele along with the double piRNA deletion and performed paternal inheritance crosses of the TA. In contrast to the partial de-repression of *slow-1/grow-1* observed in either *prg-1.2* or double piRNA mutants (Fig. 3f), de-repression of the TA was almost complete when *prg-1.1* and the two piRNAs were mutated (Fig. 3f). As a control, no developmental defects were observed in the triple mutant parental line (0%, *n* = 89). Immunoprecipitation of PRG-1.1 and PRG-1.2 followed by sRNA sequencing (sRNA-seq) revealed that these Argonautes bind at large the same piRNA population—including both *Ctr-21ur-06949* and *Ctr-21ur-06917* (Extended Data Fig. 4 and Supplementary Data 2). However, their binding preference are not entirely equivalent, probably

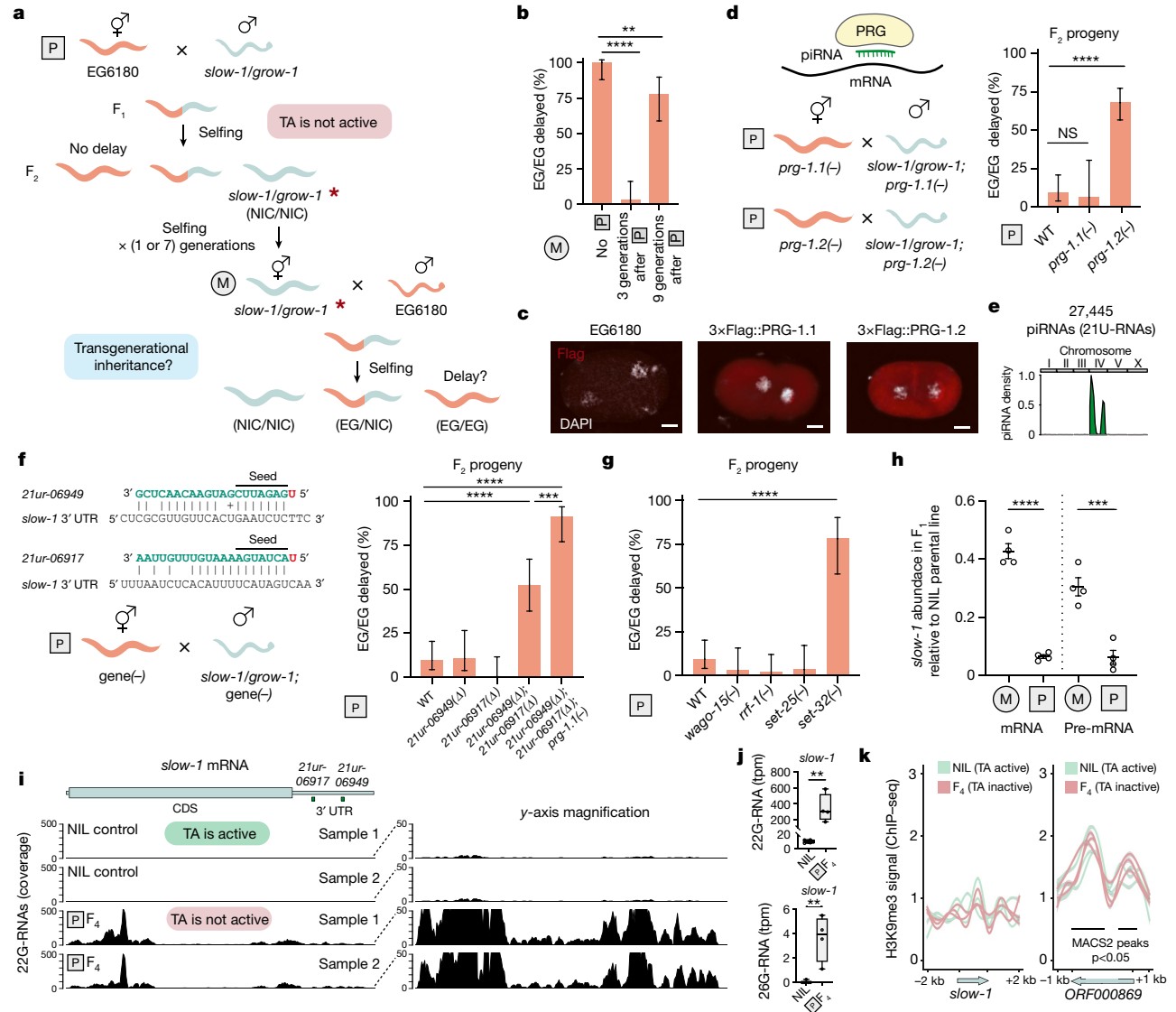

**Fig. 3 | *slow-1* is targeted by the piRNA pathway. a**, Crossing scheme to test transgenerational inheritance of *slow-1/grow-1* repression. Red asterisk denotes repressed allele. **b**, Comparison of *slow-1/grow-1* activity with no paternal inheritance and 3 and 9 generations following paternal inheritance (no P, *n* = 34; 3 generations, *n* = 34; 9 generations, *n* = 27; two-sided Fisher's exact test; ****P* < 0.0001 and ***P* = 0.0053; data are mean ± 95% confidence interval). **c**, Representative immunostaining images for 3×Flag::PRG-1.1 and 3×Flag:: PRG-1.2 lines in 2-cell stage embryos. EG6180 as negative control. Quantification in Extended Data Fig. 3c. Scale bars, 10 μm. **d**, Effect of *prg-1.1* or *prg-1.2* (WT, *n* = 53; *prg-1.1(–)*, *n* = 16; prg-1.2(–), *n* = 75; two-sided Fisher's exact test; NS, *P* > 0.99 and ****P* < 0.0001; data are mean ± 95% confidence interval) null mutations in *slow-1/grow-1* paternal inheritance. **e**, Genome-wide distribution of *C. tropicalis* piRNAs. **f**, Left, scheme showing piRNA candidates binding to the 3′ UTR of *slow-1*. + Denotes a G:U wobble base pair. Right, paternal crosses between strains with various combinations of piRNAs and *prg-1.1* mutations (WT, *n* = 53; *21ur-06949(Δ);21ur-06917(Δ)*, *n* = 40; *21ur-06949(Δ);21ur-06917(Δ);prg-1-1(–)*, *n* = 34; two-sided Fisher's exact test; ****P* < 0.0001 and ****P* = 0.0003; data are

mean ± 95% confidence interval). **g**, Testing the requirement for components of the piRNA pathway in *slow-1* repression (WT, *n* = 53; *set-32*, *n* = 23; two-sided Fisher's exact test; *P* < 0.0001; data are mean ± 95% confidence interval. **h**, RT–qPCR quantification of *slow-1* mRNA and pre-mRNA abundance from reciprocal crosses between NIL and EG6180 normalized to parental NIL (M, *n* = 4; P, *n* = 4; two-sided unpaired *t*-test; ****P* < 0.0001 and ****P* = 0.0008; data are mean ± s.e.m.). **i**, Coverage of 22G-RNAs mapping to *slow-1* mRNA in licensed or repressed states (*n* = 4). Two repeats are shown for simplicity (total number of aligned 22G-RNAs per library is the same). **j**, Quantification of 22G-RNA and 26-RNA populations mapping to *slow-1* (*n* = 4) (two-sided unpaired *t*-test; 22G-RNA: *P* = 0.0013; 26G-RNA: *P* = 0.0013). In box plots, the centre line is the mean, box edges represent first and third quartile boundaries, and whiskers extend to minimum and maximum values. tpm, transcripts per million mapped reads. **k**, H3K9me3 ChIP–seq in samples in which *slow-1* is licensed or repressed. Lines correspond to the ratio of H3K9me3 ChIP over chromatin input coverage normalized by their respective library sizes. Left, there are no apparent peaks at the *slow-1* locus. Right, example of reproducible peaks identified by MACS2.

contributing to their differential effects on *slow-1* repression (Extended Data Fig. 4 and Supplementary Note 1).

## *slow-1* is epigenetically repressed

In *C. elegans*, piRNAs trigger the production of secondary 22G-RNAs that are complementary to the target mRNA. These 22G-RNAs are

bound by nuclear Argonautes HRDE-1 and WAGO-10, which in turn recruit chromatin-modifying enzymes to the target locus and mediate its epigenetic repression[34,35] (Extended Data Fig. 5a). Two putative histone methyltransferases, SET-25 (H3K9me3) and SET-32 (H3K23me3), have a crucial role in this process[35,36]. To test whether effectors of the piRNA pathway mediate *slow-1* repression, we generated putative null alleles of several known factors (Fig. 3g and Extended Data Fig. 5a,d).

*Ctr-hrde-1* (the closest homologue of *hrde-1* and *wago-10* in *C. elegans*), *Ctr-simr-1* and *Ctr-mut-16* were essential for fertility, preventing further characterization (Extended Data Fig. 5a–d). However, *Ctr-rrf-1*, *Ctr-wago-15* (a close paralogue of *Ctr-hrde-1*), *Ctr-set-25* and *Ctr-set-32* mutants were viable and fertile. We set up paternal crosses using these four mutants and found that loss of *Ctr-set-32* was sufficient to impair *slow-1/grow-1* repression when inherited through the paternal lineage, phenocopying loss of piRNAs (Fig. 3g and Extended Data Fig. 5e). The involvement of a histone methyltransferase in the parent-of-origin effect strongly suggested that *slow-1* repression occurs at the transcriptional level. In agreement with this model, we performed quantitative PCR with reverse transcription (RT–qPCR) and found that both *slow-1* mRNA and pre-mRNA levels were markedly reduced in the $F_1$ generation following paternal inheritance of the TA (15.2% and 20.2% of levels following maternal inheritance, respectively; Fig. 3h).

Next, we asked whether paternal inheritance of *slow-1/grow-1* leads to the accumulation of 22G-RNAs targeting *slow-1*. To test this, we leveraged the transgenerational inheritance of the *slow-1/grow-1* repressed state (Fig. 3a). First, we carried a *slow-1/grow-1* paternal cross, we then isolated $F_2$ homozygous TA carriers, propagated and expanded the population for two generations, and finally sequenced the sRNA pool of the $F_4$ young hermaphrodites (Extended Data Fig. 5f). Paternal inheritance of *slow-1/grow-1* resulted in a marked 33.7-fold up-regulation of 22G-RNAs complementary to *slow-1* compared to the control line, in which the TA is active (Fig. 3i,j; $P = 0.0013$). We observed a local peak of 22G-RNA biogenesis within the predicted piRNA recognition sites (Extended Data Fig. 5g); however, most 22G-RNAs were derived from the 5′ of the transcript (Fig. 3i). We also identified 26G-RNAs complementary to *slow-1* in the repressed state. These 26G-RNAs were significantly less abundant than 22G-RNAs but were almost completely absent from control samples (Fig. 3j). Since 22G-RNAs were readily detectable in the great-great-granddaughters of the original male TA carriers and most of them were not derived from predicted piRNA binding sites (Fig. 3i), our results suggest that these sRNAs mediate the inheritance of the *slow-1* epigenetic state[37]. Next, given that *set-25* and *set-32* are jointly required for the deposition of H3K9me3 in *C. elegans*[38], we investigated whether paternal inheritance of the TA could lead to the accumulation of this repressive histone mark in *slow-1* (refs. 32,39). To test this, we performed H3K9me3 chromatin immunoprecipitation with sequencing (ChIP–seq) in $F_4$ individuals following paternal inheritance of the TA, as well as in the parental NIL control. We did not observe any significant H3K9me3 enrichment in *slow-1*, even though we detected H3K9me3 enrichment in other loci (Fig. 3k). Although we cannot rule out potential limitations of our assay such as dilution of the germline signal or unspecific binding of the antibody, this result suggests that H3K9me3 may not be required for the maintenance of silencing, in line with recent findings[40–42].

## Maternal *slow-1* mRNAs counter piRNAs

The repressive action of piRNAs accounts for the low levels of *slow-1* following paternal inheritance; however, piRNAs alone cannot explain the parent-of-origin effect. Thus, we investigated whether mechanisms that are known to facilitate the expression of genes in the *C. elegans* germline—periodic 10-bp motif of $A_n/T_n$ clusters (PATCs) and CSR-1—might prevent *slow-1* repression. PATCs are typically found in the introns of germline-expressed genes and can promote the expression of transgenes in the germline[43,44]. However, *slow-1* introns exhibited very low PATC scores (Extended Data Fig. 6a). CSR-1 is the only Argonaute that can activate transgenes silenced by piRNAs; however, loss of maternal CSR-1 did not impair *slow-1* activity, suggesting that CSR-1 is not responsible for the parent-of-origin effect (Extended Data Fig. 6 and Supplementary Note 2).

While performing reciprocal crosses between the wild-type NIL and a *slow-1(−)/grow-1(−)* double mutant NIL strain, we made an intriguing observation. Analogous to crosses between NIL hermaphrodites and EG6180 males (Fig. 1b), when wild-type NIL hermaphrodites were mated to the double mutant males in which both toxin and antidote carry null frameshift mutations, 28.9% ($n = 190$) of the $F_2$ progeny were delayed and all homozygous double mutant individuals were delayed (100%, $n = 31$; Fig. 4a). However, we observed the same inheritance pattern in the reciprocal cross. 22.1% ($n = 140$) of the $F_2$ progeny were delayed and delayed individuals were homozygous double mutants (95.8%; $n = 24$), indicating that *slow-1/grow-1* was fully active when inherited via the paternal lineage (Fig. 4a). These results indicated that the *slow-1/grow-1* double mutant and EG6180 haplotypes were not equivalent, and that maternal inheritance of a null *slow-1* allele could somehow prevent its piRNA-mediated repression. Furthermore, given that *slow-1* was able to protect the paternal allele from repression despite carrying a frameshift null mutation, we hypothesized that *slow-1* mRNA, but not SLOW-1 protein, is necessary for this phenomenon.

To test this hypothesis, we used CRISPR–Cas9 to delete the full coding region of *slow-1* in an otherwise identical genetic background to the double mutant NIL strain carrying *slow-1(−)/grow-1(−)*. In contrast to the frameshift allele, the deletion allele (*slow-1Δ*) removes the entirety of the *slow-1* transcript. Then, we performed reciprocal crosses between the *slow-1(Δ)/grow-1(−)* NIL strain and the wild-type NIL and inspected their $F_2$ progeny (Fig. 4b). When NIL hermaphrodites were crossed to *slow-1(Δ)/grow-1(−)* NIL males, we observed 26.8% ($n = 190$) delay among the $F_2$ offspring, whereas all genotyped worms homozygous for the mutant allele were delayed (100%, $n = 18$; Fig. 4b). By contrast, when *slow-1(Δ)/grow-1(−)* NIL hermaphrodites were crossed to NIL males, we observed baseline delay among their $F_2$ progeny (3.8%, $n = 129$) (Fig. 4b). These results indicate that the *slow-1(−)* allele but not *slow-1(Δ)* is able to protect a paternally inherited slow-*1/grow-1* TA from piRNA repression and identify *slow-1* mRNA as the 'licensing' signal.

To test whether *slow-1* mRNA is sufficient for licensing, we transcribed *slow-1* RNA in vitro and injected it into the gonads of 16 *slow-1(Δ)/grow-1(−)* NIL hermaphrodites, mated those to *slow-1/grow-1* NIL males, and inspected their $F_2$ progeny. Critically, we mutated the start codon of the *slow-1* cDNA that served as a transcription template, resulting in RNA that cannot be translated into SLOW-1 protein. Following injection of noncoding *slow-1* RNA into the gonads of *slow-1(Δ)/grow-1(−)* mothers, there was a significant increase in the proportion of delayed individuals among the $F_2$ compared to a control injection (13.8% delayed, $n = 650$ and 4% delayed, $n = 296$ respectively, $P < 0.0001$; Fig. 4c). Most (84.7%, $n = 61$) of genotyped delayed individuals were homozygous for the *slow-1(Δ)/grow-1(−)* allele, showing that the effect was highly specific. Overall, injection of *slow-1* RNA increased the proportion of delayed $F_2$ individuals among double mutants from 9.6% ($n = 62$) in the control cross to 47.6% ($n = 128$) in the RNA-injected animals ($P < 0.0001$; Fig. 4f). The partial rescue of zygotic *slow-1* expression probably reflects technical limitations in the injection protocol, as we observed a wide range of rescue depending on the injected mother (Extended Data Fig. 7a). These results show that *slow-1* RNA is sufficient to license a paternally inherited slow-*1/grow-1* allele and that this effect does not depend on SLOW-1 protein. Epigenetic licensing by maternal transcripts has only been described for one gene to date: the *C. elegans* sex-determining gene *fem-1* (ref. 45) (Supplementary Discussion). Because licensing could be a common mechanism in nematodes and offers a physiological framework to better understand other epigenetic phenomena, we sought to study its requirements[25,46]

## Molecular requirements of licensing

First, we studied the effect of maternal *slow-1* dosage on licensing. To do so, we deleted 620 bp upstream of the *slow-1* coding region and then proceeded to knock out the antidote *grow-1*. RNA-seq of the resulting promoter deletion strain revealed a 176-fold decrease in *slow-1* mRNA levels, whereas neighbouring genes were unaffected (Extended

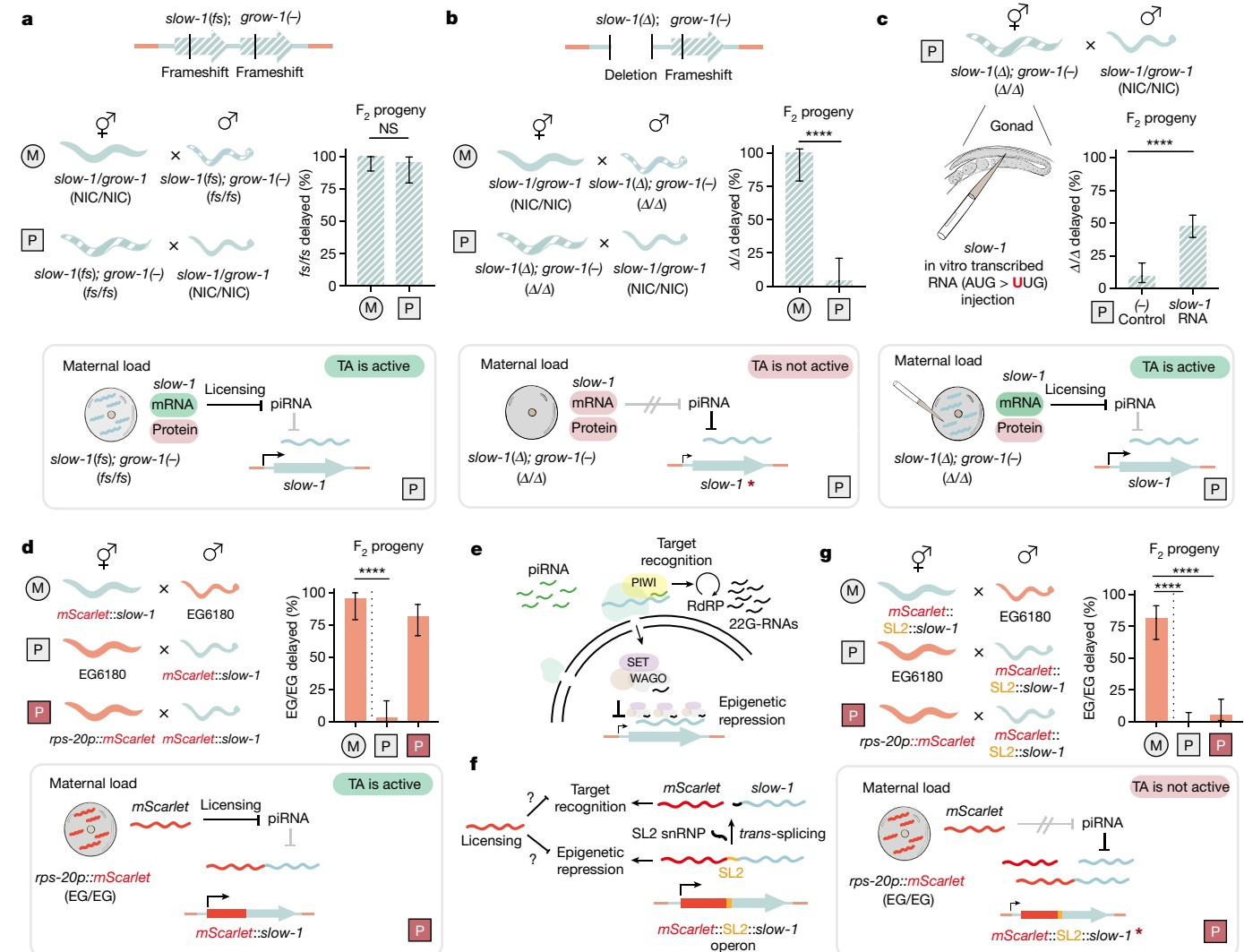

**Fig. 4 | Maternal *slow-1* transcripts inhibit piRNA-mediated repression.**
**a**, Top, in the *slow-1(fs)/grow-1(−)* double mutant NIL strain, both *slow-1* and *grow-1* carry frameshift (fs) mutations. Bottom, TA activity is observed regardless of maternal or paternal inheritance (M, *n* = 31; P, *n* = 24; two-sided Fisher's exact test; *P* = 0.43; data are mean ± 95% confidence interval). **b**, Top, in the *slow-1(Δ)/grow-1(−)* mutant NIL strain, the full *slow-1* gene (including coding sequence and UTR) is deleted, and *grow-1* carries a frameshift mutation. Bottom, *slow-1/grow-1* is only active when maternally inherited (M, *n* = 18; P, *n* = 25; two-sided Fisher's exact test; *P* < 0.0001; data are mean ± 95% confidence interval).
**c**, In vitro transcribed *slow-1* RNA with a mutated start codon was injected in the gonad of *slow-1(Δ)/grow-1(−)* double mutant NILs and later crossed to NIL males. Approximately half of their *Δ/Δ* F$_2$ progeny were delayed. Control mothers were injected with DEPC H$_2$O (slow-1 RNA, *n* = 128; control, *n* = 62; two-sided Fisher's exact test *P* < 0.0001; data are mean ± 95% confidence interval). **d**, Reciprocal crosses between worms carrying an N-terminally tagged *mScarlet::slow-1* and EG6180 (top and middle crosses). Maternal *mScarlet* expression (*rps-20* p::*mScarlet::rps-20* 3′ UTR chromosome V) licenses paternal *mScarlet::slow-1*

(bottom cross). Maternal *mScarlet::slow-1*; *n* = 23; paternal maternal *mScarlet::slow-1*, *n* = 31; maternal *mScarlet*, paternal maternal *mScarlet::slow-1*, *n* = 38; two-sided Fisher's exact test; *P* < 0.0001; data are mean ± 95% confidence interval. **e**, Schematic of the *C. elegans* piRNA pathway. Target recognition and secondary sRNA amplification depend on the target mRNA, whereas epigenetic repression depends on complementarity to the nascent transcript of the target. **f**, Schematic of the *mScarlet*::SL2::*slow-1* operon. The operon is transcribed as a single polycistronic transcript and later *trans*-spliced into two independent mRNA transcripts. Licensing could counter piRNA-mediated repression either during target recognition (mRNA) or epigenetic repression (nascent transcript). snRNP, small nuclear ribonucleoprotein particle. **g**, Reciprocal crosses between worms carrying the *mScarlet*::SL2::*slow-1* operon and EG6180 (top and middle crosses). Maternal *mScarlet* expression does not license a paternally inherited *mScarlet*::SL2::*slow-1* (bottom cross). Maternal *mScarlet*::SL2::*slow-1*, *n* = 32; paternal *mScarlet*::SL2::*slow-1*, *n* = 49; maternal *mScarlet*, paternal *mScarlet*::SL2::*slow-1*, *n* = 37; two-sided Fisher's exact test; ****P* < 0.0001; data are mean ± 95% confidence interval.

Data Fig. 7b). We observed only limited abnormal phenotypes in the double mutants, even though they lacked the *grow-1* antidote (8.71% delay, *n* = 70), suggesting that the amount of SLOW-1 toxin made in the promoter deletion line was insufficient to poison embryos. However, when we crossed *slow-1(Δprom)/grow-1(−)* hermaphrodites to NIL males, the paternal allele was fully active: 27.7% of F$_2$ individuals were delayed (Extended Data Fig. 7c). These results indicate that a 176-fold reduction in *slow-1* maternal mRNA abundance abolishes its toxicity but not its licensing activity, suggesting that licensing does not rely

on a sponge-like mechanism but probably involves a catalytic step (Extended Data Fig. 7c).

Sequence similarity between *slow-1* maternal transcripts and their zygotic counterparts is probably key for the establishment of licensing. To explore whether this requirement is an intrinsic property of *slow-1* or a general feature, we asked whether sequence similarity to a foreign sequence could also license *slow-1*. To do so, we took advantage of the *mScarlet*::slow-1 fusion strain (Fig. 2a). Importantly, tagging of SLOW-1 with an N-terminal mScarlet reporter did not interfere

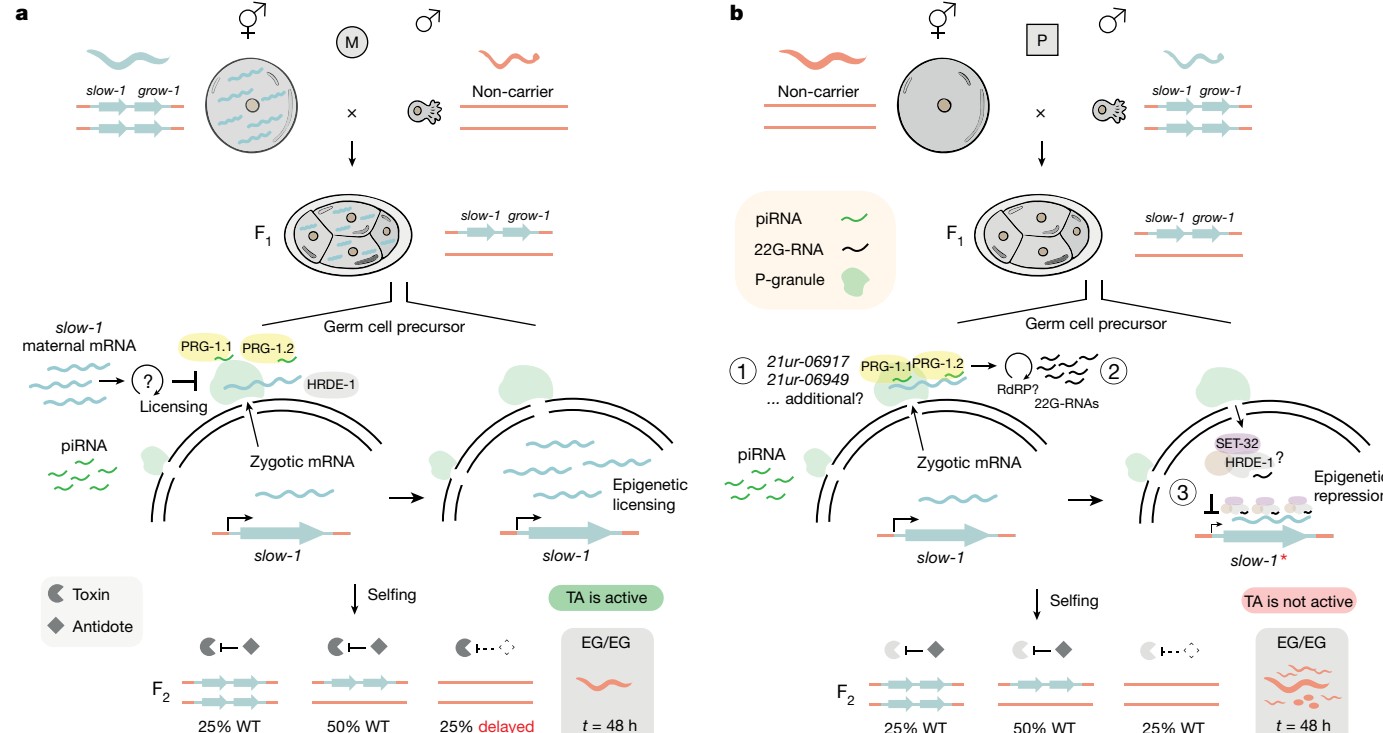

**Fig. 5 | Model illustrating the *slow-1/grow-1* parent-of-origin effect.**
**a**, Maternal inheritance of the *slow-1/grow-1* TA. *slow-1* transcripts deposited in the egg by the mother are sufficient and necessary to activate zygotic *slow-1* in the germline of the $F_1$ progeny. Epigenetic licensing stems from inhibiting the repressive action of piRNAs. Licensing occurs post-transcriptionally, probably by inhibiting piRNA-target recognition or secondary sRNA amplification in the perinuclear nuage (green condensates). $F_1$ heterozygous mothers load SLOW-1 toxin into all their eggs. $F_2$ homozygous non-carrier individuals are developmentally delayed because they do not express the zygotic antidote.
**b**, Paternal inheritance of the *slow-1/grow-1* TA. In the absence of *slow-1* maternal transcripts, piRNAs repress the transcription of *slow-1* in the germline of

heterozygous $F_1$ mothers. Initiation of repression requires maternal PRG-1 activity, which uses the *slow-1* zygotic transcript as a template for the generation of 22G-RNAs complementary to the target. These 22G-RNAs are then probably bound by nuclear Argonaute proteins, such as HRDE-1, which in turn recruit chromatin-modifying enzymes to the target locus. The histone methyltransferase SET-32, a known co-factor of HRDE-1 in *C. elegans*, is necessary to repress *slow-1*. This epigenetic repression results in decreased transcription and SLOW-1 levels that are insufficient to poison $F_2$ homozygous non-carrier progeny. The repressed state of *slow-1(*)* is transgenerationally inherited for more than five generations.

with its toxicity or the parent-of-origin effect (Fig. 4d). To emulate the licensing signal, we first generated a strain carrying *mScarlet* in a germline-permissive site (chromosome V) in an otherwise EG6180 background. We then crossed hermaphrodites expressing maternal *mScarlet* to *mScarlet::slow-1* males and scored their $F_2$ progeny. Notably, maternal *mScarlet* transcripts fully licensed endogenous tagged *mScarlet::slow-1* (Fig. 4d). These results indicate that sequence similarity to a foreign maternal transcript is sufficient for epigenetic licensing. Moreover, they suggest that the licensing signal can spread through the zygotic transcript, as maternal *mScarlet* countered piRNAs targeting *slow-1* despite the lack of sequence similarity between the two genes.

Finally, we set out to investigate at what step of the piRNA pathway licensing countered repression: target recognition or transcriptional silencing. Target recognition depends on complementarity to the mature mRNA, whereas transcriptional silencing relies on complementarity to the nascent transcript, which guides the repression machinery to the target locus (Fig. 4e). We reasoned that we could distinguish between these possibilities by testing whether maternal *mScarlet* could license *slow-1* in the context of a polycistronic operon[47]. To do this, we inserted the 256-bp intergenic region from the *C. tropicalis gpd-2::gdp-3* operon in between *mScarlet* and *slow-1* using CRISPR–Cas9. This intergenic sequence (hereafter termed SL2) contains the 3′ acceptor site for the SL2 RNA *trans*-splicing leader[48]. The resulting operon, *mScarlet::SL2::slow-1*, is under the control of the native *slow-1* promoter−*mScarlet* and *slow-1* are transcribed as a single polycistronic

pre-mRNA in the germline and later trans-spliced into two independent mRNAs (Fig. 4f). As expected, we detected mScarlet in the germline of these worms and their early embryos (Extended Data Fig. 7d).

To validate our approach, we performed reciprocal crosses between *mScarlet::SL2::slow-1* worms and the EG6180 parental strain and found that *slow-1* was active only when maternally inherited, indicating that the operon architecture did not interfere with the parent-of-origin effect (Fig. 4g). Furthermore, lack of maternal *slow-1* led to co-repression of mScarlet when the operon was paternally inherited, in agreement with silencing being guided by the nascent transcript (Extended Data Fig. 7e). We then crossed hermaphrodites expressing maternal *mScarlet* mRNA to males carrying the *mScarlet::SL2::slow-1* operon and scored their $F_2$ progeny. We observed no delayed EG/EG $F_2$ individuals, indicating that homology to *mScarlet* was not sufficient to license *slow-1*, despite being part of the same pre-mRNA molecule. Given that maternal *mScarlet* mRNA efficiently licensed *slow-1* when both genes were part of a monocistronic transcript, our results indicate that zygotic *slow-1* is licensed post-transcriptionally. For instance, licensing could hinder the binding of piRNAs to their target or the subsequent amplification of 22G-RNAs in the perinuclear nuage. One implication of this model is that licensing should be incapable of countering transcriptional silencing mediated by pre-existing 22G-RNAs. Supporting this idea, maternal *slow-1* transcripts originating from a repressed allele lost their ability to license a naïve paternal allele (Extended Data Fig. 7f), presumably because repressive 22G-RNAs that are loaded into eggs alongside maternal transcripts[49] can effectively by-pass the licensing signal.

## From piRNAs to parent-of-origin effects

Haig's kinship theory explains why natural selection favours different levels of expression of maternally and paternally inherited alleles. However, it does not address how these epigenetic differences evolve in the first place. Here we show that in the nematode *C. tropicalis*, parent-specific expression originates by co-option of the piRNA pathway, which in worms is essential to distinguish self from non-self[32,35,50] (Fig. 5). Similar to classical imprinting, *slow-1* expression levels depend on whether the TA is maternally or paternally inherited. However, there are two important differences: (1) the *slow-1* parent-of-origin effect is not acquired by gametic identity but specifically triggered by outcrossing; and (2) imprinted loci reset in the germline every generation, whereas *slow-1* repression resets only after multiple generations of selfing. We propose that this parent-of-origin effect could represent an intermediate evolutionary state, which we refer to as proto-imprinting.

Our results also indicate that parent-of-origin effects could provide a selective advantage to the host. Repression of *slow-1* following paternal inheritance of the TA hinders its gene drive activity for multiple generations and decreases the incidence of intraspecific genetic incompatibilities. Remarkably, an evolutionary related but highly divergent TA, *slow-2/ grow-2*, does not show a parent-of-origin effect, suggesting that this trait can evolve quickly in nature (Extended Data Figs. 8 and 9 and Supplementary Note 3). Because TAs and analogous maternal-zygotic lethal factors are not only present in nematodes but also segregate in wild insect, plant, and mouse populations[7,20,21,51,52], we propose that co-option of sRNA-mediated defence systems originating from selfish conflict might be a recurrent event facilitating the evolution of imprinting.

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

## Methods

### Maintenance of worm strains

Nematodes were grown on modified nematode growth medium (NGM) plates with 1% agar/0.7% agarose to prevent *C. tropicalis* burrowing. Experiments were conducted at either 25 °C (*C. tropicalis*) or 20 °C (*C. elegans*). *csr-1(+/−)* strains were cultured on 6-cm NGM plates supplemented with 500 µl of G418 (25 mg ml$^{-1}$) for selecting heterozygous null individuals. Supplementary Table 2 lists all study strains, some of which were provided by the Caenorhabditis Genetics Centre, funded by the NIH Office of Research Infrastructure Programs (P40 OD010440).

### Phenotyping and genotyping of crosses

For crosses, 4–5 L4 hermaphrodites were mated with 30–40 males in a 12-well plate with modified NGM. After 2 days, 10 L4 $F_1$ progeny were transferred to separate plates, genotyped by PCR, and at least 10 embryos per $F_1$ hermaphrodite were singled into 6-cm NGM plates. Each F2 individual was visually inspected daily for up to 7 days, classified for developmental stage, and any phenotypic abnormalities. Embryonic lethality, arrested development, and delayed reproduction were assessed. Sterility was noted for adults not producing progeny. After 7 days, worms were lysed and genotyped. A list of primers used for genotyping can be found in Supplementary Table 3. Crosses involving *csr-1(−); slow-1/grow-1* hermaphrodites vs EG6180 males or injected hermaphrodites vs NIL males were selected based on a *pmyo-2::mScarlet* reporter.

### Generation of *C. tropicalis* transgenic lines

For CRISPR–Cas gene editing, we adapted previous protocols[53]. In brief, 250 ng µl$^{-1}$ Cas9 or Cas12a proteins were incubated with 200 ng µl$^{-1}$ CRISPR RNA (crRNA) and 333 ng µl$^{-1}$ *trans*-activating crRNA (tracrRNA) before adding 2.5 ng µl$^{-1}$ co-injection marker plasmid (pCFJ90-mScarlet-I). For HDR, donor oligos (IDT) or biotinylated and melted PCR products were added at a final concentration of 200 ng µl$^{-1}$ or 100 ng µl$^{-1}$, respectively. Following injections into young hermaphrodites, mScarlet-positive F1 were singled, and their offspring screened by PCR and Sanger sequencing to detect successful editing. To clone the mScarlet::SLOW-1 donor, we added ~300-bp homology arms amplified from QX2345 genomic DNA to mScarlet-I (from pMS050) in pBluescript via Gibson assembly. Because *csr-1* is essential for viability in *C. elegans*, we first devised a strategy to stably propagate a *csr-1* heterozygous line in the absence of classical genetic balancers. To do so, we used CRISPR–Cas9 to introduce a premature stop mutation in the endogenous *csr-1* locus followed by a *neoR* cassette, which confers resistance to the G418 antibiotic (Extended Data Fig. 6d). For the *csr-1::neoR* donor, we first replaced the *C. elegans rps-27* promoter and *unc-54* 3′ UTR in pCFJ910 with 500 bp upstream and 250 bp downstream of the *C. tropicalis rps-20* gene. This *rps-20::neoR* cassette was then flanked with ~550-bp homology arms amplified from EG6180 worms and inserted into pBluescript. Correct targeting introduces a stop codon after residue L337 of CSR-1 followed by a ubiquitously expressed neomycin resistance. We propagated the mutant line in plates containing G418 and thus actively selecting for heterozygous *csr-1(−)* null individuals. Upon drug removal, most homozygous *csr-1(−)* individuals derived from heterozygous mothers developed into adulthood but were either sterile or laid mostly dead embryos. However, a small fraction of null mutants was partially fertile and homozygous *csr-1(−)* lines could be stably propagated for multiple generations despite extensive embryonic lethality in the population (Extended Data Fig. 6d). All gRNAs and HDR templates are available on Supplementary Tables 4 and 5.

### In vitro RNA transcription and injection

The *slow-1* cDNA was cloned into pGEM-T Easy (Promega, A1360), with a 5′ T7 RNA polymerase site and the start codon mutated RNA-only transcription (ATG>TTG). The plasmid was digested with NotI to release the insert (NEB, R0189), which was subsequently purified by gel-extraction and used as template for RNA synthesis. RNA was prepared using the HiScribe T7 Quick High Yield kit (NEB, E2050) with the following modifications: addition of 3 µl of 10 mM DTT and 1 µl of RNaseOUT (Thermo, 10777019). After overnight transcription, the reaction was diluted, treated with RNase-free DNase I (NEB, M0303S), bead-purified (Vienna Biocenter MBS 5001111, High Performance RNA Bead Isolation), quantified (Thermo, Q32852), and stored at −80 °C. Injections were repeated twice using independently transcribed RNA at concentrations: 150 nM and 400 nM yielding identical results.

### Reciprocal crosses with the *mScarlet::slow-1* reporter line

To assess SLOW-1 expression in $F_1$ progeny from reciprocal crosses between mScarlet::SLOW-1 NIL and EG6180 strains, we conducted 2 sets of crosses: (1) SLOW-1::mScarlet *dpy* (INK461) hermaphrodites to EG6180 males for maternal inheritance; and (2) EG6180 *dpy* (QX2355) hermaphrodites to mScarlet::SLOW-1 NIL males (INK459) for paternal inheritance. Wild-type young adult $F_1$ progeny were immobilized in NemaGel on a glass slide and imaged using an Axio Imager.Z2 (Carl Zeiss) widefield microscope with a Hamamatsu Orca Flash 4 camera, (excitation 545/30 nm filter). The analysis was performed in FIJI, by tracing the germline in the DIC channel and measuring mean fluorescence, including gut autofluorescence.

### Sequencing and genome assembly of EG6180

We extracted high molecular weight genomic DNA using the Masterpure Complete DNA and RNA purification kit (tissue sample protocol, Lucigen). We prepared 8 kb, 20 kb and unfragmented sequencing libraries using the 1D Ligation Sequencing Kit (Oxford Nanopore SQK-LSK109). The 8 kb fragmentation was done using g-TUBE (Covaris). Library was loaded on a MinION MK1B device (Oxford Nanopore). Read calling was done using MinKNOW software. We performed a hybrid assembly, incorporating Illumina sequencing reads of EG6180 with some modifications as detailed below[9]. We used assembled Illumina reads to correct raw Nanopore reads, which were assembled using Flye Assembler[54]. The preliminary assembly included 119 contigs in 107 scaffolds (Scaffold N50 was 1,489,504 bp). We derived synteny blocks between the provisional assembly and our chromosome-level NIC203 assembly using Sibelia[55] and used the synteny blocks to scaffold the contigs to chromosome level using Ragout[56].

### Identification of *C. tropicalis* Argonaute proteins and piRNA pathway effectors

We annotated functional domains in *C. tropicalis* NIC203 using Interproscan 5 as part of our previous NIC203 genome assembly[9]. We identified Argonaute proteins with PFAM domains, including Piwi (PF02171), PAZ (PF02170), N-terminal domain of Argonaute (PF16486), Argonaute linker 1 (PF08699), Mid domain of Argonaute (PF16487) and Argonaute linker 2 (PF16488) domains. We excluded a protein with low molecular weight (41 kDa) as unlikely to be an Argonaute and the orthologue of *C. elegans* Dicer that represented an outgroup to the rest of the proteins. After aligning those sequences to *C. elegans* Argonautes identified in a previous study[57] using Clustal Omega we conducted phylogenetic analysis using iqtree2 (ref. 58), with 1,000 replicates of the approximate likelihood-ratio test (--alrt 1000) and 1,000 boostraps (-b 1000). iqtree2 carries out an initial model selection step, and a substitution model with the general Q matrix, empirical codon frequencies, a proportion of invariable sites and a free rate heterogeneity (Q.pfam+F + I + R4) was selected. Additional orthologues of *C. elegans* piRNA effector genes were identified through reciprocal blastp searches, synteny conservation, and gene trees from Wormbase Parasite[59]. *C. elegans mut-16*, *rrf-1*, and *simr-1* have 1:1 orthologues in *C. tropicalis*. The evolutionary history of SET proteins is complex due to their propensity to gain and lose paralogues within *Caenorhabditis*. The gene annotated gene as *C. tropicalis set-25*, is the closest among six paralogues in its genome.

Thus, the absence of a phenotype in the mutant may be attributed to genetic redundancy. The gene annotated as *C. tropicalis set-32* is a close orthologue of two *C. elegans* genes: *set-21* and *set-32*. The SET domains of *C.tr*-SET-32 and *C.el*-SET-32 are ~48% identical at the protein level. Additionally, using Alphafold2 (ref. 60) we found that these two proteins have high structural similarity (root mean square deviation = 0.962) and using the predicted structure of *C.tr*-SET-32 as a query retrieved *C.el*-SET-32 as the top hit in *C. elegans* (Foldseek)[61].

## Transgenerational silencing of slow-1/grow-1

In the transgenerational inheritance experiments, EG6180 hermaphrodites were crossed to NIL (QX2345) males. $F_1$ individuals were genotyped after laying embryos to distinguish between self-progeny from cross-progeny. $F_2$ embryos from cross-progeny mothers were singled, allowed to lay eggs and genotyped. $F_3$ homozygous carriers for *slow-1/grow-1* propagated for multiple generations and mated to EG6180 males. The *slow-1/grow-1* TA activity was assessed by determining the proportion of delayed EG/EG non-carriers.

## Single molecule in situ hybridization

Stellaris FISH Probes targeting *slow-1, slow-2* and *pgl-1* were designed using the Stellaris RNA FISH Probe Designer (Biosearch Technologies). The probes were labelled with Quasar 570, CAL Fluor Red 610 or Quasar 670, respectively (Biosearch Technologies). The protocol was adapted from Raj et al.[62] and described in ref. 9. For imaging, an Axio Imager. Z2 (Carl Zeiss) widefield microscope with a Hamamatsu Orca Flash 4 camera and a 63×/1.4 plan-apochromat Oil DIC objective was used. Filters used were: DAPI excitation 406/15 nm, emission 457/50 nm and Quasar 570 excitation 545/30 nm, emission 610/75 nm. *z*-stack images with 40 slices (step size 0.2 μm) were acquired. Image analysis was performed with the FIJI plugin RS-FISH[63] with parameters set at Sigma 1.44, and threshold 0.0062.

## RNA extraction and RNA-seq

Total RNA was extracted from approximately 100 young adult hermaphrodites and $F_1$ progeny, with the later using recessive mutations to visually discriminate cross-progeny from self-progeny. Reciprocal crosses were set up between parental strains for maternal or paternal inheritance of *slow-1/grow-1* by mating INK531 hermaphrodites (*uncoordinated* worms in NIC203 background) to EG6180 males and QX2355 hermaphrodites (*dumpy* worms in EG6180 background) to NIC203 males and selecting phenotypically wild-type progeny for RNA extraction. Reciprocal crosses between NIL and EG6180 strains were performed analogously (INK255 hermaphrodites (*dumpy* worms in NIL background) to EG6180 males and QX2355 hermaphrodites (*dumpy* worms in EG6180 background) to QX2345 NIL males). Total RNA was extracted following a modified version of the protocol in[64] including multiple M9 washes, TRizol and chloroform incubation, phase-separation, isopropanol precipitation and resuspension in RNase-free water. Samples with RNA integrity number (RIN) > 8 were used for library preparation using the NEBNext Poly(A) kit and sequenced on NextSeq2000 P2 SR100 or NovaSeq S1 PE100 at the Vienna Biocenter NGS facility. To reduce reference bias, raw reads were aligned to a concatenated NIC203 + EG6180 genome/transcriptome assembly using STAR and bcbio-nextgen (https://github.com/bcbio/bcbio-nextgen). Transcript quantification and normalization were performed with tximport and Deseq2 (ref. 65). We used Deseq2 to fit a model for the normalized counts using the strain identity of the mother and sequencing batch (Nextseq vs NovaSeq libraries) as fixed effects and compared the model to a null model that included only batch using a likelihood-ratio test. Despite identifying an outlier in the *slow-1/grow-1* paternal inheritance samples (Fig. 1d), no obvious difference between the outlier and the other samples in terms of RNA quality and mRNA-seq quality control were identified. However, since each library was derived from an independent genetic cross,

we cannot discard a human error, and therefore decided that it would be best practice to keep the outlier in the final analysis.

## RT–qPCR

RNA was extracted from adult worms (50 males or 100 hermaphrodites per biological replicate) using TRIzol-chloroform extraction, followed by Dnase I digestion[66] and then RNA concentrations were measured using the Qubit High-Sensitivity RNA fluorescence kit (Thermo). cDNA was prepared with SuperScript III reverse transcriptase (Thermo) using random hexamers. Intron-spanning primers were validated with standard curves from QX2345 cDNA to ensure amplification efficiency and an $r^2$ value above 0.95. The following primers were used: FW-slow-1-mRNA: 5′-GAGCTACCGGAACTGGATAAAG-3′, RV-slow-1-mRNA: 5′-CAGAGTTCTCGGAAGTCTCCTC-3′, FW-slow-1-pre-mRNA: 5′-CGGACTGGATGAAACATTTAGC-3′, RV-slow-1-pre-mRNA: 5′-GAGCGGTGTTGACctgaatc-3′, FW-cdc-42: 5′-CGATTAAATG TGTCGTCGTAGG-3′, and RV-cdc-42:5′-ACCGATCGTAATCTTCTTGTCC-3′. All samples had at least 3 biological replicates. We used the $\Delta\Delta C_t$ method to calculate relative fold change and chose *cdc-42* as a housekeeping gene[67,68]. *Cdc-42* expression showed a low coefficient of variation in our RNA-seq datasets suggesting its validity as a housekeeping gene. All RT–qPCR reactions were prepared with the Luna Universal qPCR and RT–qPCR kit (NEB) and run with an annealing temperature of 58 °C. All biological replicates were run in technical quadruplicate and any reactions with abnormal amplification curves or melting temperatures were omitted before analysis (distinct from reactions for which we observed no amplification, which were not omitted). Representative samples from each condition were Sanger sequenced. We confirmed the absence of genomic DNA contamination in RNA samples by performing PCRs with gDNA-specific primers using the RNA as template and observed no amplification after 40 cycles. RT–qPCR indicated specific amplification of *slow-1* in both hermaphrodites and males. However, the higher $C_t$ values for males (34.27 versus 28.31 on average) and greater variability (s.d. of 1.55 versus 0.65 in the NIL) suggest much lower expression levels in males. This variability hinders a reliable estimate of abundance and assessment of the parent-of-origin effect in males.

## Small RNA library preparation and sequencing

We isolated sRNAs, using the TraPR protocol[69]. In brief, frozen worm pellets (2,000 worms per parental line) were supplemented with 350 μl lysis buffer, (20 mM HEPES-KOH, pH 7.9, 10% (v/v) glycerol, 1.5 mM MgCl$_2$, 0.2 mM EDTA, 1 mM DTT, 0.1% v/v Triton X-100). Samples were mechanically disintegrated and subjected to 4 freeze–thaw cycles in liquid nitrogen. The resulting lysates were cleared by centrifugation and the sRNA fraction was isolated using the TraPR Small RNA Isolation Kit (135.24, LEXOGEN). Isolated sRNA was treated with RppH (M0356S, BioLabs), to ensure 5′ monophosphate-independent capturing of small RNAs[70], following purification with Agencourt RNA Clean XP magnetic beads (BECKMAN COULTER). The sRNA was ligated to a 32-nt 3′ adapter with unique barcodes (sRBC, Supplementary Table 6, IDT) using truncated T4 RNA ligase 2 (M0373L, NEB). The resulting RNA was run on 12% SequaGel–UreaGel (National Diagnostics) and purified with ZR small-RNA PAGE Recovery Kit (R1070, ZYMO RESEARCH). The 37-nt-long 5′ adapter was ligated to the sRNAs using T4 RNA ligase (M0204S, NEB). The resulting RNA was cleaned up (R1015, ZYMO RESEARCH), reverse-transcribed, and PCR amplified. The cDNA fragments (160–190 nt) were extracted and gel purified (D4008, ZYMO RESEARCH). Small RNA Libraries were sequenced in triplicates on a NovaSeq S1 SR100 mode (Illumina) at the Vienna Biocenter NGS facility. All sequencing libraries generated for this project are listed in Supplementary Table 7.

## sRNA immunoprecipitation

To study piRNA binding preferences of PRG-1.1 and PRG-1.2, we performed sRNA immunoprecipitation of N-terminally Flag-tagged PRG-1.1

(INK775) and PRG-1.2 (INK735) followed by sRNA-seq. For each of the 3 biological replicates (50,000 worms each), 18 worm plates (9 cm) were bleached to synchronize the population. Young adults were collected, frozen at −70 °C, thawed and washed with RIP buffer (50 mM Hepes pH 7.2, 150 mM NaCl, 0.01% NP-40). For lysis, RIP buffer and Benzonase were added and sonicated in a Diagenode Bioruptor followed by cleaning via centrifugation. For immunoprecipitation, 200 µl of Anti-Flag M2 Magnetic Beads (Millipore) were used (4 °C, overnight). The bound proteins were eluted in 500 µl 0.1 M GlycinHCl pH 2.7 for 5 min at room temperature. And transferred into a vial with 50 µl 1 M Tris-HCl pH 8. The proteins were digested with Proteinase K (0.7 mg ml$^{-1}$), and denatured proteins were removed by centrifugation following proteinase K inactivation. Samples were stored at −70 °C until library preparation.

## Small RNA analysis

Sequencing adapters were trimmed from 5′ and 3′ ends using Cutadapt v1.18 (ref. 71). Extracted 21U and 22G reads aligned to the genome using hisat2 v2.1 (ref. 72). For 22 G, only reads mapped to the coding sequences were analysed; for 21U, reads mapped to coding sequences, tRNAs and rRNAs were excluded using seqkit v0.13 and samtools v1.10. 22 G reads were quantified using featureCounts (Rsubread, R), normalized by the total number of 22 G per replicate, and visualized using the Gviz R package[62]. Candidate 21U-RNAs were identified based on perfect mapping and abundance criteria (>0.1 ppm). A custom script quantified 21U-RNAs and reads were normalized to miRNAs predicted based on homology to *C. elegans* miRNAs. To identify potential 21U-RNAs *slow-1* candidates we used known targeting rules in *C. elegans* and binding energies. First, putative binding sites and energies for all 21U-RNAs against *slow-1* mRNA were predicted with RNAduplex (ViennaRNA Package v2.0.58)[63], of which five best duplexes for every piRNA were taken. Candidate piRNAs without bubbles during binding and no more than 4 mismatches outside the seed region were extracted and ranked by binding energy (Supplementary Data 1). The second candidate list was generated considering the overall level of binding continuity by using Nucleotide blast v2.2.26 in blastn-short mode. Only *21U-RNA*s with no mismatches or gaps in the seed region were selected for further analysis. Finally, we ranked *21U-RNA*s by the total length of the ungapped alignment to *slow-1* (Supplementary Data 1).

## Chromatin immunoprecipitation

For chromatin immunoprecipitation, we collected an $F_4$ population of homozygous carriers for the repressed *slow-1* allele after paternal inheritance, which was highly enriched in s22G-RNA complementary to *slow-1* (Fig. 3i,j). First, we crossed EG6180 hermaphrodites to NIL males. The $F_2$ were genotyped to identify repressed *slow-1/grow-1* (NIC/NIC) worms which were expanded for two generations ($F_4$) and collected as young adults. Each ChIP sample represents an independent genetic cross. Worms (200 µl) were collected, washed and incubated to minimize bacterial content and frozen in liquid nitrogen. For ChIP, we used the protocol described[64]. Shortly the frozen worm pellet was pulverized by grinding in mortar with liquid nitrogen and the powder was crosslinked in 1 ml ice-cold RIPA buffer supplemented with 2% formaldehyde to crosslink (10 min, 4 °C). After quenching by addition of 100 µl 1 M Tris-HCl (pH 7.5), the sample was sonicated using Covaris for 600 s to achieve chromatin fragments of 200–500 bp. Fifty microlitres of the lysate was saved as an input fraction. Chromatin was immunoprecipitated using anti-H3K9me3 antibody (Ab8898, Abcam). The immunoprecipitation product was incubated with Protein A Dynabeads (Thermofisher scientific) and washed with LiCl. The immunoprecipitation product was eluted from beads and DNA was purified using ChIP DNA Clean and Concentrator kit (Zymo Research). Input control fractions were treated similarly to immunoprecipitation samples. DNA libraries were prepared with NEBNext Ultra II DNA Library Prep Kit (Illumina), deduplicated using bbmap v38.26, aligned using bwa mem v0.7.17 (ref. 65), and normalized by the number of reads that mapped

to the genome with samtools v1.10 (ref. 73). Peaks were called by macs2 v2.2.5 with −broad and −mfold 1 50 options[74]. Quality control plots were made using deeptools v3.3.1 (ref. 75). H3K9me3 signal was calculated as read counts per genomic position in the ChIP sample normalized by counts in the corresponding input sample using bedtools v2.27 (ref. 76) and custom R (v4.3) script.

## Immunohistochemistry

Gravid nematodes were washed from plates, and embryos were extracted using bleach solution. The embryo suspension was applied to prepared poly-L-lysine slides (Sigma-Aldrich, P8920), and immersed into liquid nitrogen, fixed in ice-cold methanol (10 min) followed by acetone (10 min), and rehydrated in descending ethanol concentrations (95%, 70%, 50% and 30% ethanol). Fixed embryos were blocked in 3% BSA (VWR Life Science, 422351 S), followed by incubation with anti-Flag M2 primary antibody (Sigma-Aldrich, F3165, diluted 1:3,000). After washing, a secondary antibody Alexa Fluor A568 (ThermoFisher Scientific, A-11031, diluted 1:3,000) was applied, followed by additional washes. The final wash contained DAPI (Merck, D9542, 5 ng ml$^{-1}$). Processed embryos were mounted with Fluoroshield (Sigma-Aldrich, F6182) and imaged at Axio Imager 2 (ZEISS).

## Fluorescence intensity quantification

Twenty-four-bit raw images were analysed in Fiji (v1.53r)[77]. Embryos were selected by freehand tool and the same selection mask was used to capture background fluorescence intensity for each embryo. To compare fluorescence intensities between strains we used corrected total cell fluorescence (CTCF) parameter (CTCF = integrated density − (area of selected cell × mean fluorescence of background readings)). At least 23 embryos were used for quantification.

## Worm protein lysate preparation and western blot

Gravid adult worms were collected, washed, and flash-frozen in the liquid nitrogen. Worm pellets were resuspended in ice-cold lysis buffer (30 mM HEPES pH 7.4, 100 mM KCl, 2 mM MgCl2, 0.05% IGEPAL, 10% glycerol and 1 tablet of protease inhibitors (Roche, 11836153001)) and lysed by sonication in Bioruptor (UCD-200, Diagenode) followed by centrifugation to obtain the supernatant. After protein quantification by Bradford assay (Thermo Scientific, 23238), samples were diluted, resuspended in SDS loading buffer, and loaded onto NuPAGE gels (Invitrogen). Samples were transferred to 0.45 µm PVDF membrane (Thermo Scientific, 88518) and blocked with 4% non-fat milk in TBS-T. Membranes were incubated with anti-Flag M2 (mouse, 1:2,000, Sigma-Aldrich, F3165) or anti-actin (rabbit, 1:3,000, Abcam, ab13772) primary antibody overnight followed by incubation with HRP-conjugated anti-mouse (1:10,000, Invitrogen, G-21040) or anti-rabbit (1:10,000, Jackson Immuno, 111-035-045) secondary antibody. Detection was performed using ECL reagent (Cytiva, RPN2106) and imaged with ChemiDoc MP (Bio-Rad). Membranes were stripped before reprobing (Thermo Scientific, 21059).

## Live imaging of mScarlet::SLOW-1

Approximately 20 gravid adults were dissected in M9 medium under a stereo microscope. Embryos were transferred to individual wells in a Thermo Scientific Nunc MicroWell 384-Well Optical-Bottom Plate (Thermo Scientific). Embryos were imaged using an Olympus spinning disk confocal based on an Olympus IX3 Series (IX83) inverted microscope, equipped with a dual-camera Yokogawa W1 spinning disk (Yokogawa Electric Corporation) and two ORCA-Flash 4.0 V3 Digital CMOS cameras (Hamamatsu). Each field was imaged using a 40×/0.75 NA (air) objective, 16 z-sections at 2 µm and conditions were as follows: bright-field (100% power 30 ms) 568 nm, (100% power, 500 ms). Image acquisition was performed using CellSense software (Olympus). Image processing and montages were created using Fiji and embryoCropUI[78].

## Reporting summary

Further information on research design is available in the Nature Portfolio Reporting Summary linked to this article.

## Data availability

All sequencing data generated in this study are available under NCBI Project accession PRJNA850171.

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

**Acknowledgements** The authors thank members of the Burga laboratory and J. Mueller for critical reading of the manuscript. Research in the Burga laboratory is supported by the Austrian Academy of Sciences, the city of Vienna, and a European Research Council (ERC) Starting Grant under the European 20 Union's Horizon 2020 programme (ERC-2019-StG-851470). A.K. is supported by a Boehringer Ingelheim Fonds (BIF) PhD Fellowship. S.A.W. is supported by the European Union's Framework Programme for Research and Innovation Horizon 2020 (2014-2020) under the Marie Curie Skłodowska Grant Agreement Nr. 847548. Work by the Ben-David laboratory for this study was supported by the Israel Science Foundation (grant no. 2023/20). We thank Life Science Editors for editing services.

**Author contributions** P.P. conducted the genetic analyses for this study under the supervision of E.B.-D. and A.B. sRNA-seq, ChIP–seq and other molecular assays were led by H.M. with the support of P.P., D.H., A.K., P.T., D.K., A.H., Y.K. and S.A.W. under the supervision of J.B. and A.B. Genome assemblies and all bioinformatic analyses were carried out by A.K. and supported by E.B.-D. Design and generation of worm transgenic lines was performed by J.G. and P.D. A.B. conceptualized the study, designed experiments together with P.P., and wrote the manuscript.

**Competing interests** The authors declare no competing interests.

### Additional information

**Correspondence and requests for materials** should be addressed to Alejandro Burga.

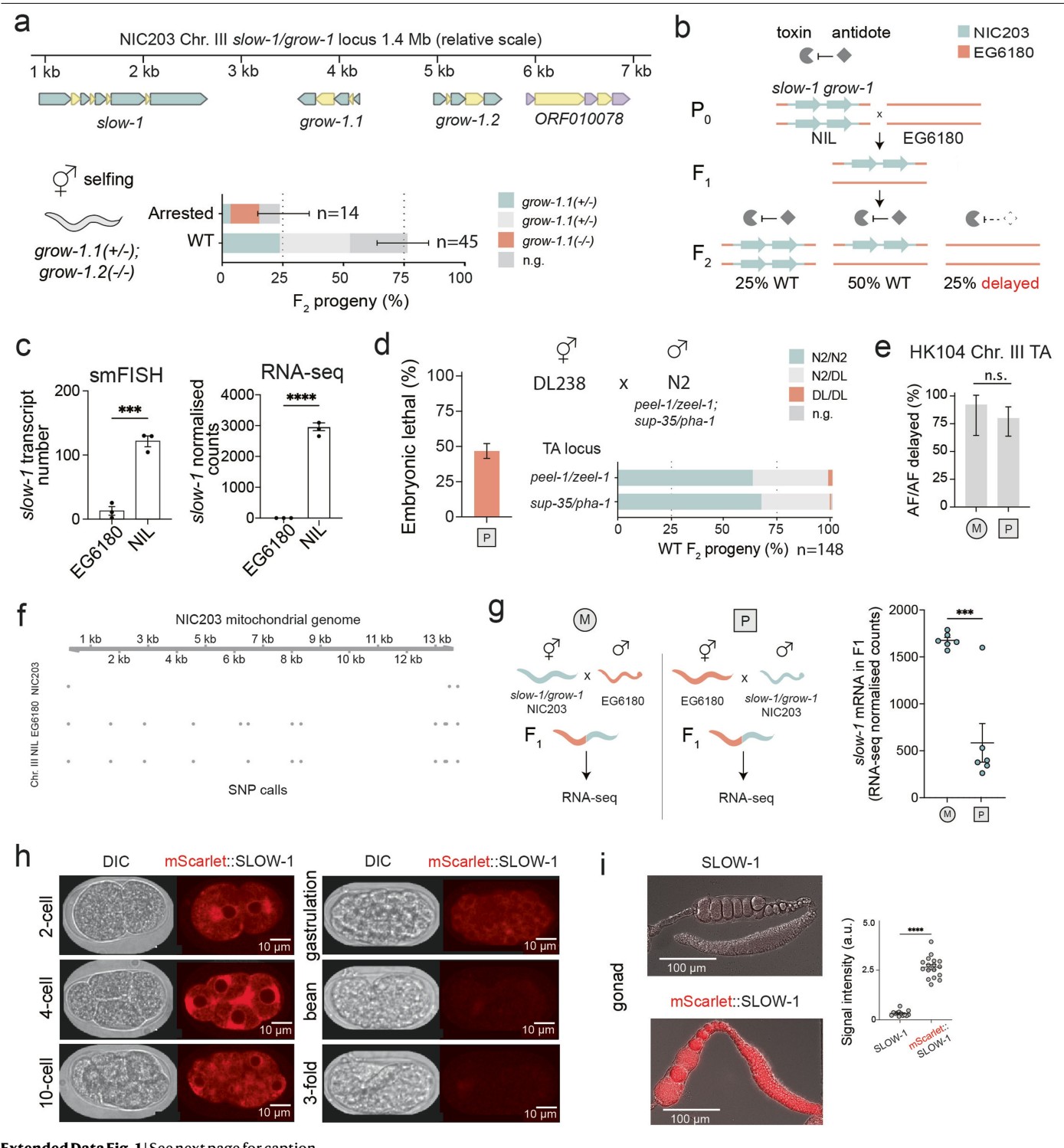

**Extended Data Fig. 1** | See next page for caption.

**Extended Data Fig. 1 | *slow-1* has a parent-of-origin effect and *grow-1.1* and *grow-1.2* are redundant antidotes. a**, Corrected NIC203 genome assembly showing segmental duplication of the *grow-1* antidote (top). Selfing of *grow-1.1(+/−); grow-1.2(−/−)* strain. All g*row-1.1(−/−); grow-1.2(−/−)* individuals were developmentally arrested during larval development and did not produce any viable offspring. Thus, *grow-1.1* and *grow-1.2* are genetically redundant (bottom). Data are presented as mean values +/−95% CI. **b**, Model illustrating the mechanism of action of the *slow-1/grow-1* TA. In crosses between the carrier strain (*slow-1/grow-1* Chr. III NIL) and non-carrier strain (EG6180), 25% of the $F_2$ progeny is developmentally delayed because EG/EG homozygotes did not inherit the TA and cannot express the antidote to counteract the maternally deposited toxin. **c**, Quantification of *slow-1* mRNA expression by smFISH (N = 3, n = 16–30 per repeat, two-sided unpaired t-test, p = 0.0006, mean +/−SEM) and RNA-seq in both NIL and EG6180 parental strains (two-sided unpaired t-test, p < 0.0001, mean +/−SEM). **d**, Previously, we showed that *sup-35/pha-1* is active when maternally inherited (Ben-David, et al. *Science* 2017). To test whether *sup-35/pha-1* is active when paternally inherited, we crossed DL238 hermaphrodites and N2 males. N2 carries two TAs, *peel-1/zeel-1* and *sup-35/pha-1*. We observed 46.7% embryonic lethality among their $F_2$ progeny (n = 340, mean +/−95% CI is shown), as expected from the activity of two TAs segregating independently. To confirm the activity of both TAs, we genotyped wild-type $F_2$ progeny for both *peel-1/zeel-1* (Chr. I) and *sup-35/pha-1* (Chr. III) and found that the vast majority of WT progeny were either homozygous or heterozygous carriers, indicating that both *peel-1/zeel-1* and *sup-35/pha-1* non-carrier individuals died as embryos. **e**, Activity of the *C. briggsae* HK104 Chr. III *msft-1* TA in reciprocal crosses. Penetrance of the toxin, the percentage of $F_2$ non-carrier individuals that are phenotypically affected, is used as a proxy for TA activity (HK Chr. III TA: $n_M = 13$, $n_P = 35$, p = 0.42; two-sided Fisher's exact test; mean +/−95% CI is shown). **f**, Illumina short-reads from NIC203, EG6180, and Chr. III NIL DNA libraries were aligned against the NIC203 mitochondrial genome. Each dot represents a SNP. As expected from our cross scheme, the Chr. III NIL has the EG6180 mitochondrial genotype. Those SNPs shared by all strains likely reflect an error in the original NIC203 assembly. **g**, The *slow-1* parent-of-origin effect is also present in a reciprocal cross between NIC203 and EG6180 parental lines. The abundance of *slow-1* transcripts is higher when the *slow-1* locus is maternally inherited (two-way ANOVA, interaction p = 0.0005, Holm-Sidak post hoc test, $p_{slow-1} < 0.0001$, mean +/−SEM is shown). **h**, Expression pattern of mScarlet::SLOW-1 during embryonic development. In early embryos, SLOW-1 appeared to be associated with the nuclear envelope and was quickly degraded during embryogenesis. SLOW-1 was not detectable in the soma by the time embryos reached the comma-stage (N = 2, number of embryos imaged = 20). **i**, Expression pattern of mScarlet::SLOW-1 in the hermaphroditic gonad. Quantification of signal intensity in gonads of NIL and mScarlet::SLOW-1 strains (two-sided unpaired t-test, p < 0.0001, mean +/−SEM is shown).

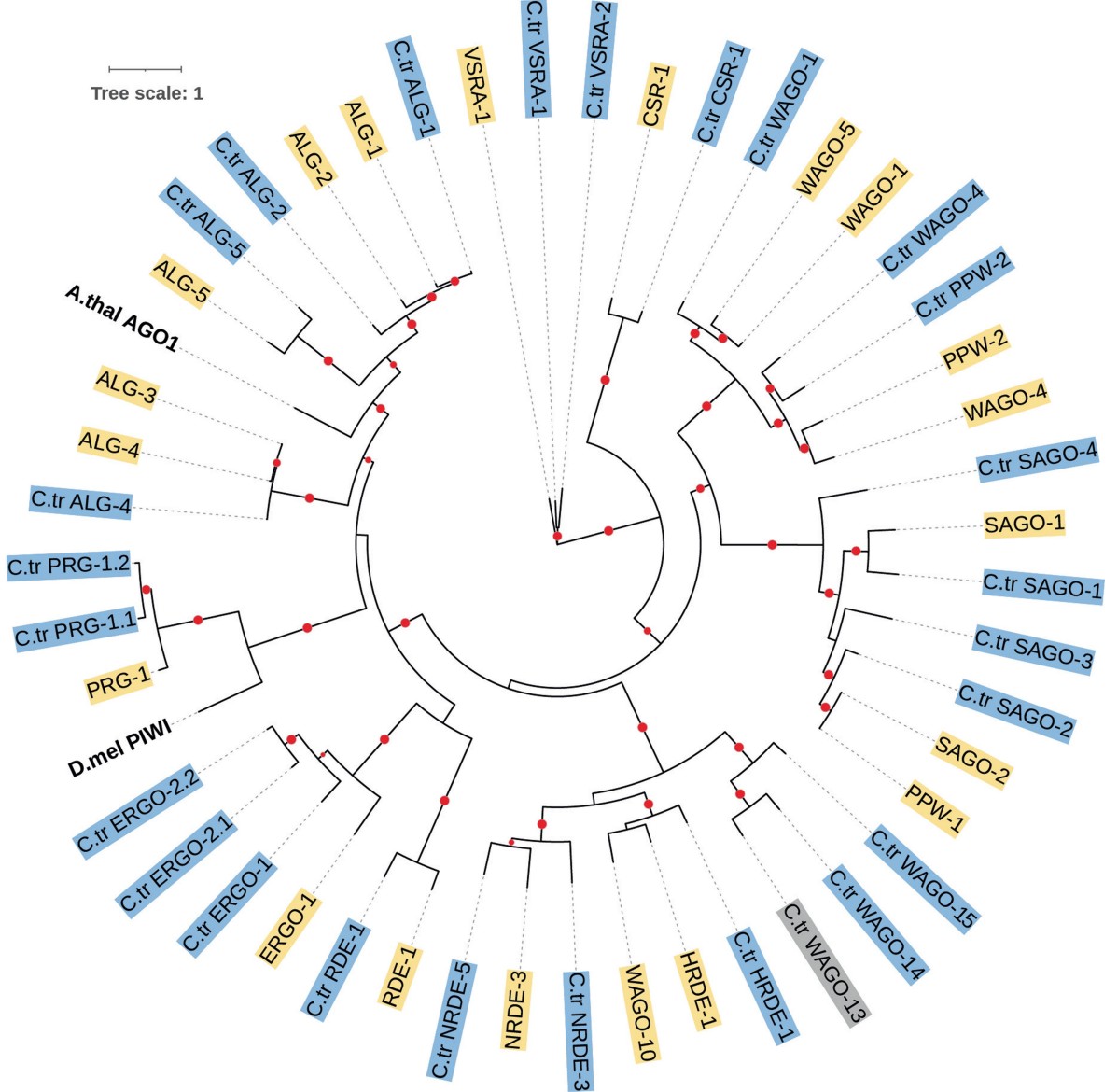

**Extended Data Fig. 2 | Phylogenetic tree of *C. tropicalis* and *C. elegans* Argonaute proteins.** *C. elegans* (yellow) and *C. tropicalis* (blue) Argonaute proteins. Putative pseudogene in gray. Also included *A. thaliana* AGO1 and *D. melanogaster* PIWI. Red circles denote bootstrap values > 95%.

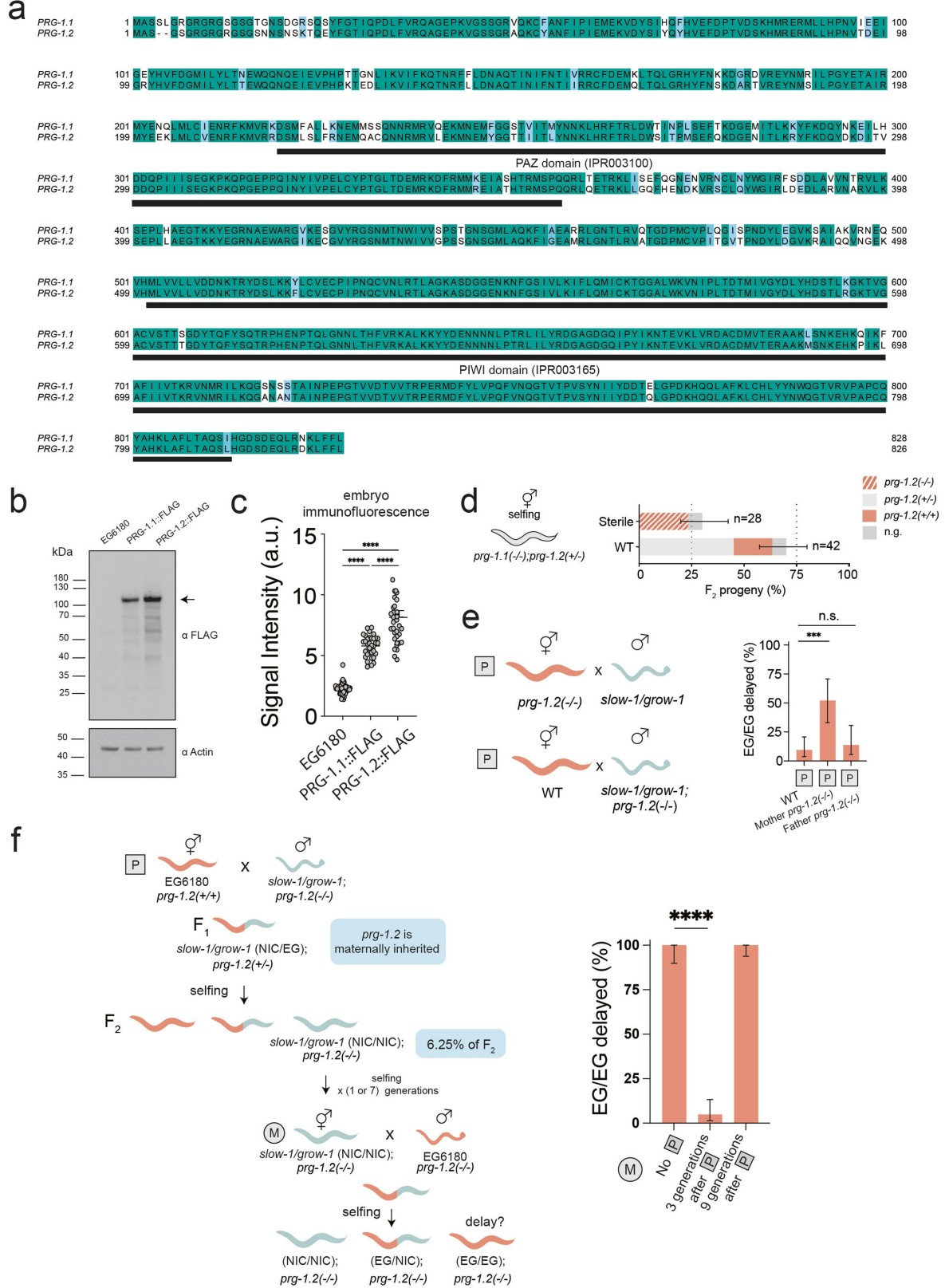

**Extended Data Fig. 3** | See next page for caption.

**Extended Data Fig. 3 | *C. tropicalis* PRG-1.1 and PRG-1.2 are redundant paralogs and PRG-1.2 acts maternally. a**, Protein alignment of PRG-1.1 and PRG-1.2. The two proteins share 87% amino-acid pairwise identity with 722/828 identical sites. Conserved PAZ and PIWI domains are highlighted. **b**, Detection of endogenously tagged PRG-1.1::FLAG and PRG-1.2::FLAG by western blot. Black arrow indicates the expected MW. EG6180 was used as a negative control. Western blot against β-Actin serves as a sample processing control. Uncropped gels available in Supplementary Fig. 1. **c**, Quantification of FLAG immunofluorescence quantification of *C. tropicalis* PRG-1.1 and PRG-1.2 expression from embryos, ($n_{EG6180} = 53$, $n_{prg-1.1} = 34$, $n_{prg-1.2} = 35$, one-way ANOVA, $p < 0.0001$, Tukey post hoc test, $p_{EG vs prg-1.1} < 0.0001$, $p_{EG vs prg-1.2} < 0.0001$, $p_{prg-1.1 vs prg-1.2} < 0.0001$, mean +/−SEM is shown). **d**, Selfing of *prg-1.1(–)*; *prg-1.2(+/−)* strain. All prg-*1.1(–)*; *prg-1.2(–)* individuals were sterile, therefore the line couldn't be propagated (mean +/−95% CI is shown). **e**, Cross of *prg-1.2(–)* hermaphrodites to NIL males (top) and NIL hermaphrodites to *prg-1.2(–)* males (bottom) indicates that paternal *slow-1* is repressed only when *prg-1.2* is maternally inherited. Percentage of F$_2$ EG/EG delayed progeny (right) when *prg-1.2* is absent from the mother or the father compared to the WT cross. When *prg-1.2* null mutant mothers were crossed to wild-type NIL males, we observed that 52.1% of F$_2$ homozygous EG/EG individuals were delayed. In contrast, when EG6180 mothers were crossed to *prg-1.2* null mutant NIL males, F$_2$ homozygous EG/EG individuals were mostly wild-type (13.8% delay), indicating that maternal *prg-1.2* is necessary for *slow-1* repression ($n_p = 53$, $n_{mother prg-1.2(–)} = 23$, $n_{father prg-1.2(–)} = 29$, two-sided Fisher's exact test compared to WT, $p_{mother prg-1.2(–)} = 0.0001$, $p_{father prg-1.2(–)} = 0.71$, mean +/−95% CI is shown). **f**, PRG-1.2 is not necessary for the maintenance of *slow-1* repression. In this crossing scheme mothers provide PRG-1.2 to their F$_1$ progeny, which is sufficient for piRNA-mediated repression, and the *slow-1/grow-1* is paternally inherited. Since EG6180 hermaphrodites do not provide maternal *slow-1* transcripts, then the *slow-1* paternal allele is epigenetically repressed in the F$_1$. Then F$_2$ progeny are singled, and their offspring genotyped for both the TA and the *prg-1.2* locus. Hermaphrodites that are homozygous carriers for both the TA and the *prg-1.2(–/–)* null allele are identified (6.25% the F$_2$ progeny) and propagated for 1 or 7 generations. Then these hermaphrodites are crossed to EG6180 *prg-1.2 (–/–)* males to test whether the TA is active. Inset shows the observed activity of the TA measured as percentage of delayed (EG/EG) individuals ($n_M = 34$, $n_{prg-1.2\_3\_gen} = 62$, two-sided Fisher's exact test, $p < 0.0001$, mean +/−95% CI is shown).

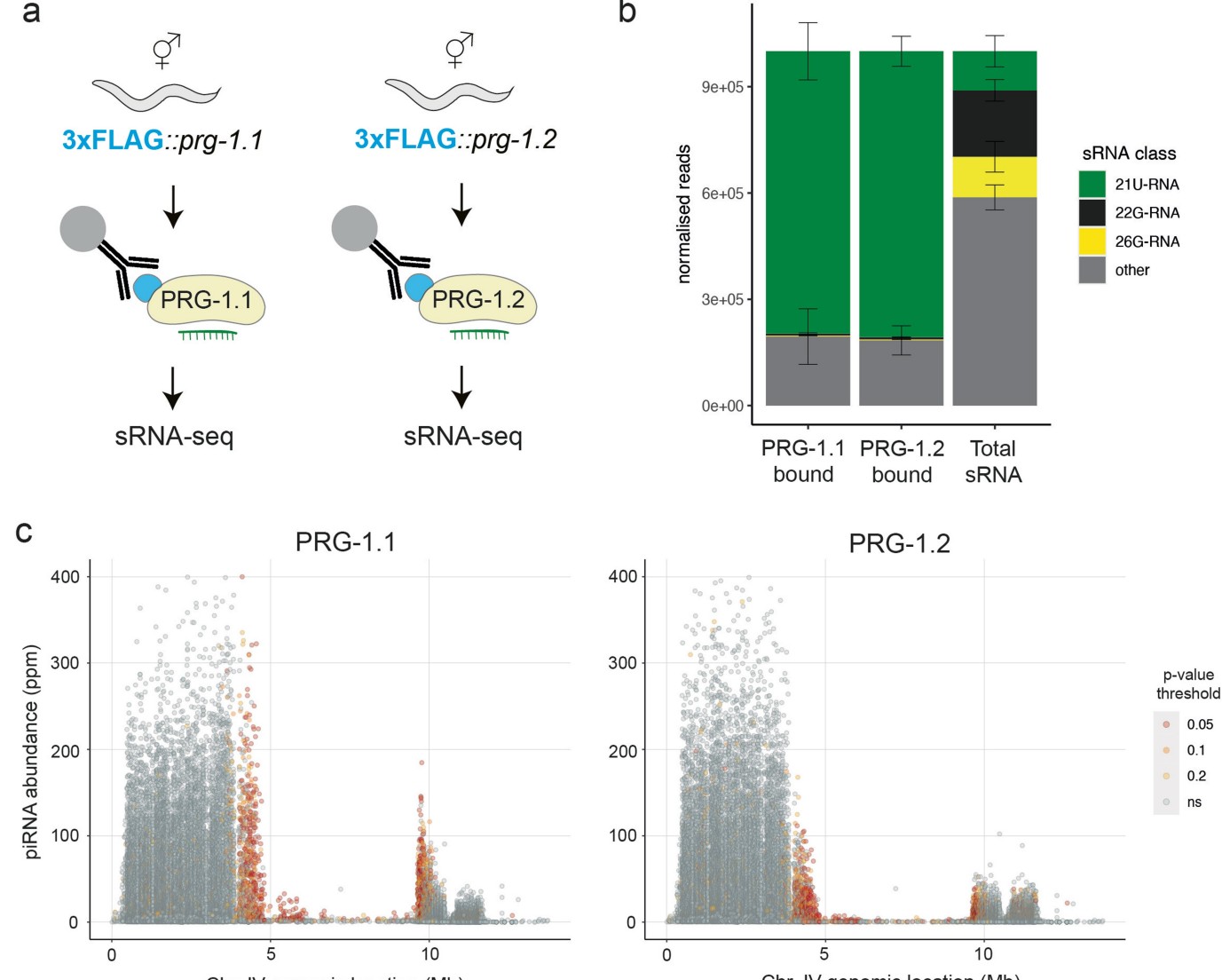

**Extended Data Fig. 4 | Characterization of the redundant role of PRG-1.1 and PRG-1.2 in *C. tropicalis*. a**, Schematic of the sRNA immunoprecipitation (RIP) protocol. RIP was performed in biological triplicates on N-terminally FLAG-tagged PRG-1.1 and PRG-1.2 strains (tagging the endogenous locus) followed by sRNA-seq. **b**, Class enrichment among sRNAs bound either to PRG1-1.1 and PRG-1.2 compared to a total sRNA-seq protocol ($n_{PRG-1.1} = 3$, $n_{PRG-1.2} = 3$). **c**, Differences in the abundance of piRNAs bounds by each PRG-1 paralog.

Normalized abundance (transcript per million) of PRG-1.1 (left) and PRG-1.2 (right) bound piRNAs (21U-RNAs) are plotted on the Y-axis and genomic coordinate of the respective piRNA loci on the X-axis. Differentially bound piRNAs are shown with different thresholds of significance and are enriched within specific genomic clusters (moderated t-test, p-value adjusted with Benjamini & Hochberg correction).

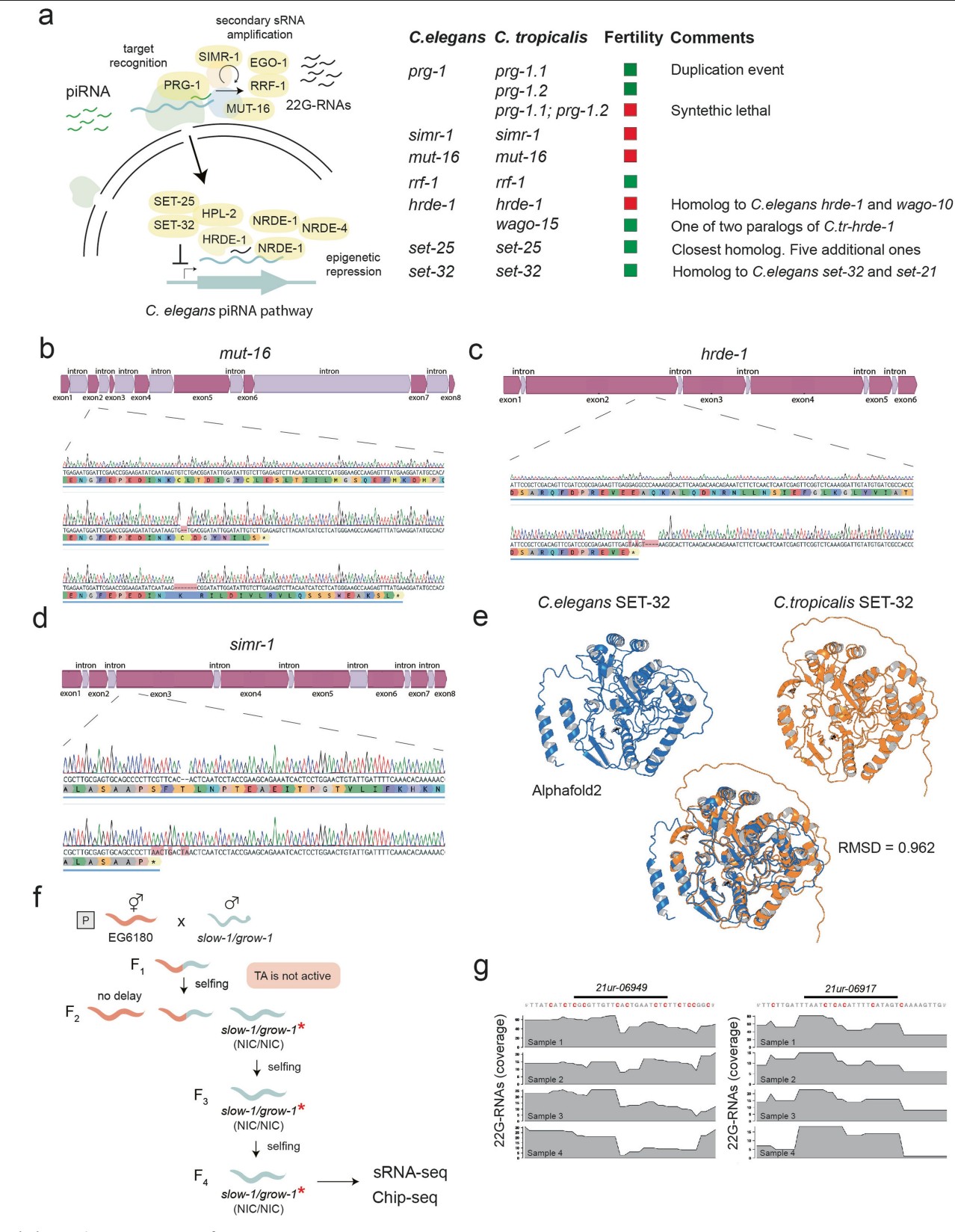

**Extended Data Fig. 5** | See next page for caption.

**Extended Data Fig. 5 | Characterization of mutants in the in *C. tropicalis* piRNA pathway and the repressive signal. a**, Schematic of the piRNA pathway in *C. elegans* (left). *C. tropicalis* mutants generated in this study, their fertility status, and evolutionary relationship to *C. elegans* proteins (right). **b**, *C. tropicalis mut-16* mutant alleles generated with CRISRP/Cas9. Homozygous mutants were sterile. **c**, *C. tropicalis hrde-1* mutant allele generated with CRISRP/Cas9. Homozygous mutants were sterile. **d**, *C. tropicalis simr-1* mutant allele generated with CRISRP/Cas9. Homozygous mutants were sterile. **e**, Alphafold2 predictions of *C. elegans* SET-32 and *C. tropicalis* SET-32 proteins and their structural alignment (PYMOL). Both proteins are highly similar (RMSD = 0.962).

**f**, Mating scheme to generate an $F_4$ homozygous population for the repressed *slow-1/grow-1* TA following its paternal inheritance. This $F_4$ population was subjected to sRNA-seq in biological quadruplicates (each time starting from an independent initial parental cross). As a control for an active or licensed TA, we performed sRNA-seq in the NIL parental line in biological quadruplicates. Red asterisk denotes *slow-1* repressed state. **g**, Zoom-in into the *21ur-06949* and *21ur-06917* predicted binding sites in *slow-1* and the 22G-RNAs mapping to these regions. The 22G-RNAs are derived from the $F_4$ "*slow-1* repressed" population. Each track is one of four biological quadruplicates. A modest but clear peak is observed within the predicted binding region.

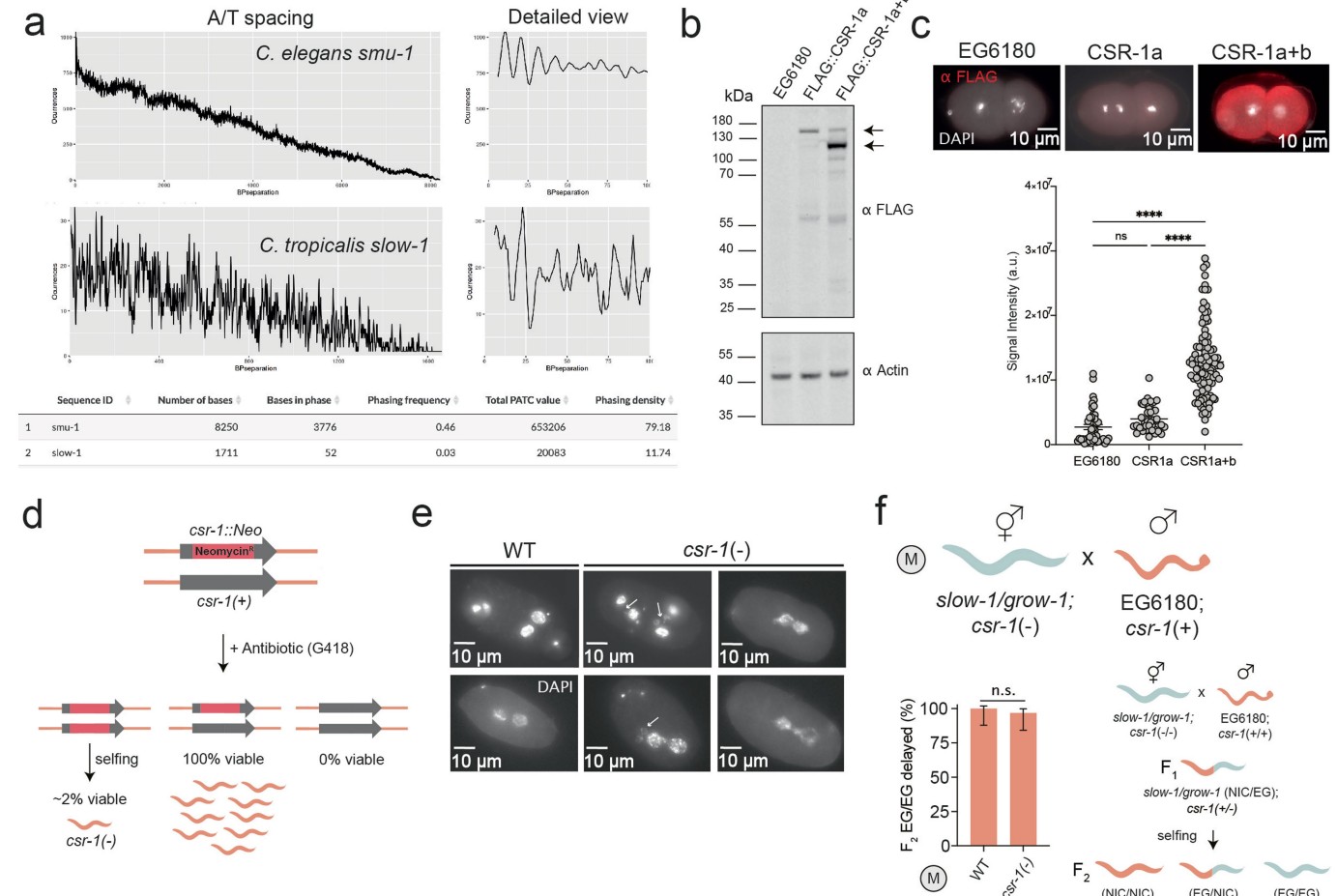

**Extended Data Fig. 6 | Examining the role of PATCs and CSR-1 in *slow-1* licensing. a**, *slow-1* lacks PATCs in its intronic sequences. PATC periodicity analysis for positive control *C. elegans smu-1* and *C. tropicalis slow-1*. Highest periodicity found at 10.5 bp and 20 bp respectively. Summary table of PATC analysis. Phasing threshold was set to 60 (-1% Phasing in random DNA). Analysis was run on https://wormbuilder.org/patc/. **b**, Detection of endogenously tagged FLAG::CSR-1a and FLAG::CSR-1a+b by western blot. Black arrows indicate the expected MW of each isoform. Western blot against β-Actin serves as a sample processing control. Uncropped gels available in Supplementary Fig. 1. **c**, Representative immunostaining images against FLAG::CSR-1a and FLAG::CSR-1a+b line in 2-cell stage embryos. EG6180 serves as a negative control (top). FLAG immunofluorescence quantification of *C. tropicalis* CSR-1

expression from 2-cell stage embryos of EG6180 (negative control), FLAG::CSR-1a, and FLAG::CSR-1a+b (2 repeats, $n_{EG6180} = 44$, $n_{CSR1a} = 40$, $n_{CSR1a+b} = 105$, one-way ANOVA, p < 0.0001, Tukey post hoc test, $p_{EG\,vs\,CSR1a} = 0.41$, $p_{EG\,vs\,CSR1a+b} < 0.0001$, $p_{CSR1a\,vs\,CSR1a+b} < 0.0001$, mean +/−SEM is shown). **d**, Generation of balanced *csr-1* null strain by inserting *NeoR* in *csr-1* and growing worms in G418 antibiotic. **e**, Representative images of wild-type (left), and *csr-1(−)* (right) embryos. Anaphase bridging events were observed in 14.75% of *csr-1(−)* early embryos (n = 61) compared to 0% (n = 36) of EG6180 WT. **f**, Crosses between *csr-1(−)*; *slow-1/grow-1* hermaphrodites and EG6180 males indicate that slow-1/grow-1 is fully active when maternally inherited. *csr-1(−)* hermaphrodites were derived from selfing of *csr-1(+/−)* mothers. ($n_M = 34$, $n_{csr-1(−/−)} = 32$, two-sided Fisher's exact test, p = 0.4848, mean +/−95% CI is shown).

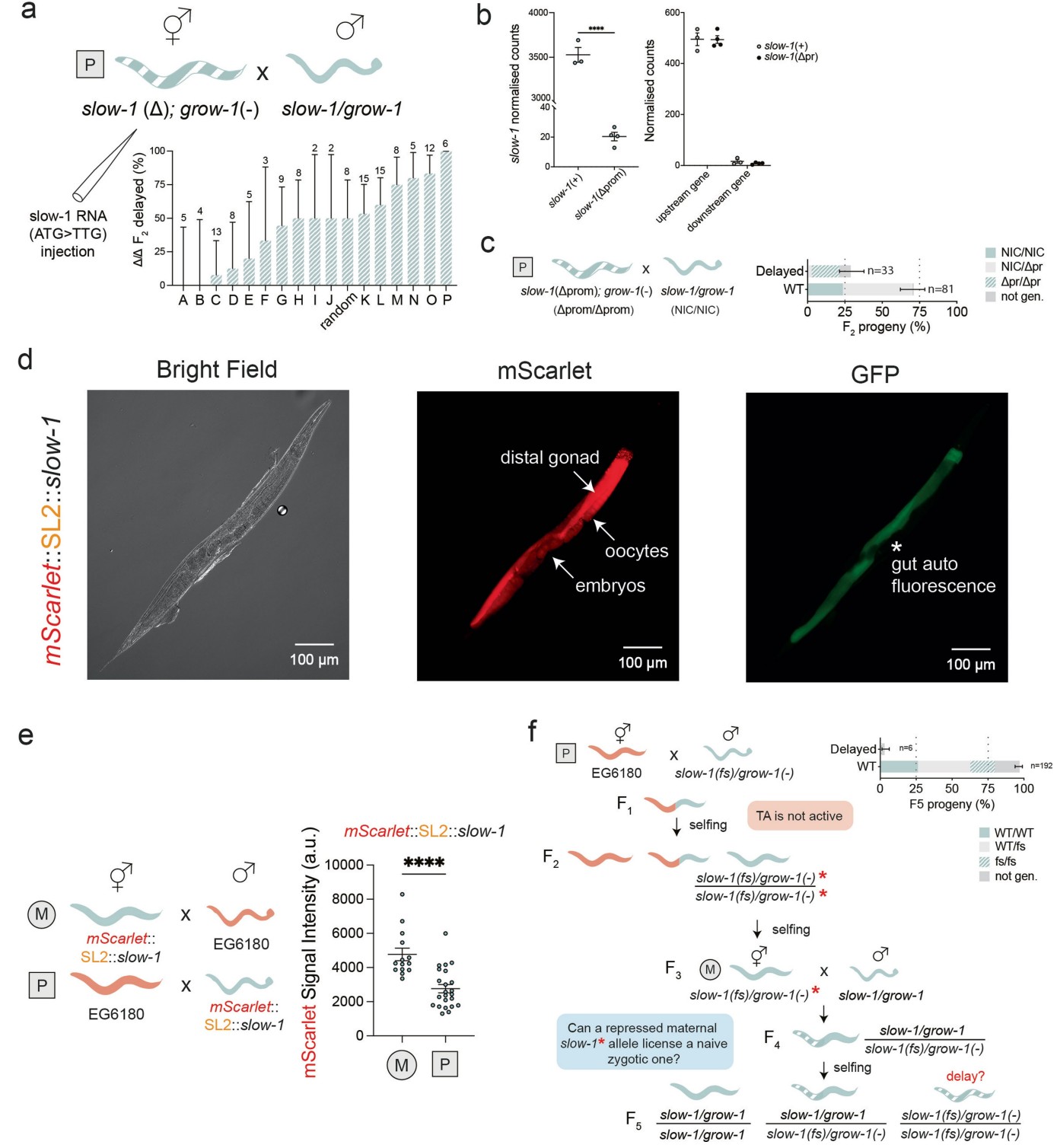

**Extended Data Fig. 7** | See next page for caption.

**Extended Data Fig. 7 | Characterization of the licensing signal. a**, Variability in the rescue of *slow-1/grow-1* activity across injected hermaphrodites following paternal inheritance. Sixteen *slow-1(Δ)/grow-1(−)* NIL hermaphrodites (A to P) were injected in both gonad arms with in vitro transcribed *slow-1* RNA (mutated start codon) and later mated to *slow-1/grow-1* NIL males. The total number of homozygous *slow-1(Δ)/grow-1(−)* $F_2$ progeny from each hermaphrodite is shown on top of each bar. The sample labeled as "random" represents embryos randomly picked from different injected mothers (mean +/−95% CI is shown). **b**, Quantification of *slow-1* transcripts following deletion of a 620 bp region upstream of *slow-1*. Comparison between *slow-1(+)/ grow-1.1(+)/grow-1.2(−)* NIL and *slow-1(Δpr)/grow-1.1(+)/grow-1.2(−)* NIL strains by RNA-seq. Deletion of the slow-1 promoter causes a 176-fold decrease in *slow-1* transcript levels (two-sided unpaired t-test, p < 0.0001, mean +/−SEM is shown). Quantification of the genes immediately upstream of *slow-1* and downstream of *grow-1*. Deletion of the *slow-1* promoter does not change the transcript level of the two neighboring genes (two-way ANOVA, interaction p = 0.81, mean +/−SEM is shown). **c**, Cross between *slow-1(Δpr)/grow-1(−)* NIL hermaphrodites and NIL males. All homozygous *slow-1(Δpr)/grow-1(−)* NIL $F_2$ progeny are delayed (100%, n = 21, mean +/−95% CI is shown) (bottom). **d**, Expression pattern of the *mScarlet*::SL2::*slow-1* operon in a gravid hermaphrodite. *mScarlet* fluorescence is ubiquitous in the germline (including oocytes) an also present in early embryos, as

expected from being driven by the endogenous *slow-1* promoter. GFP channel shows the autofluorescence of the gut as a reference (n = 35). **e**, Quantification of *mScarlet* fluorescence in the $F_1$ progeny of reciprocal crosses between the *mScarlet*::SL2::*slow-1* operon and EG6180 strains. Each data point represents one individual. Expression levels of *mScarlet* are lower when the operon is paternally inherited likely through piRNA-mediated co-repression due to lack of *slow-1* licensing ($n_m$ = 14, $n_p$ = 23, two-sided unpaired t-test, p < 0.0001, mean +/−SEM is shown). **f**, Schematic of a cross designed to test whether an epigenetically repressed *slow-1* allele can license a naïve zygotic one. To this end, we took advantage of the *slow-1(fs)/grow-1(−)* double mutant line that carries a *slow-1* frameshift mutation. This mutation renders the *slow-1* toxin inactive, but it doesn't affect the ability of its maternal mRNA to license a paternally inherited *slow-1* copy (Fig. 4a). We first crossed EG6180 hermaphrodites to *slow-1*(fs) *grow-1*(−) males and collected $F_2$ homozygous carriers of the repressed *slow-1*(fs) allele. We let these hermaphrodites self-fertilize for one generation, collected $F_3$ hermaphrodites and crossed them to NIL males carrying the wild-type TA. Finally, we phenotyped and genotyped their granddaughters ($F_5$ generation). If the wild-type TA were active, i.e. licensed, then progeny homozygous for the *slow-1(fs)/grow-1(−)* allele should be developmentally delayed. We found no significant delay among the progeny, and all homozygous mutants were wild-type indicating that licensing had been compromised.

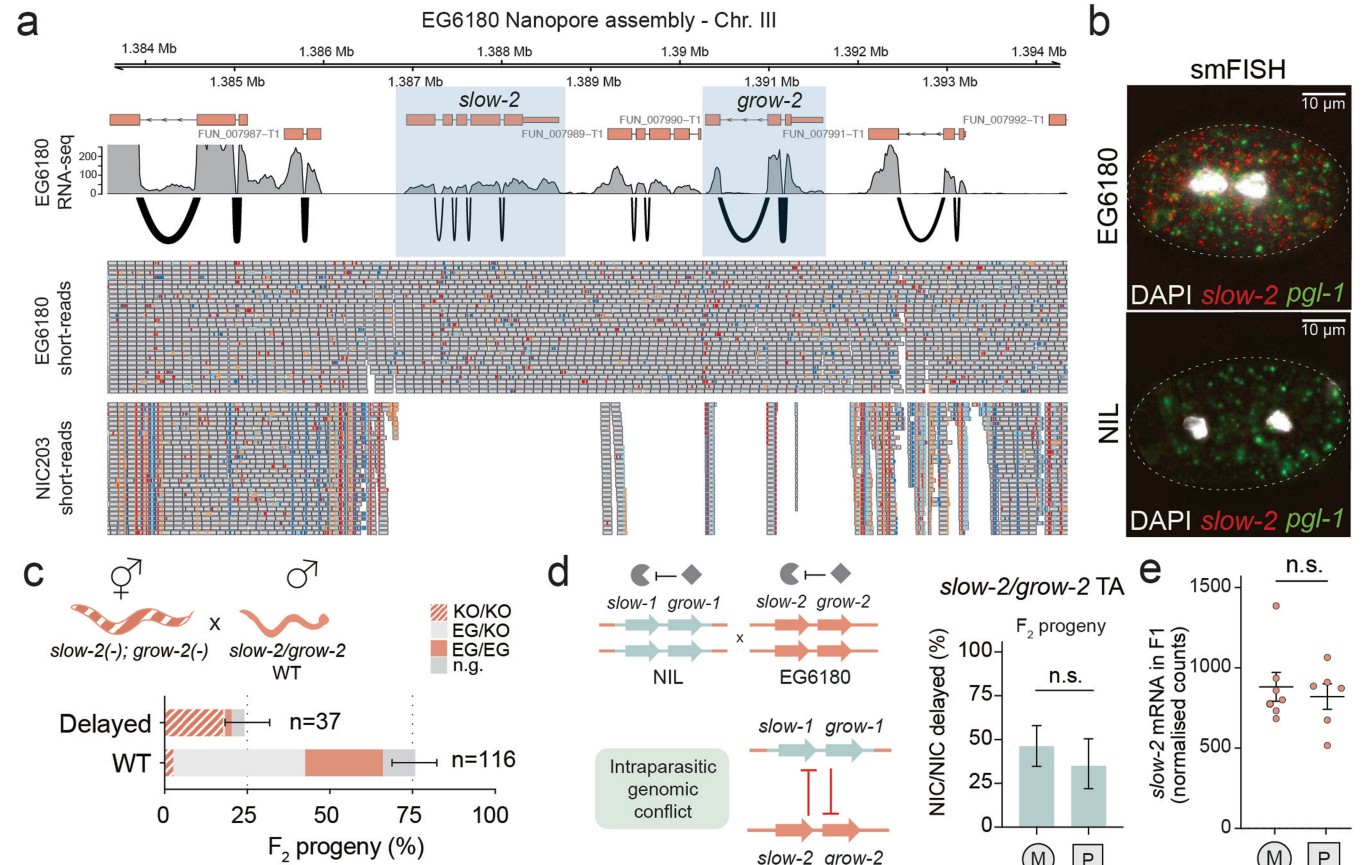

**Extended Data Fig. 8 | A related but divergent TA, *slow-2/grow-2*, is active when paternally inherited. a**, Alignment of Illumina short-reads to the EG6180 de novo assembly. Highlighted region is Chr. III EG6180 region homologous to NIC203 *slow-1/grow-1*. **b**, *slow-2* expression in EG6180 2-cell stage embryos by smFISH. *pgl-1* serves as a positive control (quantification in Extended Data Fig. 9a). **c**, Mating of *slow-2(–)/grow-2(–)* double mutant NIL (hermaphrodites) to the parental EG6180 line causes developmental delay of homozygous double-mutant F₂ individuals indicating that *slow-2/grow-2* is a TA, (mean +/−95% CI is shown).

**d**, Two TAs are active in crosses between the NIL and EG6180: *slow-1/grow-1* and *slow-2/grow-2*, respectively (Fig. 1b). The penetrance of the *slow-2/grow-2* TA is incomplete; however, the toxin is equally active when maternally or paternally inherited (n_M = 40, n_P = 67, two-sided Fisher's exact test, p = 0.31, mean +/−95% CI is shown). **e**, Reciprocal crosses between the NIL and EG6180 followed by RNA-seq of their F₁ progeny indicates that *slow-2* transcripts are equally abundant when maternally or paternally inherited (two-sided unpaired t-test, p = 0.6451, mean +/−SEM is shown).

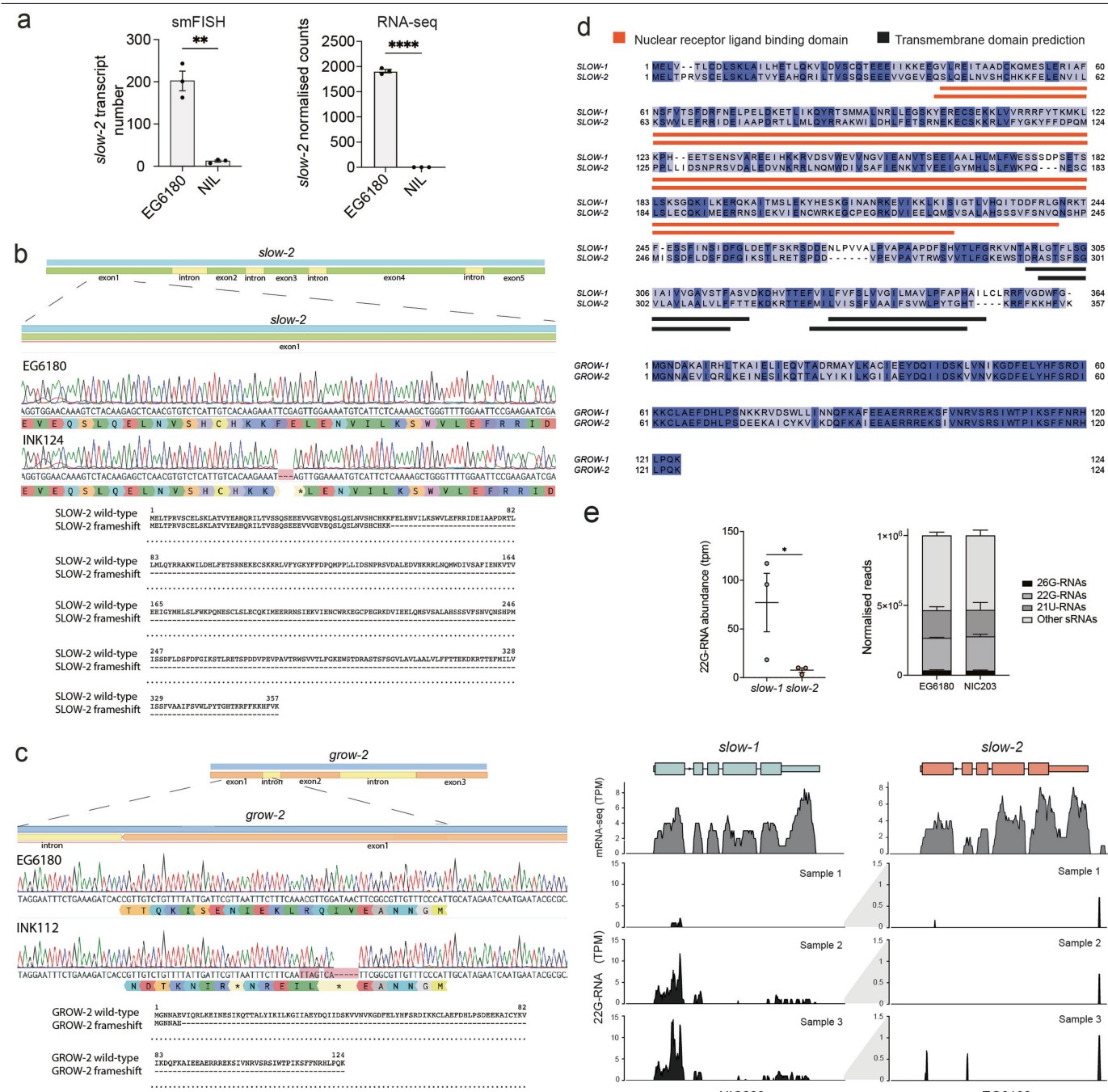

**Extended Data Fig. 9 | Characterization of *slow-2* and *grow-2* null alleles and comparison to *slow-1*. a**, Quantification of *slow-2* expression in EG6180 embryos by smFISH (N = 3, n = 16–30 per repeat, two-sided unpaired t-test, p = 0.0013, mean +/−SEM is shown). *slow-2* expression in EG6180 and absence in the NIL was also confirmed by RNA-seq data (n = 3, two-sided unpaired t-test, p < 0.0001, mean +/−SEM is shown). **b**, The *slow-2* null allele generates a three base pair deletion in the first exon, which creates a premature stop codon. This results in a shorter peptide (54 aa compared to 357 aa). Representative Sanger sequences of WT and mutant alleles. **c**, The *grow-2* null allele generates a deletion and

frameshift in the first exon which creates a premature stop codon. This results in a shorter peptide (6 aa compared to 124 aa). Representative Sanger sequences of WT and mutant alleles. **d**, Protein alignment of SLOW-1 and SLOW-2 (top), and protein alignment of GROW-1 and GROW-2 (bottom). **e**, Quantification of total 22G-RNAs derived from *slow-1* and *slow-2* (top left, two-sided unpaired t-test p = 0.042, mean +/−SEM is shown). Distribution of sRNA per parental strain (top right, mean + SEM is shown). RNA-seq and 22G-sRNA short-reads aligned against *slow-1* and *slow-2*. sRNA libraries were generated in biological triplicates. Notice the differences in the y-axis scale between *slow-1* and *slow-2*.

# Reporting Summary

## Statistics

For all statistical analyses, confirm that the following items are present in the figure legend, table legend, main text, or Methods section.

| n/a | Confirmed | |
|---|---|---|
| ☐ | ☒ | The exact sample size (*n*) for each experimental group/condition, given as a discrete number and unit of measurement |
| ☐ | ☒ | A statement on whether measurements were taken from distinct samples or whether the same sample was measured repeatedly |
| ☐ | ☒ | The statistical test(s) used AND whether they are one- or two-sided *Only common tests should be described solely by name; describe more complex techniques in the Methods section.* |
| ☒ | ☐ | A description of all covariates tested |
| ☒ | ☐ | A description of any assumptions or corrections, such as tests of normality and adjustment for multiple comparisons |
| ☐ | ☒ | A full description of the statistical parameters including central tendency (e.g. means) or other basic estimates (e.g. regression coefficient) AND variation (e.g. standard deviation) or associated estimates of uncertainty (e.g. confidence intervals) |
| ☐ | ☒ | For null hypothesis testing, the test statistic (e.g. *F*, *t*, *r*) with confidence intervals, effect sizes, degrees of freedom and *P* value noted *Give P values as exact values whenever suitable.* |
| ☒ | ☐ | For Bayesian analysis, information on the choice of priors and Markov chain Monte Carlo settings |
| ☒ | ☐ | For hierarchical and complex designs, identification of the appropriate level for tests and full reporting of outcomes |
| ☒ | ☐ | Estimates of effect sizes (e.g. Cohen's *d*, Pearson's *r*), indicating how they were calculated |

*Our web collection on statistics for biologists contains articles on many of the points above.*

## Software and code

Policy information about availability of computer code

| Data collection | N/A |
|---|---|
| Data analysis | Flye Assembler v2.7.1 , Sibelia, Ragout, DEseq2, STAR, Cutadapt, samtools v1.18, hisat2 v2.1, seqkit v0.13 , Gviz, ViennaRNA Package v2.0.58, blast v2.2.26, Fiji (v1.53r) |

For manuscripts utilizing custom algorithms or software that are central to the research but not yet described in published literature, software must be made available to editors and reviewers. We strongly encourage code deposition in a community repository (e.g. GitHub). See the Nature Portfolio guidelines for submitting code & software for further information.

## Data

Policy information about availability of data

All manuscripts must include a data availability statement. This statement should provide the following information, where applicable:
- Accession codes, unique identifiers, or web links for publicly available datasets
- A description of any restrictions on data availability
- For clinical datasets or third party data, please ensure that the statement adheres to our policy

Sequencing data are available under NCBI project PRJNA850171. Raw data for all genetic crosses including references to figures is found in Extended Data Table 1

## Human research participants

Policy information about studies involving human research participants and Sex and Gender in Research.

| | |
|---|---|
| Reporting on sex and gender | It does not apply to our study |
| Population characteristics | It does not apply to our study |
| Recruitment | It does not apply to our study |
| Ethics oversight | It does not apply to our study |

Note that full information on the approval of the study protocol must also be provided in the manuscript.

# Field-specific reporting

Please select the one below that is the best fit for your research. If you are not sure, read the appropriate sections before making your selection.

☒ Life sciences          ☐ Behavioural & social sciences          ☐ Ecological, evolutionary & environmental sciences

For a reference copy of the document with all sections, see nature.com/documents/nr-reporting-summary-flat.pdf

# Life sciences study design

All studies must disclose on these points even when the disclosure is negative.

| | |
|---|---|
| Sample size | For each genetic cross, we phenotyped at least 100 F2 individuals. This is in our experience sufficient to significantly distinguish between an active or inactive toxin-antidote element (segregation of a single locus Mendelian trait). |
| Data exclusions | We excluded from the analysis the progeny of individuals resulting from self-fertilization (genotyping of F1 by PCR). Progeny from heterozygous F1 individuals that could not be genotyped due to to technical problems were noted as "n.g." and included in the analyses. |
| Replication | All genetic crosses were independently performed at least twice starting from independent nematode cultures. And each replicate included ten independent F1 individuals and 10 F2 progeny per F1 (on average). All raw numbers can be found on Extended Data Table 1 |
| Randomization | We did not perform randomization |
| Blinding | Phenotypic scoring of all F2 progeny was performed prior and independently of genotyping. |

# Reporting for specific materials, systems and methods

We require information from authors about some types of materials, experimental systems and methods used in many studies. Here, indicate whether each material, system or method listed is relevant to your study. If you are not sure if a list item applies to your research, read the appropriate section before selecting a response.

### Materials & experimental systems

| n/a | Involved in the study |
|---|---|
| ☐ | ☒ Antibodies |
| ☒ | ☐ Eukaryotic cell lines |
| ☒ | ☐ Palaeontology and archaeology |
| ☐ | ☒ Animals and other organisms |
| ☒ | ☐ Clinical data |
| ☒ | ☐ Dual use research of concern |

### Methods

| n/a | Involved in the study |
|---|---|
| ☐ | ☒ ChIP-seq |
| ☒ | ☐ Flow cytometry |
| ☒ | ☐ MRI-based neuroimaging |

## Antibodies

| | |
|---|---|
| Antibodies used | Membranes were incubated with anti-FLAG M2 (mouse,1:2000, Sigma-Aldrich, F3165) or anti-Actin (rabbit, 1:3000, Abcam, ab13772) primary antibody in a blocking solution overnight at 4°C. |
| Validation | Both antibodies are highly used commercial antibodies. We have also validated the anti-FLAG M2 antibody for WB and immunofluoresence in C. tropicalis using negative controls (lines without FLAG tag) |

## Animals and other research organisms

Policy information about <u>studies involving animals</u>; <u>ARRIVE guidelines</u> recommended for reporting animal research, and <u>Sex and Gender in Research</u>

| Laboratory animals | In this study we only used previously characterized C. tropicalis nematode lines  NIC203 and EG6180 described  in Ben-David et al. Current Biology (2021) All mutants lines are available in Extended Table 3 |
|---|---|
| Wild animals | N/A |
| Reporting on sex | N/A |
| Field-collected samples | N/A |
| Ethics oversight | N/A |

Note that full information on the approval of the study protocol must also be provided in the manuscript.

## ChIP-seq

### Data deposition

☒ Confirm that both raw and final processed data have been deposited in a public database such as <u>GEO</u>.

☐ Confirm that you have deposited or provided access to graph files (e.g. BED files) for the called peaks.

| Data access links
*May remain private before publication.* | Sequencing data are available under NCBI project PRJNA850171. |
|---|---|
| Files in database submission | H3K9me3 ChIP and Input controls samples for two conditions (NIL and F4 population) are provided |
| Genome browser session
(e.g. <u>UCSC</u>) | *Provide a link to an anonymized genome browser session for "Initial submission" and "Revised version" documents only, to enable peer review.  Write "no longer applicable" for "Final submission" documents.* |

### Methodology

| Replicates | three biological replicates |
|---|---|
| Sequencing depth | We aim to generate at least 20M reads per sample, which corresponds to ~25X coverage of the whole genome of C.tropcalis. |
| Antibodies | 2 µg of anti-H3K9me3 antibody (Ab8898, Abcam) |
| Peak calling parameters | . Peaks were called by macs2 v2.2.5 with --broad and --mfold 1 50 options85 |
| Data quality | *Describe the methods used to ensure data quality in full detail, including how many peaks are at FDR 5% and above 5-fold enrichment.* |
| Software | MACS2 |

