## [Peer Review File · Nature]

Manuscript Title: Selfish conflict underlies RNA-mediated parent-of-origin effects

Reviewer Comments & Author Rebuttals

Reviewer Reports on the Initial Version:

Referees' comments:

The manuscript by Pilota et al. examines an intriguing and novel example of a toxin-antidote (TA) system which results in a development delay for the progeny without the antidote in a parent-of-origin-specific manner. The authors meticulously dissect the mechanisms of this process using genetic, cell biology and genomic approaches. First, they show that if the slow-1/grow-1 TA system is introduced via a cross from the maternal side ("grandmother," P0), the resulting F2s who do not inherit the slow-1/grow-1 locus have a delay in development. However, when the slow-1/grow-1 system is introduced via a cross from the paternal side (grandfather, P0), the resulting F2s who do not have the slow-1/grow-1 locus do not have a delay in development, despite the fact that F1 hermaphrodites are heterozygous for the locus. The authors also demonstrate that the parent of origin effect is unique to this TA system relative to similar ones (e.g. slow-2/grow-2), further emphasizing the novelty of their discovery.

The authors then systematically dissect the mechanisms of this effect, in an effort to identify what factors, pathways, and molecules influence the parent-of-origin differences. They examine a role for the piRNA pathway, and implicate one of two PIWI paralogs, PRG-1.2, and two specific piRNAs in the process of silencing paternally inherited slow-1/grow-1 in the F1. They also use careful genetic approaches to demonstrate that slow-1/grow-1 transcripts must be maternally inherited to "license" the maternal allele for expression in the F1 (i.e., make it exempt from piRNA silencing) and for the toxin system to function in the F2.

Overall, the authors have done an exceptional job of defining this novel and exciting system, especially considering the technically challenging nature of the experiments and of a non-model nematode. The progression of the manuscript is logical and convincing overall. The conclusions are well supported by the data. However, the way in which the authors articulate and outline their experiments in the figures (and in some parts of the text) could be significantly improved for clarity, especially for those not well versed in *C. elegans/tropicalis* genetics. There is a certain amount of mental gymnastics required to understand these data, which is understandable given the complexity of the genetic/epigenetic phenomena described here, but there are ways the authors could communicate the experimental setup and conclusions to make them more accessible to a broad audience.

Major concerns:

1. The small RNA aspect of this mechanism should be better described and fleshed out. 22G-RNAs are thought to be amplified downstream of the piRNAs. Therefore, small RNA sequencing in the piRNA deletion mutant could be used to gain insight into the regulation and any licensing of the system by other types of small RNAs. For instance, are the 22Gs mapping to the 5' end of slow-1 still present even when piRNA pathway activity is abrogated?

2. Related to this point, a brief mention or overview of the Argonautes present in the *C. tropicalis* genome could enable additional conclusions about the potential mechanisms of the system. For example: Does *C. tropicalis* have a CO4F112.1 ortholog(s) (the Argonaute most closely related to CSR-1) and if so could it potentially compensate for CSR-1 in terms of licensing? Alternatively, are there other CSR-1 paralogs/orthologs that could be active in the system? Could HRDE-1 homologues be involved in the epigenetic silencing in males? No experiments are required here, but some mention of this might be helpful for those who are familiar with *C. elegans* small RNA pathways.

3. Figure 2 shows the protein inheritance of SLOW-1 protein, yet slow-1 mRNA is the predominant effector of the epigenetic phenomenon studied here. The manuscript would benefit from the inclusion of smFISH of slow-1 transcripts, as was done with slow-2 in Fig 5e. This would show if the RNA is stable throughout embryogenesis. In addition, males should be included to assess protein and mRNA levels of slow-1 in figure 2.

4. Overall, the model should be significantly changed to spell out the data presented in the paper more clearly and accurately. The reader would be able to better understand the mechanism as they move through the experiments as they read the paper by using the model as a point of reference. First, the germline does not add anything to the figure and could be replaced with the 3 scenarios for F2s similar to that of Fig 1a. The model also makes certain logical leaps and implies direct effects where they have not been shown. By adding "?" in certain places the authors can better describe what they found out about the mechanism and demonstrate what remains to be explored. For example - the mRNAs would not be directly repressing piRNAs, the addition of a yet unknown intermediate step helps clarify this part of the model.

5. Figure 3A outlines the crosses/transgenerational nature of the phenomenon, but skips some steps. It should fully lay out the genotypes from the P0 to the 3rd (or 9th) generation, especially because as shown it is not clear if the gene was maintained as heterozygous or homozygous while self-fertilizing.

6. A clearer (graphical) representation of the experiments performed for figure 4d-f should be provided. These are among the core experiments of the manuscript and also the most complex to follow by the description in the text alone. The depiction of the crosses should be clarified and include when and what proteins and mRNAs are present to help the readers understand what molecules/factors are being assessed as potential couriers of epigenetic information.

7. The call-outs of Supplemental Figures in the text should detail which panel of the figure is being referenced, as it is not always clear and led to some confusion.

Minor concerns:

1. Clarification of genotypes of animals in Figure 2. (This is actually a general comment throughout the figures-spell everything out, try to not skip steps.)
2. Mapping of the protein domains to the amino acid alignment in extended figure 2a (PRG-1) would be helpful (as was done for the SLOW proteins in extended 4d) to understand potential differences in the two PRG proteins and their different effects.
3. It might be helpful to mention what kinds of proteins SLOW-1 and GROW-1 are and how SLOW-1 inhibits development.

Typos and minor corrections:

- The embryos in Fig 3C appear to be oriented with posterior (P) cells to the left. Could the authors double check on this, and if it is the case, please rotate them so that the posterior is to the right.
- Line 229: levered should be leveraged.
- Line 256: CSR-1b is technically constitutively expressed in the germline, not only the “female” (perhaps oogenic would be a better term here) germline.

Referee #2 (Remarks to the Author):

The nonequivalence of maternal and paternal genomic imprinting is the inevitable consequence of conflictive selective forces acting on differentially expressed parental alleles, but how these epigenetic differences evolve in the first place is poorly understood. In this manuscript by Pliota et al, the authors identified parasitic conflict as an evolutionary force fueling the emergence of imprinting. First, they found that the *slow-1/grow-1* TA was specifically inactive when paternally inherited. Next, they showed that PIWI protein PRG-1.2 and two complementary piRNAs were required to repress the paternal inherited *slow-1/grow-1* allele. Intriguingly, they showed that maternal *slow-1* transcripts overrode the repressive effect of piRNAs on *slow-1* expression when *slow-1/grow-1* was maternally inherited, thereby resulting the toxin induced-developmental delay. Overall, this is an interesting story supported by comprehensive nematode genetic data, bringing a mechanistic insight into the parent-of-origin effect on TA expression. Most of the experiments are well designed, and many of the data presented are convincing. Nonetheless, the data are weaker for the involvement of piRNAs in controlling parent-specific gene expression. Following comments may help for further improvement.

Major concerns:

1. The authors provided genetic evidence demonstrating that piRNA pathway represses paternal

slow-1 expression when maternal *slow-1* transcripts are absent in the eggs. More interestingly, this repressed state could be inherited for at least 3 generations and reverted after 9 generations. The authors found that the piRNAs paired with the 3'UTR of *slow-1* transcripts dominated the repression. How these piRNAs function to repress *slow-1* expression? Post-transcriptionally or epigenetically? Biochemical evidence should be provided to validate this regulation, given it as the key part of the working model.

2. PRG-1.2 and PRG-1.1 are highly identical Piwi proteins. What makes PRG-1.2, rather than PRG-1.1, critical for *slow-1* repression in the germline of the F1. This should be further explored. For examples, whether the two proteins are equivalently expressed and similarly associate with Ctr-21ur-06949 and Ctr-21ur-06917?

3. The authors proposed an intriguing model that maternally inherited *slow-1* mRNA itself, even at very low levels, might act as a “licensing” signal probably by a catalytic way to counteract piRNA-mediated repression in the zygote. How the maternally inherited *slow-1* mRNA inhibits piRNA repression? This should be experimentally explored.

4. Fig. 1d shows that *slow-1* mRNAs were still present in F1 mothers when *slow-1/grow-1* was paternally inherited, despite significantly lower than maternally inherited. However, the authors argue that very low levels of *slow-1* mRNA could inhibit piRNA repression. How can the authors reconcile these data?

Minor points

1. The specific panel should be referred when the Extended Data were mentioned in the text.
2. In Fig 5c, the labeling of hermaphrodites should be “*slow-2(-); grow-2(-)*” rather than “*slow-2(-); grow(-)*”.

Referee #3 (Remarks to the Author):

In this solid, well-documented study, Alejandro Burga and his colleagues explored an unusual ‘toxin-antidote’ (TA) element in *C. elegans*, called *slow-1/grow-1*. They describe a novel mechanism that in special cases of heterozygosity (obtained by outcrossing) leads to a parent-of-origin dependent repression of the developmentally toxic *slow-1* gene in the TA. Although the underlying mechanisms are still only partially understood, this novel finding provides a first example of a parent-of-origin dependent effect on an endogenous gene in (isolates of) a *Caenorhabditis* species (i.e., *C. tropicalis*).

Specifically, they find that the *slow-1/grow-1* TA element becomes inactive when paternally inherited (in F2 animals), and that this involves repression in the germline of heterozygous mothers. Interestingly, they demonstrate that piRNA loci and Piwi Argonaute activity are required for this repressive process.

Similarly as had been reported before for transgenes in *C. elegans*, once achieved, the repressed state of *slow-1* is trans-generationally inherited across many generations. Importantly, the *slow-1* repression is not observed in case there is loading of *slow-1* transcripts into the oocyte (which explains that the effect

on the paternally inherited *slow-1* is seen only in heterozygotes) and the authors show that these transcripts specifically inhibit the piRNA mediated repression.

This interesting study provides important novel insights into the rapid evolution of parasitic elements in worms through genetic conflict and into the emergence of a mechanism that bears similarities (in the F2 animals) with parental genomic imprinting in some insect species and mammals. The finding that the piRNA pathway is involved in the process is very interesting. In mice, for instance, at only one (retrotransposon-derived) imprinted gene locus, the establishment of DNA methylation-linked allelic gene repression is controlled by piRNAs, suggesting that this could be an ancestral imprinting mechanism.

Nevertheless, at the molecular level many key questions remain, particularly as concerns the trans-generational nature of the piRNA induced *slow-1* repression. It is somewhat unfortunate that this study includes mostly gene expression studies, and did not explore what happens at the chromatin level at the studied TA element. Without chromatin-level insights, it is difficult to know what might be the epigenetic feature that confers the long-term, transgenerational inheritance of the repressed *slow-1* state.

Main points:

-What underlies the transgenerational inheritance of the *slow-1* silenced state? Is this linked to acquisition of heterochromatin-like features, similarly as has been reported for trans-generational inheritance of piRNA mediated transgene repression in *C. elegans*. The authors should explore this important outstanding question by performing chromatin immunoprecipitation experiments. Importantly, such studies could provide insights into the nature of the epigenetic 'mark' that confers the trans-generational inheritance.

-The authors convincingly show that the piRNA pathway Prg-1.2 protein is essential for the establishment of the silencing of the paternally inherited *slow-1* gene (in their specific heterozygous background). Throughout subsequent generations, is this protein in the oocyte/zygote also important for the observed trans-generational inheritance of the repressed gene state? Once established, can the repressed state of *slow-1* be trans-generationally inherited without the expression of Prg-1.2?

-It is not clear from the presented data how precisely the expression of the *slow-1* mRNAs inhibits the piRNA-mediated repression of the paternally-inherited TA element? Could the authors provide further insights into this mechanism, or explain this aspect better in their text?

-The title seems misleading. Particularly, the term 'parent-specific' is very confusing. The repression pattern observed here is not at all parent-specific (specific to the parent). Rather, it is parent-of-origin dependent.

Minor point:

- In Figure 1d, what explains the outlier that shows a high level of slow-1 mRNA expression upon paternal inheritance (NIL)? Could the authors comment on this in the text?

Author Rebuttals to Initial Comments:

Referee #1 (Remarks to the Author):

The manuscript by Pilota et al. examines an intriguing and novel example of a toxin-antidote (TA) system which results in a development delay for the progeny without the antidote in a parent-of-origin-specific manner. The authors meticulously dissect the mechanisms of this process using genetic, cell biology and genomic approaches. First, they show that if the *slow-1/grow-1* TA system is introduced via a cross from the maternal side ("grandmother," P0), the resulting F2s who do not inherit the *slow-1/grow-1* locus have a delay in development. However, when the *slow-1/grow-1* system is introduced via a cross from the paternal side (grandfather, P0), the resulting F2s who do not have the *slow-1/grow-1* locus do not have a delay in development, despite the fact that F1 hermaphrodites are heterozygous for the locus. The authors also demonstrate that the parent of origin effect is unique to this TA system relative to similar ones (e.g. *slow-2/grow-2*), further emphasizing the novelty of their discovery.

The authors then systematically dissect the mechanisms of this effect, in an effort to identify what factors, pathways, and molecules influence the parent-of-origin differences. They examine a role for the piRNA pathway, and implicate one of two PIWI paralogs, PRG-1.2, and two specific piRNAs in the process of silencing paternally inherited *slow-1/grow-1* in the F1. They also use careful genetic approaches to demonstrate that *slow-1/grow-1* transcripts must be maternally inherited to "license" the maternal allele for expression in the F1 (i.e., make it exempt from piRNA silencing) and for the toxin system to function in the F2.

Overall, the authors have done an exceptional job of defining this novel and exciting system, especially considering the technically challenging nature of the experiments and of a non-model nematode. The progression of the manuscript is logical and convincing overall. The conclusions are well supported by the data. However, the way in which the authors articulate and outline their experiments in the figures (and in some parts of the text) could be significantly improved for clarity, especially for those not well versed in *C. elegans/tropicalis* genetics. There is a certain amount of mental gymnastics required to understand these data, which is understandable given the complexity of the genetic/epigenetic phenomena described here, but there are ways the authors could communicate the experimental setup and conclusions to make them more accessible to a broad audience.

We thank Reviewer #1 for their positive evaluation, which specifically highlights our careful and systematic dissection of this novel epigenetic phenomenon. In this revised manuscript, we have taken special care to clarify and better communicate our findings to make them more accessible to our colleagues.

Major concerns:

1. The small RNA aspect of this mechanism should be better described and fleshed out. 22G-RNAs are thought to be amplified downstream of the piRNAs. Therefore, small RNA sequencing in the piRNA deletion mutant could be used to gain insight into the regulation and any licensing of the system by other types of small RNAs. For instance, are the 22Gs mapping to the 5' end of *slow-1* still present even when piRNA pathway activity is abrogated?

Encouraged by this comment, we decided to conduct a more comprehensive characterization of the downstream sRNA populations. In our first submission, we reported the detection of 22G-RNAs complementary to *slow-1* with a strong bias toward these sRNAs mapping to the 5' of the transcript. However, an important limitation to this approach was that the sRNAs were extracted from the *slow-1/grow-1* NIL parental line, which readily expresses the *slow-1* toxin. Thus, not surprisingly, the overall abundance of

22G-RNAs complementary to *slow-1* was exceptionally low compared to the total population, making it challenging to quantify their relative abundance in wild type versus mutant background conditions. To illustrate this challenge, we needed to sequence approximately 80 million reads from a sRNA-seq library to detect at most three copies of a given 22G-RNA complementary to *slow-1*. Therefore, we reasoned that it would be more informative and cost-effective to study sRNAs following paternal inheritance of the TA, as we expected repressive sRNAs to accumulate. To accomplish this, we conducted paternal crosses, isolated F₂ homozygous carriers of the repressed TA allele, and expanded these populations for two additional generations to obtain enough starting material for the sRNA-seq protocol (see Extended Data Fig. 5f and below).

Extended Data Fig. 5f. (highlight of new panel) f, Mating scheme to generate an F₄ homozygous population for the repressed *slow-1/grow-1* TA following its paternal inheritance. This F₄ population was subjected to sRNA-seq in biological quadruplicates (each time starting from an independent initial parental cross). As a control for an active or licensed TA, we performed sRNA-seq in the NIL parental line in biological quadruplicates. Red asterisk denotes repressed *slow-1* state.

Remarkably, we found that paternal inheritance of the TA led to a >30-fold increase in the population of 22G-RNAs complementary to *slow-1* compared to the maternally licensed TA (see Fig. 3i,j and below; only two of four biological repeats are shown for simplicity). It is important to highlight that this 22G-RNA population was detected in the great-great-granddaughters of the original male TA carriers, suggesting that these sRNA could mediate the inheritance of the *slow-1* repressed epigenetic state. Interestingly, we also identified a significant accumulation of 26G-RNAs complementary to *slow-1*, which were virtually undetectable in the parental NIL strain (see Fig. 3j and below).

Fig. 3 (highlight of new panel) i, Coverage of 22G-RNAs mapping to *slow-1* mRNA when *slow-1* is licensed (NIL control line) or repressed (four generations following its paternal inheritance). For each condition we

performed four sRNA-seq biological replicates. Only two repeats are shown for simplicity, the trend was identical in all of them. Total number of 22G-RNAs in each library is the same. j, Quantification of 22G-RNA and 26-RNA populations mapping to *slow-1*. Four biological replicates are included. sRNA abundance is normalized as transcripts per million (tpm).

In agreement with the *C. elegans* literature^{1,2}, we detected a local peak of 22G-RNA amplification within the predicted binding site of the two piRNAs that are necessary for *slow-1* repression (Extended Data Fig. 5g and below). However, it is evident that these peaks are modest and are not the major sites from which 22G-RNAs are generated (largest peaks are located within the first exon). Since 22G-RNAs were readily detectable in the great-great-granddaughters of the original male TA carriers and most of them are not found near predicted piRNA binding sites (Fig. 3i and Supplementary Data 1), our results suggest that most of these repressive sRNA populations are ‘tertiary’ in origin and could mediate the inheritance of the *slow-1* epigenetic state.

In summary, we now report the identification of 22G-RNA and 26G-RNA populations specifically associated with the transgenerational inheritance of *slow-1* repression following paternal inheritance of the TA. These new results can now be found in the main text lines 303-317.

Extended Data Fig. 5g. (highlight of new panel) g, Zoom-in into the *21ur-06949* and *21ur-06917* predicted binding sites in *slow-1* and the 22G-RNAs mapping to these regions. The 22G-RNAs are derived from the F₄ “*slow-1* repressed” population. Each track is one of four biological quadruplicates. A modest but clear peak is observed within the predicted binding region.

2. Related to this point, a brief mention or overview of the Argonautes present in the *C. tropicalis* genome could enable additional conclusions about the potential mechanisms of the system. For example: Does *C. tropicalis* have a C04F112.1 ortholog(s) (the Argonaute most closely related to CSR-1) and if so could it potentially compensate for CSR-1 in terms of licensing? Alternatively, are there other CSR-1 paralogs/orthologs that could be active in the system? Could HRDE-1 homologues be involved in the epigenetic silencing in males? No experiments are required here, but some mention of this might be helpful for those who are familiar with *C. elegans* small RNA pathways.

We have now comprehensively annotated all Argonaute proteins in *C. tropicalis* based on: i) homology to known PFAM Argonaute domains, ii) completeness compared to *C. elegans* orthologs, and iii) synteny. Overall, we identified 26 Argonaute proteins in *C. tropicalis*, including one putative pseudogene. In comparison, there are 20 Argonautes present in *C. elegans* including WAGO-5, which was recently classified as a pseudogene. To facilitate further analysis and comparison to previous approaches³, we built a phylogenetic tree that included all known *C. elegans* and *C. tropicalis* Argonautes (see Extended Data Fig. 2 and below).

Extended Data Fig. 2 (new figure). Phylogenetic tree of *C. elegans* (yellow) and *C. tropicalis* (blue) Argonaute proteins. Putative pseudogene in gray. Also included *A. thaliana* AGO1 and *D. melanogaster* PIWI. Red circles denote bootstrap values >95%.

In addition to *Ctr*-CSR-1, we did not find evidence of additional closely related homologs to CSR-1. *C. elegans* CSR-1 and *C. tropicalis* CSR-1 are clear 1:1 orthologs. However, we identified two homologs of *C. elegans* C04F112.1/VSRA-1 (see Extended Data Fig. 2). While it is conceivable that these genes could partially compensate for the loss of CSR-1, it seems unlikely because VSRA-1 and CSR-1 bind largely non-overlapping small RNA populations in *C. elegans*³. Redundancy between VSRA-1 and CSR-1 is an interesting possibility that could be explored in future work.

The *C. elegans* nuclear Argonaute HRDE-1 binds secondary 22G-RNAs and mediates the epigenetic repression of piRNA targets in the germline. HRDE-1 has a close paralog, WAGO-10, which is only expressed cytoplasmically during spermatogenesis. Our phylogenetic analysis revealed that *C. tropicalis* has one ortholog that is closely related to both HRDE-1 and WAGO-10, which we named *Ctr*-HRDE-1. To test the contribution of *Ctr*-HRDE-1 to *slow-1* repression, we generated a putative loss of function *Ctr*-hrde-1 allele using CRISPR/Cas9 (Extended Data Fig. 5a,c). While heterozygous *Ctr*-hrde-1 were viable and fertile, their homozygous mutant progeny was 100% sterile. Because the infertility of *hrde-1* prevented us from testing its role in *slow-1* repression, we next focused on other putative nuclear Argonautes. The *C. tropicalis* genome codes for an additional branch of three Argonautes related to HRDE-1, which we named *Ctr*-WAGO-11, *Ctr*-WAGO-12, and *Ctr*-WAGO-13. The first two are *bona fide* genes, whereas the last one is likely a pseudogene (see Methods). To study the role of these novel *hrde-1* paralogs, we generated a *Ctr*-wago-11 mutant. Unlike *hrde-1*, *wago-11* mutants were viable and had no obvious developmental defects. Yet, genetic crosses indicated that loss of *wago-11* had no impact on *slow-1* repression (Fig. 3h). Thus, the essential role of the piRNA pathway in *C. tropicalis* and potential layers of redundancy pose an important challenge when dissecting the

contribution of nuclear Argonautes to *slow-1* repression. However, we successfully identified the histone methyltransferase SET-32 as a protein required for *slow-1* repression (Fig. 3h). In *C. elegans*, HRDE-1 recruits SET-32 and other epigenetic co-factors to the target locus in order to repress it (please refer to Reviewer#3 point 1 for further details). These new results can now be found in the main text lines 284-296.

Overall, this comprehensive annotation and initial characterization Argonaute proteins in *C. tropicalis* illustrates the importance of studying different nematode species. For instance, *C. elegans prg-1* and *hrde-1* mutants are viable and only develop a gradual decline in fertility across generations, whereas *C. tropicalis prg-1.1; prg1.2* and *hrde-1* mutants are fully sterile within a generation. Conversely, *csr-1* is essential in *C. elegans* but partially viable in *C. tropicalis*. We hope these new mutants will prove valuable for researchers in the sRNA field studying the function and evolution of Argonaute proteins.

3. Figure 2 shows the protein inheritance of SLOW-1 protein, yet *slow-1* mRNA is the predominant effector of the epigenetic phenomenon studied here. The manuscript would benefit from the inclusion of smFISH of *slow-1* transcripts, as was done with *slow-2* in Fig 5e. This would show if the RNA is stable throughout embryogenesis. In addition, males should be included to assess protein and mRNA levels of *slow-1* in figure 2.

We agree with the reviewer that the expression pattern of *slow-1* is important to interpret our experiments. The reason why we did not include *slow-1* smFISH data in this manuscript is that we already performed and published these experiments in a previous paper (see Fig. 6C in Ben-David et al Current Biology 2021)⁴. In summary, smFISH revealed that, as expected, *slow-1* mRNA is maternally loaded into eggs (*slow-1* present in embryos before 4-cells stage), levels are relatively stable from 5-20 cell-stage and from 20 cell-stage onwards there is a decline in mRNA levels. By the comma-stage *slow-1* is only detectable in the Z2/Z3 germline precursor cells. We acknowledge that this was not clear in our manuscript, and we have now modified the main text to reflect this (see main text lines 99-102).

To investigate the expression of *slow-1* in males, we took two complementary approaches, RT-qPCR and fluorescence microscopy to quantify mRNA and protein levels, respectively. First, we performed RT-qPCR experiments to quantify *slow-1* expression levels in hermaphrodites and males in the NIL and EG6180 (negative control) backgrounds. For each biological replicate we handpicked 100 hermaphrodites or 50 males. While RT-qPCR revealed specific amplification of *slow-1* for both hermaphrodites and males, the Ct values for males were much higher than those for hermaphrodites (34.27 compared to 28.31 on average, despite 32x more template used in the RT-qPCR reactions for male samples), and very variable (standard deviation of 1.55 compared to 0.65 in the NIL), suggesting that *slow-1* is expressed in males but at much lower levels, which prevented us from deriving a reliable estimate.

Quantification of the *mScarlet::slow-1* reporter in males using fluorescence microscopy painted a similar picture. We could confidently detect mScarlet::SLOW-1 in males above background (compared to a non-fluorescent strain); however, the signal intensity was much lower compared to the female gonad. We then performed reciprocal crosses between *mScarlet::slow-1* and EG6180 parental strains and quantified the fluorescence of F₁ males. Unfortunately, mScarlet::SLOW-1 signal in heterozygous F₁ males was not significantly different from autofluorescence (negative control), indicating that mScarlet::SLOW-1 was not-abundant enough to be quantified.

In summary, due to the low level of expression of *slow-1* in the male gonad, various technical limitations hindered our ability to confidently investigate the parent-of-origin effect in males. As this result is not central to the main claim of the paper and would necessitate substantial technical investment for relatively marginal gains, we respectfully propose that exploring this phenomenon in a separate study would be more suitable. A paragraph summarizing our findings on *slow-1* expression in males and associated challenges can now be found in the methods section (see method section line 1162-1167).

4. Overall, the model should be significantly changed to spell out the data presented in the paper more clearly and accurately. The reader would be able to better understand the mechanism as they move through the experiments as they read the paper by using the model as a point of reference. First, the germline does not add anything to the figure and could be replaced with the 3 scenarios for F2s similar to that of Fig 1a. The model also makes certain logical leaps and implies direct effects where they have not been shown. By adding “?” in certain places the authors can better describe what they found out about the mechanism and demonstrate what remains to be explored. For example - the mRNAs would not be directly repressing piRNAs, the addition of a yet unknown intermediate step helps clarify this part of the model.

As suggested by the reviewer, we made significant updates to the model illustrating the *slow-1/grow-1* parent-of-origin effect to facilitate the critical reading of the manuscript (Fig. 6). We replaced the germline drawings for the genotypes like those in Fig 1a and added much more detail to our model. These updates are not only aesthetic but also reflect a deeper molecular understanding of both licensing and piRNA-mediated repression that we gained during the revision of this manuscript.

The new findings are described in detail in the response to specific comments made by reviewers. Here we only summarize the highlights: a) Maternal *slow-1* transcripts counteract the initiation of piRNA-mediated repression, b) Both PRG-1.1 and PRG-1.2 are involved in *slow-1* repression (the effect of PRG-1.1 is masked by genetic redundancy), c) Maternal PRG-1.1 activity is required for the initiation of piRNA-mediated repression but not for its maintenance, d) The histone methyltransferase SET-32 is necessary for *slow-1* repression, e) Repression of *slow-1* occurs at the transcriptional level (pre-mRNA). We have also highlighted those intermediates steps and players that are unknown for now.

Fig. 6 in the original manuscript:

Fig. 6 in the revised manuscript:

Fig. 6. Model illustrating the *slow-1/grow-1* parent-of-origin effect. a, Maternal-inheritance of the *slow-1/grow-1* TA. *slow-1* transcripts deposited in the egg by the mother are sufficient and necessary to activate zygotic *slow-1* in the germline of the F₁ progeny. Epigenetic licensing stems from inhibiting the repressive action of piRNAs. Licensing occurs post-transcriptionally, likely by inhibiting piRNA target recognition or secondary sRNA amplification in the perinuclear nuage (green condensates). F₁ heterozygous mothers load SLOW-1 toxin into all their eggs. F₂ homozygous non-carrier individuals are developmentally delayed because they do not express the zygotic antidote. **b**, Paternal-inheritance of the *slow-1/grow-1* TA. In the absence of *slow-1* maternal transcripts, piRNAs repress the transcription of *slow-1* in the germline of heterozygous F₁ mothers. Initiation of repression requires maternal PRG-1 activity, which uses the *slow-1* zygotic transcript as a template for the generation of 22G-RNAs complementary to the target. These 22G-RNAs are then likely bound by nuclear Argonautes, such as HRDE-1, which in turn recruit chromatin-modifying enzymes to the target locus. The histone methyltransferase SET-32, a co-factor of HRDE-1 in *C. elegans* is necessary to repress *slow-1*. This epigenetic repression results in decreased transcription and SLOW-1 levels that are insufficient to poison F₂ homozygous non-carrier progeny. The repressed state of *slow-1*(*) is transgenerationally inherited for over five generations.

5. Figure 3A outlines the crosses/transgenerational nature of the phenomenon, but skips some steps. It should fully lay out the genotypes from the P₀ to the 3rd (or 9th) generation, especially because as shown it is not clear if the gene was maintained as heterozygous or homozygous while self-fertilizing.

Thank you for pointing this out, we agree that important details of this complex cross were missing. We have now included the full mating scheme in Fig. 3a, highlighting all genotypes in each step.

In short, following the initial paternal inheritance of the TA, we isolated homozygous F₂ carriers for the repressed *slow-1/grow-1* allele (denoted with an asterisk), and these lines were propagated in the homozygous state via selfing.

- Fig. 3a in the original manuscript:

- Fig. 3a in revised manuscript:

Fig. 3 (highlight of updated panel). *slow-1* is transgenerationally silenced and targeted by the piRNA pathway. a, Mating scheme to test whether *slow-1/grow-1* repression is transgenerationally inherited

6. A clearer (graphical) representation of the experiments performed for figure 4d-f should be provided. These are among the core experiments of the manuscript and also the most complex to follow by the description in the text alone. The depiction of the crosses should be clarified and include when and what proteins and mRNAs are present to help the readers understand what molecules/factors are being assessed as potential couriers of epigenetic information.

We agree with the reviewer assessment. We have modified Fig. 4 panels d-f to further clarify the genotype of the crosses as well as to highlight the key differences with respect of what factors are present (RNA or protein) in each cross (see new Fig. 4a-c below). Similar changes have been applied to the rest of the figure (Fig. 4d-f)

- Fig. 4d-f in the original manuscript:

- Fig. 4a-c in the revised manuscript:

Fig. 4 (highlight of updated panels). Maternal *slow-1* transcripts inhibit piRNA-mediated repression.
a, In the *slow-1(-)/grow-1(-)* double mutant NIL strain, both *slow-1* and *grow-1* carry frameshift mutations (top). Reciprocal crosses between the NIL and *slow-1(-)/grow-1(-)* double mutants indicate that the TA is fully active both when maternally and paternally inherited (bottom, $n_M=31$, $n_P=24$, Fisher's exact test $p=0.43$). **b**, In the *slow-1(Δ)/grow-1(-)* mutant NIL strain, the full *slow-1* gene (including CDS and UTR) has been deleted and *grow-1* carries the frameshift mutation as in (A) (top). Reciprocal crosses between the NIL and *slow-1(Δ)/grow-1(-)* mutants indicate that *slow-1/grow-1* is only active when maternally inherited (bottom, $n_M=18$, $n_P=25$, Fisher's exact test $p<0.0001$). **c**, *In vitro* transcribed *slow-1* RNA with a mutated start codon was injected in the gonad of *slow-1(Δ)/grow-1(-)* double mutant NILs and later crossed to NIL males. Approximately half of their Δ/Δ F_2 progeny were delayed. Control mothers were injected with DEPC H_2O ($n_{\text{slow-1 RNA}}=128$, $n_{\text{ctrl}}=62$, Fisher's exact test $p<0.0001$).

7. The call-outs of Supplemental Figures in the test should detail which panel of the figure is being referenced, as it is not always clear and led to some confusion.

Thank you, all call-outs to Extended Data and Supplementary Figures now include the specific panels that are being referenced.

Minor concerns:

1. Clarification of genotypes of animals in Figure 2. (This is actually a general comment throughout the figures-spell everything out, try to not skip steps.)

We have updated Fig. 2 to include a scheme of the experimental crosses and genotypes of the individuals, as well as a more friendly visual guide to follow the experiments.

2. Mapping of the protein domains to the amino acid alignment in extended figure 2a (PRG-1) would be helpful (as was done for the SLOW proteins in extended 4d) to understand potential differences in the two PRG proteins and their different effects.

We requested alignment highlighting both the PAZ and PIWI domains of PRG-1.1 and PRG-1.2 are now available in Extended Data Fig. 4a.

3. It might be helpful to mention what kinds of proteins SLOW-1 and GROW-1 are and how SLOW-1 inhibits development.

These details are now mentioned in the introduction section (lines 102-104):

“SLOW-1 is homologous to nuclear hormone receptors (NHR) but lacks a DNA binding domain and has two transmembrane domains on its C terminus. The antidote, GROW-1, has no homology to known proteins.”

Unfortunately, how SLOW-1 inhibits development is unknown. In our previous publication, we speculated that SLOW-1 may act as dominant negative isoform, sequestering an unknown ligand to a membrane (Ben-David et al, 2021).

Typos and minor corrections:

-The embryos in Fig 3C appear to be oriented with posterior (P) cells to the left. Could the authors double check on this, and if it is the case, please rotate them so that the posterior is to the right.

The reviewer is correct. We have rotated the embryos so that the posterior side is located to the right side.

-Line 229: levered should be leveraged.

This typo has been fixed.

-Line 256: CSR-1b is technically constitutively expressed in the germline, not only the “female” (perhaps oogenic would be a better term here) germline.

Thank you. We agree with the reviewer’s comment. We changed “female” for “oogenic” (see line 345)

Referee #2 (Remarks to the Author):

The nonequivalence of maternal and paternal genomic imprinting is the inevitable consequence of conflictive selective forces acting on differentially expressed parental alleles, but how these epigenetic differences evolve in the first place is poorly understood. In this manuscript by Pliota et al, the authors identified parasitic conflict as an evolutionary force fueling the emergence of imprinting. First, they found that the *slow-1/grow-1* TA was specifically inactive when paternally inherited. Next, they showed that PIWI protein PRG-1.2 and two complementary piRNAs were required to repress the paternal inherited *slow-1/grow-1* allele. Intriguingly, they showed that maternal *slow-1* transcripts overrode the repressive effect of piRNAs on *slow-1* expression when *slow-1/grow-1* was maternally inherited, thereby resulting the toxin induced-developmental delay. Overall, this is an interesting story supported by comprehensive nematode genetic data, bringing a mechanistic insight into the parent-of-origin effect on TA expression. Most of the experiments are well designed, and many of the data presented are convincing. Nonetheless, the data are weaker for the involvement of piRNAs in controlling parent-specific gene expression. Following comments may help for further improvement.

We thank Reviewer #2 for underlining the comprehensiveness of our approach and mechanistic insights. In response to their comments, we have performed new genetic and biochemical experiments that specifically dissect and clarify the involvement of piRNAs in controlling parent-of-origin gene expression.

Major concerns:

1. The authors provided genetic evidence demonstrating that piRNA pathway represses paternal *slow-1* expression when maternal *slow-1* transcripts are absent in the eggs. More interestingly, this repressed state could be inherited for at least 3 generations and reverted after 9 generations. The authors found that the piRNAs paired with the 3'UTR of *slow-1* transcripts dominated the repression. How these piRNAs function to repress *slow-1* expression? Post-transcriptionally or epigenetically? Biochemical evidence should be provided to validate this regulation, given it as the key part of the working model.

We agree with the reviewer that the data in the original manuscript did not allow us to discriminate whether *slow-1* repression occurred at the post-transcriptional or transcriptional (epigenetic) level. To answer this question, we quantified *slow-1* pre-mRNA levels using RT-qPCR, a method that is standard in the field^{2,5}. As a positive control and to validate the RT-qPCR approach, we first designed and validated primers to amplify *slow-1* mRNA. In perfect agreement with our genetic crosses, mRNA-seq, and mScarlet::SLOW-1 reporter experiments (Fig. 1c,d and Fig. 2d,e), the RT-qPCR assay performed on a new set of NIL x EG6180 reciprocal crosses revealed a strong parent-of-origin effect: *slow-1* mRNA levels were significantly lower in F₁ mothers when *slow-1/grow-1* was paternally inherited (15.2% of the maternally inherited allele; $p < 0.0001$; Fig 3.h and see below).

Next, we designed and validated primers to amplify *slow-1* pre-mRNA, which included the fourth intron of the transcript. Analogously to the mRNA RT-qPCR, we observed that *slow-1* pre-mRNA levels were significantly lower following paternal inheritance of the TA (20.5% of the maternally inherited allele; $p < 0.001$; Fig. 3h). Importantly, we observed no amplicons in our RNA samples when using the pre-mRNA primer pairs in the absence of the RT enzyme (negative control), excluding the possibility of gDNA contamination.

Furthermore, we confirmed the correct pre-mRNA amplicon by Sanger sequencing, indicating that our assay is specific.

In addition to these results, we now also show that 22G-RNA and 26-RNA are highly upregulated when *slow-1* is repressed and that this effect is transgenerationally inherited (see response to Reviewer #1 point 1). And that the histone methyltransferase SET-32 is required for the repression of *slow-1* (see response to Reviewer #3 point 1). Overall, all these new results are consistent with the *C. elegans* piRNA literature and strongly indicate that *slow-1* is epigenetically repressed at the transcriptional level. These new experiment can be found in the main text lines 298-301.

Extended Data Fig. 3h (highlight of new panel). h, RT-qPCR quantification of *slow-1* abundance in F₁ young adults from reciprocal crosses between NIL and EG6180 normalized to the parental NIL. Specific primers were used to amplify mRNA (left) and pre-mRNA (right) transcripts.

2. PRG-1.2 and PRG-1.1 are highly identical Piwi proteins. What makes PRG-1.2, rather than PRG-1.1, critical for *slow-1* repression in the germline of the F₁. This should be further explored. For examples, whether the two proteins are equivalently expressed and similarly associate with Ctr-21ur-06949 and Ctr-21ur-06917?

We were also very intrigued by the fact that loss of PRG-1.2 but not PRG-1.1 could impair the repression of *slow-1*. For this revised version, we performed additional genetic and biochemical experiments that have clarified this specific point and provide further insights into PRG-1 function and how piRNAs regulate endogenous targets in nematodes.

Previously, we showed that deletion of two genetically linked piRNAs, *Ctr-21ur-06949* and *Ctr-21ur-06917*, was sufficient to impair *slow-1* repression following its paternal-inheritance (Fig. 3f). During the revision of this manuscript, we wondered whether the loss of just one of these two piRNAs would be sufficient to observe the same effect. To investigate this, we generated single piRNA mutants and performed paternal inheritance crosses (Fig. 3f). Strikingly, we observed that unlike the double piRNA mutant, the absence of individual piRNA resulted in efficient repression of the *slow-1/grow-1* TA. These experiments strongly suggest that the two piRNAs act epistatically to repress *slow-1* expression. To our knowledge, this is the first report of piRNAs repressing their target in a non-additive fashion.

Next, inspired by the epistatic nature of piRNA-mediated repression and the synthetic lethality of *prg-1.1* and *prg-1.2* double mutants (Fig. 3d), we hypothesized that any role that *prg-1.1* may play in *slow-1* repression could be masked by genetic redundancy (Fig. 3d

and below). To test this, we generated triple mutant worms carrying the *prg-1.1* mutant allele along with the double piRNA deletion and performed paternal inheritance crosses of the TA in this genetic background. In contrast to the partial de-repression of *slow-1/grow-1* observed in either *prg-1.2* or double piRNA mutants (Fig. 3f), de-repression of the TA was virtually complete when *prg-1.1* and the two piRNAs were mutated (Fig. 3g). **Thus, our crosses indicate that *prg-1.1* genetically interacts with *Ctr-21ur-06949* and *Ctr-21ur-06917*. As a control, no developmental defects were observed in the triple mutant parental line (0%, n=89). We conclude that both PRG-1.1 and PRG-1.2 mediate *slow-1* repression; however, the effect of PRG-1.1 is masked in the single *prg-1.1* mutant due to partial redundancy.**

Fig. 3. (highlight of relevant panel) *slow-1* is transgenerationally silenced and targeted by the piRNA pathway d, Effect of either *prg-1.1* or *prg-1.2* null mutations in *slow-1/grow-1* paternal inheritance. Crosses between *prg-1.1(-)* hermaphrodites and NIL *prg-1.1(-)* males result in mostly wild-type EG/EG F₂ individuals (n_{ctl}=53, n_{*prg-1.1(-)*}=16, Fisher's exact test p>0.99). However, crosses between *prg-1.2(-)* hermaphrodites and NIL *prg-1.2(-)* males result in delayed EG/EG F₂ individuals (n_{ctl}=53, n_{*prg-1.2(-)*}=75, Fisher's exact test p<0.0001).

Fig. 3. (highlight of updated panel) *slow-1* is transgenerationally silenced and targeted by the piRNA pathway f, Scheme showing the binding of piRNA candidates *Ctr-21ur-06949* and *Ctr-21ur-06917* into the 3'UTR of *slow-1*. (+) denotes a G:U wobble base pair (left). Paternal crosses between EG6180 hermaphrodites and NIL males. Both strains carry various combinations of piRNAs and *prg-1.1* mutations (right). (n_{ctl}=53, n_{*Ctr-21ur-06949(Δ); Ctr-21ur-06917(Δ)*}=40, n_{*Ctr-21ur-06949(Δ); Ctr-21ur-06917(Δ); prg-1.1(-)*}=34, Fisher's exact test, p<0.0001, p<0.0001 and p=0.0003).

To further explore the redundancy between PRG-1.1 and PRG-1.2 at the biochemical level, we immunoprecipitated 3xFLAG::PRG-1.1 and 3xFLAG::PRG-1.2 from gravid hermaphrodites and sequenced their associated sRNAs in biological triplicates (Extended Data Fig. 4a,b). As expected, sRNA bound by these two proteins were highly enriched in piRNAs compared with total sRNA library across replicates—80,4% (70-85%) vs. 9,2% (7-11%), respectively. Most piRNAs—including *Ctr-21ur-06949* and *Ctr-21ur-06917*—were bound by both PRG-1.1 and PRG-1.2, suggesting that their redundancy stems from

targeting a common set of transcripts (Extended Data Fig. 4b, Supplementary Data 2; see below). We also identified a subset of 787 (2,9%) piRNAs that exhibit differential binding between the two Argonautes after correcting for multiple testing (Supplementary Data 2). 72,6% of these piRNAs also showed differential downregulation in *prg-1.1* and *prg-1.2* mutants compared to WT, with a majority (87,9%) preferentially bound by PRG-1.1 (Supplementary Data 2).

Notably, these differentially bound piRNAs were not randomly distributed across Chr. IV but they clustered in three distinct genomic regions (3,9-4,8Mb, 5,2-6,2Mb; and 9,5-10,5Mb, Extended Data Fig. 4c, Supplementary Data 2; see below), which suggests shared genomic features among them and is consistent with the diversification of piRNAs via local duplication and modification of pre-existing sequences³¹. The two piRNAs known to repress *slow-1*, *Ctr-21ur-06949* and *Ctr-21ur-06917*, were not among the significantly bound piRNAs. However, we think this could be explained by the fact that the sRNA-seq method is not particularly well-suited to identify relatively modest changes in the abundance (as a proxy for binding affinity) of specific piRNAs. Also, while the overall genomic “clustering” of differentially bound piRNAs strongly suggest the presence of a motif or sequence signature that differentiates PRG-1.1 and PRG-1.2 bound piRNAs, we could not identify any strong or obvious sequence signature. While these results very exciting, we believe that a deeper molecular understanding of the relationship between these paralogs falls out of the scope of the present work and is better suited for a follow-up study.

Overall, our new genetic and biochemical findings indicate that PRG-1.1 and PRG-1.2 have mostly overlapping, but not entirely equivalent, piRNA binding preferences, likely contributing to their differential impact on *slow-1* repression and overall strong synthetic lethal interaction. These new experiments can be found in the main text lines 240-280.

Extended Data Fig. 4. (new figure). Characterization of the redundant role of PRG-1.1 and PRG1-2. **a**, Schematic of the sRNA immunoprecipitation (RIP) protocol. RIP was performed in biological triplicates on N-terminally FLAG-tagged PRG-1.1 and PRG-1.2 strains (endogenous locus) followed by sRNA-seq. **b**, Class enrichment among PRG-1.1 and PRG-1 bound sRNAs compared to a total sRNA-seq protocol. **c**,

Differences in abundance of piRNAs bound by each paralog. Normalized abundance (transcript per million) of PRG-1.1 (left) and PRG-1.2 (right) bound piRNAs (21U-RNAs) are plotted on the Y-axis and genomic coordinate of piRNA loci on the X-axis. Differentially bound piRNAs are shown with different thresholds of significance and are enriched within specific genomic clusters.

3. The authors proposed an intriguing model that maternally inherited *slow-1* mRNA itself, even at very low levels, might act as a “licensing” signal probably by a catalytic way to counteract piRNA-mediated repression in the zygote. How the maternally inherited *slow-1* mRNA inhibits piRNA repression? This should be experimentally explored.

While the identification of genes required for maternal licensing is challenging due to their likely essential role in germline development and fertility, we set out to further characterize the molecular requirements of the licensing signal, as well its mechanism of action.

Sequence similarity between *slow-1* maternal transcripts and their zygotic counterparts is likely key for the establishment of licensing. To explore whether this requirement is an intrinsic property of *slow-1* or a general feature, we asked whether homology to a foreign sequence could also license *slow-1*. To do so, we took advantage of the *mScarlet::slow-1* fusion strain (Fig 2a). Tagging of SLOW-1 with an N-terminal mScarlet reporter did not interfere with its toxicity and importantly, mScarlet::SLOW-1 only caused developmental delay among F₂ progeny when maternally inherited, indicating that the parent-of-origin effect was not affected by the introduction of this exogenous tag (Fig. 4d). Next, we generated a strain carrying a *rps-20p::mScarlet::rps-20* 3'UTR single copy transgene in the EG6180 background. In these worms, mScarlet is ubiquitously expressed in the soma and the germline. We then crossed hermaphrodites carrying this transgene to *mScarlet::slow-1* males and scored their F₂ progeny. Remarkably, maternal *mScarlet* transcripts fully licensed endogenous tagged *mScarlet::slow-1* (Fig. 4d). These results indicate that sequence similarity to a foreign maternal transcript is sufficient for epigenetic licensing. Moreover, they suggest that the licensing signal could spread through the zygotic transcript, as maternal *mScarlet* countered piRNAs targeting *slow-1* despite the lack of sequence similarity between the two genes.

Then, we set out to investigate at which step of the piRNA pathway did licensing counter repression: target recognition or transcriptional silencing. Target recognition depends on complementarity to the mature mRNA, whereas silencing relies on complementarity to the nascent transcript, which guides the repression machinery to the target locus (Fig. 4e). We reasoned that we could distinguish between these possibilities by testing whether maternal *mScarlet* could license *slow-1* in the context of a polycistronic operon⁶. In *C. elegans*, >17% of all genes are part of operons^{7,8}. Genes within an operon are transcribed from a single promoter as a contiguous pre-mRNA molecule. Following transcription, the operon pre-mRNA undergoes cis-splicing to remove introns but also trans-splicing by a SL2 leader, which results in the generation of independent mRNA molecules for each of the genes that make up the operon. If licensing occurs by counteracting piRNA recognition (mature mRNA), then homology to one of the genes in the operon should not license a second gene for which there is no homology. On the other hand, if licensing counteracts transcriptional silencing (pre-mRNA level) then homology to one gene should license a second gene in the operon despite not sharing homology with the maternal transcript.

To test this, we inserted the 256bp intergenic region from the *C. tropicalis gpd-2::gdp-3* operon in between *mScarlet* and *slow-1* using CRISPR/Cas9. This intergenic sequence (hereafter, “SL2”) contains the 3' acceptor site for the SL2 RNA trans-splicing leader⁹. The resulting operon, *mScarlet::SL2::slow-1*, is under the control of the native *slow-1* promoter—*mScarlet* and *slow-1* are transcribed as a single polycistronic pre-mRNA in the

germline and later trans-spliced into two independent mRNAs (Fig. 4f). As expected, we detected *mScarlet* exclusively in the germline of these worms (Extended Data Fig. 7d).

To validate our approach, we first performed reciprocal crosses between *mScarlet::SL2::slow-1* worms and the EG6180 parental strain and found that *slow-1* was active only when maternally inherited, indicating that the operon architecture did not interfere with the parent-of-origin effect (Fig. 4g). Furthermore, lack of maternal *slow-1* led to co-repression of *mScarlet* when the operon was paternally inherited, in agreement with silencing being guided by the nascent transcript (Extended Data Fig. 7e). We then crossed hermaphrodites expressing maternal *mScarlet* mRNA to males carrying the *mScarlet::SL2::slow-1* operon and scored their F₂ progeny. We observed no delayed EG/EG F₂ individuals, indicating that homology to *mScarlet* was not sufficient to license *slow-1*, despite being part of the same pre-mRNA molecule. Given that maternal *mScarlet* mRNA efficiently licensed *slow-1* when both genes were part of a monocistronic transcript, our results indicate that zygotic *slow-1* is licensed post-transcriptionally following splicing, likely during target recognition.

Overall, our new experiments further support the view that maternal transcripts do not counter piRNA-mediated repression simply through a passive “sponge-like” mechanism that sequesters piRNAs. Instead, licensing likely involves a catalytic step, as indicated by the following evidence: 1) A maternal transcript, even when it lacks complementarity to *slow-1*-targeting piRNAs, can effectively license its zygotic *slow-1* counterpart. This suggests that the licensing signal propagates through the transcript to inhibit piRNA-mediated repression, and 2) Licensing remains unaffected even in cases of a more than 100-fold reduction in maternal transcript levels. Lastly, our operon experiments strongly suggest that licensing counters piRNAs by either obstructing binding to their targets or impeding the subsequent amplification of 22G-RNAs within the perinuclear nuage. These new experiments can be found in the main text lines 434-481.

Fig. 4. (highlight of new panels) d, Reciprocal crosses between worms carrying an N-terminally tagged *mScarlet::slow-1* and EG6180 (top and middle crosses). *mScarlet::slow-1/grow-1* is active only when maternally inherited. Expression of *mScarlet* in the germline of the mother is sufficient to license a paternally inherited *mScarlet::slow-1* (bottom cross) ($n_{\text{maternal } mScarlet::slow-1}=23$, $n_{\text{paternal } mScarlet::slow-1}=31$, $n_{\text{maternal } mScarlet}$, $n_{\text{paternal } mScarlet::slow-1}=38$, Fisher’s exact test $p<0.0001$ and $p<0.0001$). **e**, Schematic of the piRNA pathway in worms. Target recognition and secondary sRNA amplification depend on the target mRNA, whereas epigenetic repression depends on complementarity to the nascent transcript of the target. **f**, Schematic of the *mScarlet::SL2::slow-1* operon. The operon is transcribed as a single polycistronic transcript and later trans-spliced into two independent mRNA transcripts. Introns and cis-splicing events are not shown for simplicity. Licensing could counter piRNA-mediated repression either during target recognition (mRNA) or epigenetic repression (nascent transcript) **g**, Reciprocal crosses between worms carrying the *mScarlet::SL2::slow-1* operon and EG6180 (top and middle crosses). The *mScarlet::SL2::slow-1* operon is active only when maternally inherited. Expression of *mScarlet* in the germline of the mother

does not license a paternally inherited *mScarlet::slow-1* (bottom cross) ($n_{\text{maternal } m\text{Scarlet}::\text{SL2}::\text{slow-1}}=32$, $n_{\text{paternal } m\text{Scarlet}::\text{SL2}::\text{slow-1}}=49$, $n_{\text{maternal } m\text{Scarlet}}$, $n_{\text{paternal } m\text{Scarlet}::\text{SL2}::\text{slow-1}}=37$, Fisher's exact test $p<0.0001$ and $p<0.0001$).

4. Fig. 1d shows that *slow-1* mRNAs were still present in F1 mothers when *slow-1/grow-1* was paternally inherited, despite significantly lower than maternally inherited. However, the authors argue that very low levels of *slow-1* mRNA could inhibit piRNA repression. How can the authors reconcile these data?

This is an excellent point. We believe that these two observations can be reconciled if we take into consideration different requirements for the initiation and maintenance of piRNA-mediated epigenetic repression. We have indeed shown that very low levels of maternal *slow-1* transcript are sufficient to inhibit the piRNA-mediated repression of a paternally inherited *slow-1* allele. On the other hand, based on mRNA-seq and RT-qPCR, we know that *slow-1* transcripts from the repressed paternal allele are low but still detectable in the F₁ progeny and in principle, they could license zygotic *slow-1* in the F₂ or subsequent generations. However, in such paternal-inheritance crosses, failure to license could arise from the inability of maternal transcripts from a repressed allele to license and/or the inability of an already repressed allele to be licensed.

To differentiate between these two possibilities, we devised a new crossing scheme that would allow us to test whether an epigenetically repressed *slow-1* allele could license a “naïve” zygotic *slow-1* allele (i.e. an allele that was not paternally inherited in the previous generation). To do so, we took advantage of *slow-1(fs)*, an allele that is not toxic but retains licensing activity (Fig. 4a). The complete mating scheme is depicted in Extended Data Fig. 7f (also see below). Briefly, we first crossed EG6180 hermaphrodites to *slow-1(fs)/grow-1(-)* males and obtained F₂ hermaphrodites that were homozygous for the double mutant allele. Then, we mated these hermaphrodites to males carrying the *slow-1/grow-1* wild-type TA and inspected their granddaughters. If the epigenetically repressed *slow-1(fs)** allele retains maternal licensing activity, then their homozygous double mutant progeny (F₅) should be delayed. Remarkably, we found no delayed worms among their progeny and all homozygous worms for the null TA allele were wild-type. This result strongly suggests that despite being expressed in the germline, maternal transcripts derived from an epigenetically repressed locus are incapable of licensing a naïve zygotic allele.

Extended Data Fig. 7. (highlight of new panel) f, Schematic of a cross designed to test whether an epigenetically repressed *slow-1* allele can license a naïve zygotic one. To this end, we took advantage of the *slow-1(fs)/grow-1(-)* double mutant line that carries a *slow-1* frameshift mutation. This mutation renders the *slow-1* toxin inactive, but it doesn't affect the ability of its maternal mRNA to license a paternally inherited *slow-1* copy (Fig. 4a). We first crossed EG6180 hermaphrodites to *slow-1(fs) grow-1 (-)* males and collected F₂ homozygous carriers of the repressed *slow-1(fs)* allele. We let these hermaphrodites self-fertilize for one generation, collected F₃ hermaphrodites and crossed them to NIL males carrying the wild-type TA. Finally, we phenotyped and genotyped their granddaughters (F₅ generation). If the wild-type TA were active, i.e. licensed, then progeny homozygous for the *slow-1(fs)/grow-1(-)* allele should be developmentally delayed. We found no significant delay among the progeny, and all homozygous mutants were wild-type indicating that licensing had been compromised.

While the molecular basis of the loss of licensing activity is unknown, this result is consistent with our latest observations. Our operon experiments strongly suggest that licensing counters piRNA-mediated repression during the target recognition or secondary sRNA amplification stages (see Reviewer #2 point 3 and Fig. 4d-g). If 22G-RNAs complementary to *slow-1* (see Reviewer #1 point 1 and Fig. 3i-j) are loaded into the eggs of *slow-1(fs) mothers alongside maternal transcripts, then these sRNAs could by-pass the licensing mechanism by directly recruiting HDRE-1 and SET-32 resulting in transcriptional repression. In support of this model 22G-RNAs are known to be maternally provisioned¹⁰. In summary, we have reconciled these contrasting observations by showing that repression of a *slow-1* allele following its paternal inheritance is sufficient to disrupt its licensing activity. This result can be found in the main text line 477-481.**

Minor points

1. The specific panel should be referred when the Extended Data were mentioned in the text.

All references to Extended Data Figures include now the specific panels.

2. In Fig 5c, the labeling of hermaphrodites should be "*slow-2(-); grow-2(-)*" rather than "*slow-2(-); grow(-)*".

Thank you, this typo has been corrected.

Referee #3 (Remarks to the Author):

In this solid, well-documented study, Alejandro Burga and his colleagues explored an unusual 'toxin-antidote' (TA) element in *C. elegans*, called *slow-1/grow-1*. They describe a novel mechanism that in special cases of heterozygosity (obtained by outcrossing) leads to a parent-of-origin dependent repression of the developmentally toxic *slow-1* gene in the TA. Although the underlying mechanisms are still only partially understood, this novel finding provides a first example of a parent-of-origin dependent effect on an endogenous gene in (isolates of) a *Caenorhabditis* species (i.e., *C. tropicalis*).

Specifically, they find that the *slow-1/grow-1* TA element becomes inactive when paternally inherited (in F2 animals), and that this involves repression in the germline of heterozygous mothers. Interestingly, they demonstrate that piRNA loci and Piwi Argonaute activity are required for this repressive process.

Similarly as had been reported before for transgenes in *C. elegans*, once achieved, the repressed state of *slow-1* is trans-generationally inherited across many generations. Importantly, the *slow-1* repression is not observed in case there is loading of *slow-1* transcripts into the oocyte (which explains that the effect on the paternally inherited *slow-1* is seen only in heterozygotes) and the authors show that these transcripts specifically inhibit the piRNA mediated repression.

This interesting study provides important novel insights into the rapid evolution of parasitic elements in worms through genetic conflict and into the emergence of a mechanism that bears similarities (in the F2 animals) with parental genomic imprinting in some insect species and mammals. The finding that the piRNA pathway is involved in the process is very interesting. In mice, for instance, at only one (retrotransposon-derived) imprinted gene locus, the establishment of DNA methylation-linked allelic gene repression is controlled by piRNAs, suggesting that this could be an ancestral imprinting mechanism.

Nevertheless, at the molecular level many key questions remain, particularly as concerns the trans-generational nature of the piRNA induced *slow-1* repression. It is somewhat unfortunate that this study includes mostly gene expression studies, and did not explore what happens at the chromatin level at the studied TA element. Without chromatin-level insights, it is difficult to know what might be the epigenetic feature that confers the long-term, transgenerational inheritance of the repressed *slow-1* state.

We express our gratitude to Reviewer #3 for emphasizing several novel aspects of our study. These include being the first to report a parent-of-origin effect in nematodes and providing insights into how genomic conflict can lead to the emergence of imprinting-like mechanisms. The involvement of the piRNA pathway in this phenomenon draws parallels with other model systems, such as mice, illustrating its broader implications. In this revision, we have conducted additional genetic, molecular, and biochemical experiments, significantly enhancing our mechanistic understanding of both *slow-1* repression and its licensing mechanism.

Main points:

1. What underlies the transgenerational inheritance of the *slow-1* silenced state? Is this linked to acquisition of heterochromatin-like features, similarly as has been reported for trans-generational inheritance of piRNA mediated transgene repression in *C. elegans*. The authors should explore this

important outstanding question by performing chromatin immunoprecipitation experiments. Importantly, such studies could provide insights into the nature of the epigenetic 'mark' that confers the trans-generational inheritance.

To further characterize the repressed state of *slow-1*, we undertook four orthogonal experimental approaches: i) sequencing of sRNA populations in *slow-1* active and repressed states, ii) quantification of *slow-1* mRNA and pre-mRNA transcripts in active and repressed states, iii) chromatin immunoprecipitation of the H3K9me3 histone mark in the *slow-1* active and repressed states, and iv) a reverse genetic approach to identify components of the piRNA pathway that are required for *slow-1* repression.

The first two points were already covered in response to Reviewers #1 and #2. In brief, our new experiments revealed that i) 22G-RNAs complementary to *slow-1* are >30-fold upregulated in the *slow-1* repressed state and these sRNAs can be transgenerationally inherited (see Reviewer #1 point 1; Fig. 3i-j) , ii) that *slow-1* mRNA and pre-mRNA levels are downregulated in the *slow-1* repressed state (see Reviewer #2 point 2; Fig. 3h).

As requested by the reviewer, we also performed chromatin immunoprecipitation (ChIP) essays to study whether *slow-1* repression led to the acquisition of heterochromatin-like features. Based on previous work in *C. elegans* mainly performed in the context of heritable silencing of endogenous genes via RNAi and most recently, silencing via synthetic piRNA arrays, we reasoned that the *slow-1* locus could acquire H3K9me3 repressive marks^{11,12}. While *C. tropicalis* chromatin is arguably very similar to *C. elegans*, we first optimized conditions of chromatin purification and fragmentation in this nematode, as nobody had performed ChIP in this species before.

The input requirements for ChIP, made it inviable for us collect enough material from either F₁, or F₂ or F₃ individuals following paternal inheritance of the TA. Unlike RNAi treatments, *slow-1* repression does not take place after exposure to an external trigger (dsRNA) but by outcrossing, which cannot be easily scaled up. Furthermore, unlike RNAi treatment that can lead to stable silencing for hundreds of generations, *slow-1* repression is reversible. Despite these limitations, however, we could obtain enough material from F₄ homozygous carriers of a repressed *slow-1* allele. Importantly, we detected a >30-fold upregulation of repressive 22G-RNA complementary to *slow-1* in equivalent F₄ populations following initial paternal inheritance of the TA, suggesting that differences in chromatin may, in principle, still be detectable in the F₄ generation (see Fig. 3i,j).

We performed H3K9me3 ChIP-seq in biological triplicates in F₄ individuals following initial paternal inheritance of the TA (*slow-1* repressed state) and the NIL parental control (*slow-1* active state) (See below). Because this is the first time that ChIP-seq was performed in *C. tropicalis*, we first validated our implementation of the protocol originally developed by the labs of Sam Gu and Andrew Fire. QC clustering and enrichment plots indicated that all ChIP samples showed sequence enrichments compared to chromatin input controls. In *C. elegans*, it is well established that H3K9me3 is highly enriched in the chromosomal arms compared to the central region in all six chromosomes (the trend is weaker in Chr. X)^{13,14}. To test whether this is also the case in *C. tropicalis*, we called H3K9me3 peaks using the MACS2 algorithm and found that all samples showed the same chromosomal signature previously reported in *C. elegans* (see below, only two samples shown for simplicity). However, neither MACS2 nor visual inspection of aligned ChIP-seq reads revealed any consistent H3K9me3 peak in the *slow-1* locus in any of the two conditions (see Fig. 3k, left). Clear and reproducible peaks were observed in other genomic loci (see for example, Fig. 3k, right) suggesting that, in principle, our experiment could identify H3K9me3 regions. This new result can be found in the main text lines 320-327.

It is worth noting that even though H3K9me3 is by far the best studied repressive histone mark in *C. elegans*, many recent genetic studies indicate that SET-25 and SET-32 activity is required for the establishment of epigenetic silencing but dispensable for its maintenance (inheritance). Thus, our results suggest that H3K9me3 may not be involved in the maintenance of *slow-1* repression, but we cannot rule out a role in the initiation (unfortunately, this question cannot be currently addressed due to technical limitations; we cannot retrieve enough chromatin from individuals from previous generations).

Revision-only figure including technical details of the new H3K9me3 ChIP-seq experiment. a, Schematic of the genetic crosses and steps necessary to obtain a F4 population of worms with an inactive TA (same scheme as the sRNA experiment). **b**, H3K9me3 ChIP-seq was performed in biological triplicates (independent genetic crosses) in the parental NIL line (TA is active) and F4 population following paternal inheritance of the TA (TA is inactive). These two groups are genetically identical (top). Clustering of ChIP and chromatin input samples (bottom) **c**, Enrichment plots for all ChIP and Input samples. **d**, Genome-wide localization of H3K9me3 peaks in *C. tropicalis* (Chr. I, II, III, IV, V, and X). A clear enrichment of H3K9me3 signal is evident in all chromosomal arms, as it has been previously shown in *C. elegans*.

Fig. 3 (highlight of new panel) k, H3K9me3 ChIP-seq was performed in biological triplicates on samples where *slow-1* is licensed (NIL control line; green) or repressed (F4 individuals following its paternal inheritance; red). Lines correspond to the ratio of H3K9me3 ChIP coverage over chromatin input coverage each normalized by their respective library sizes. No apparent peaks on *slow-1* locus (left) and example of reproducible peaks identified by MACS2 (right)

We acknowledge that CHIP-seq has limitations, especially concerning its sensitivity. From the very limited number of examples in the literature, the relative fold enrichment of H3K9me3 in loci targeted by sRNAs is relatively modest. To the best of our knowledge, the most significant effect observed was approximately a 5-fold enrichment, which occurred in the case of the highly potent RNAi response targeted by exogenous dsRNA¹¹. In contrast, a much more modest effect of around 2-fold enrichment was observed in the case of piRNA-mediated silencing of GFP in the germline of worms². One important consideration about the epigenetic repression of *slow-1* is that it is quantitative in nature. Both RNA-seq and qPCR indicate that ~20% of *slow-1* mRNA is still present following paternal inheritance of the TA. This partial repression of *slow-1* (only expressed in germline), along with the modest levels of H3K9me3 fold-enrichment observed in the literature, may make it challenging to identify the histone marks mediating this effect. Although it took us several months to implement the CHIP-seq protocol in our lab, we are now excited to have this method in our toolkit. We look forward to continuing our investigation of this challenging question in the future, for example, by studying additional histone marks and adopting more sensitive and targeted approaches to study chromatin in worms.

In addition to the molecular and biochemical approaches, we also took a forward genetic approach to explore the contribution of epigenetic marks to the *slow-1* parent-of-origin effect. In *C. elegans*, piRNAs trigger the production of secondary 22G-RNAs that are complementary to the target mRNA. These 22G-RNAs are then bound by nuclear Argonautes HRDE-1 and WAGO-10, which in turn recruit chromatin-modifying enzymes to the target locus and mediate its epigenetic repression (Extended Data Fig. 5a). Two histone methyltransferases, SET-25 (H3K9me3) and SET-32 (H3K23me3), play a crucial role in this process^{15,16}. To test whether effectors of the piRNA pathway (in addition to PRG-1.1 and PRG-1.2) mediate *slow-1* repression, we generated seven new knockouts of known piRNA factors using CRISPR/Cas9. These included *C. tropicalis* *mut-16*, *simr-1*, *rrf-1*, *hrde-1*, *wago-11*, *set-25*, and *set-32*. Three out these seven mutants were 100% infertile (*mut-16*, *simr-1*, and *hrde-1*) preventing further characterization. Next, we set up paternal crosses using the remaining four viable mutants and found that loss of *Ctr-set-32* was sufficient to impair *slow-1/grow-1* repression when inherited through the paternal-lineage, mimicking loss of *prg-1.2* and piRNAs (Fig 3g, Extended Data Fig. 5e; see below). The involvement of a histone methyltransferase in the parent-of-origin effect strongly suggests that *slow-1* repression occurs at the transcriptional level. Importantly, our experiments do not necessarily discard a role of SET-25 in *slow-1* repression, but we think its role is likely masked by genetic redundancy, as we detected five additional paralogs of SET-25 in the genome of *C. tropicalis* compared to the one copy found in *C. elegans*.

Extended Data Fig. 3a (highlight of new panel). a, Schematic of the piRNA pathway in *C. elegans* (right). *C. tropicalis* mutants generated in this study, their fertility status, and evolutionary relationship to *C. elegans* proteins (left).

Extended Data Fig. 3g. (highlight of new panel). f, Paternal crosses between EG6180 hermaphrodites and NIL males. Both strains carry mutations in homologs to genes required for piRNA-mediated repression in *C. elegans*. Loss of *set-32* phenocopies *prg-1.2* and double piRNA deletion mutants ($n_{\text{ctl}}=53$, $n_{\text{set-32}}=23$, Fisher's exact test $p<0.0001$)

Extended Data Fig. 3e (highlight of new panel) AlphaFold2 predictions of *C. elegans* SET-32 and *C. tropicalis* SET-32 proteins and their structural alignment (PYMOL). Both proteins are highly similar (RMSD=0.962).

In summary, our new experiments revealed that: 1) 22G-RNA and 26G-RNAs complementary to *slow-1* are massively upregulated in the *slow-1* repressed state and these sRNAs can be transgenerationally inherited (see Reviewer #1 point 1), 2) that *slow-1* mRNA and pre-mRNA levels are downregulated in the repressed state (see Reviewer #2 point 2), and 3) that *slow-1* repression depends on SET-32, a histone methyltransferase (H3K23me3), which has been shown to be necessary for piRNA mediated repression in *C. elegans*. While our ChIP essays did not revealed a signature of H3K9me3 heterochromatin formation in the *slow-1* locus, collectively, our past and new experiments strongly suggests that paternal inheritance of the TA leads to epigenetic repression of *slow-1* at the transcriptional level, and based on our current understanding of the piRNA pathway in *C. elegans*, our findings are consistent with 22G-RNA-mediated repression at the target locus by nuclear Argonautes and their associated co-factors, such as the histone methyltransferase SET-32.

2. The authors convincingly show that the piRNA pathway Prg-1.2 protein is essential for the establishment of the silencing of the paternally inherited *slow-1* gene (in their specific heterozygous background). Throughout subsequent generations, is this protein in the oocyte/zygote also important for the observed trans-generational inheritance of the repressed gene state? Once established, can the repressed state of *slow-1* be trans-generationally inherited without the expression of Prg-1.2?

Thank you for suggesting this insightful experiment. To answer this question, we performed a new set of crosses in which maternal PRG1.2 was still present in the F₁ generation (initiation stage) but was subsequently lost from the F₂ generation onwards (maintenance stage). The specific details of the mating scheme are shown below and in Extended Fig. 3f (see below). Following this mating, *prg-1.2* is maternally provided to the F₁ (previously shown to be sufficient for *slow-1* repression), and homozygous F₂ *prg-1.2* (-/-) worms are identified and propagated by selfing f/ We found that PRG-1.2 was dispensable for the maintenance and epigenetic inheritance of the *slow-1* repressive state, as the TA was not active in *prg-1.2* (-/-) worms three generations following the initial paternal cross (see Extended Data Fig. 3f and Fig. 3a-b). Thus, our results thus strongly suggest that maternal PRG-1.2 is only necessary for the initiation of *slow-1* repression. This is in agreement with previous work in *C. elegans* where PRG-1 is only required for the initiation but not maintenance of piRNA-mediated silencing of transgenic reporters^{2,15,17}.

Extended Data Fig. 3f. (highlight of new panel). f, PRG-1.2 is not necessary for the maintenance of *slow-1* repression. In this crossing scheme mothers provide PRG-1.2 to their F₁ progeny, which is sufficient for piRNA-mediated repression, and the *slow-1/grow-1* is paternally inherited. Since EG6180 hermaphrodites do not provide maternal *slow-1* transcripts, then the *slow-1* paternal allele is epigenetically repressed in the F₁. Then F₂ progeny are singled, and their offspring genotyped for both the TA and the *prg-1.2* locus. Hermaphrodites that are homozygous carriers for both the TA and the *prg-1.2* (-/-) null allele are identified (6.25% the F₂ progeny) and propagated for 1 or 7 generations. Then these hermaphrodites are crossed to EG6180 *prg-1.2* (-/-) males to test whether the TA is active. Inset shows the observed activity of the TA measured as percentage of delayed (EG/EG) individuals.

3. It is not clear from the presented data how precisely the expression of the slow-1 mRNAs inhibits the piRNA-mediated repression of the paternally-inherited TA element? Could the authors provide further insights into this mechanism, or explain this aspect better in their text?

This important point was already raised by Reviewer#2, thus we kindly direct the reviewer to our previous response (see Reviewer #2 point 3), which discusses in depth new experiments that we performed to further clarified the mechanism of licensing. In summary our latest experiments indicate that maternal transcripts inhibit piRNA-repression post-transcriptionally, either by obstructing binding to their targets or impeding the subsequent amplification of 22G-RNAs. Also, our new data shows that partial homology to a completely exogenous sequence is sufficient to license *slow-1*. This is an important finding since it indicates that maternal transcripts do not rely on a sponge-like mechanism to counter piRNAs but that the licensing signal can spread throughout the transcript. Our final model has been updated and incorporates these new findings (see Fig. 6)

-The title seems misleading. Particularly, the term 'parent-specific' is very confusing. The repression pattern observed here is not at all parent-specific (specific to the parent). Rather, it is parent-of-origin dependent.

The reviewer is correct, we agree this term could generate confusion. We have changed the title to "*Genetic conflict drives the evolution of parent-of-origin gene expression.*"

Minor point:

- In Figure 1d, what explains the outlier that shows a high level of slow-1 mRNA expression upon paternal inheritance (NIL)? Could the authors comment on this in the text?

We examined the RNA-seq library preparation of this outlier in close detail (from the quality control of RNA samples to sequencing depth and percentage of PCR duplicates), and we could not detect any obvious difference between the outlier and the other libraries. However, since each library was derived from an independent genetic cross (biological replicate), we cannot discard a human error while performing the cross or subsequent steps. In the absence of any obvious discriminating factor, we decided that it would be best practice to keep the outlier in the final analysis. Given that three independent and complementary methods (mRNA-seq, RT-qPCR, and an endogenous SLOW-1 protein reporter) all show the same parent-of-origin effect, we are very confident about our overall conclusion.

We have now added a clarifying sentence to the methods section to specifically address the NIL outlier (see methods line 1128-1134)

References

1. Shen, E.-Z., Chen, H., Ozturk, A.R., Tu, S., Shirayama, M., Tang, W., Ding, Y.-H., Dai, S.-Y., Weng, Z., and Mello, C.C. (2018). Identification of piRNA Binding Sites Reveals the Argonaute Regulatory Landscape of the *C. elegans* Germline. *Cell* 172, 937-951.e18. 10.1016/j.cell.2018.02.002.
2. Shirayama, M., Seth, M., Lee, H.-C., Gu, W., Ishidate, T., Conte, D., and Mello, C.C. (2012). piRNAs Initiate an Epigenetic Memory of Nonself RNA in the *C. elegans* Germline. *Cell* 150, 65-77. 10.1016/j.cell.2012.06.015.

3. Seroussi, U., Lugowski, A., Wadi, L., Lao, R.X., Willis, A.R., Zhao, W., Sundby, A.E., Charlesworth, A.G., Reinke, A.W., and Claycomb, J.M. (2023). A comprehensive survey of *C. elegans* argonaute proteins reveals organism-wide gene regulatory networks and functions. *eLife* *12*, e83853. 10.7554/eLife.83853.
4. Ben-David, E., Pliota, P., Widen, S.A., Koreshova, A., Lemus-Vergara, T., Verpukhovskiy, P., Mandali, S., Braendle, C., Burga, A., and Kruglyak, L. (2021). Ubiquitous Selfish Toxin-Antidote Elements in *Caenorhabditis* Species. *Current Biology* *31*, 990-1001.e5. 10.1016/j.cub.2020.12.013.
5. Buckley, B.A., Burkhart, K.B., Gu, S.G., Spracklin, G., Kershner, A., Fritz, H., Kimble, J., Fire, A., and Kennedy, S. (2012). A nuclear Argonaute promotes multigenerational epigenetic inheritance and germline immortality. *Nature* *489*, 447-451. 10.1038/nature11352.
6. Blumenthal, T., and Gleason, K.S. (2003). *Caenorhabditis elegans* operons: form and function. *Nat Rev Genet* *4*, 110-118. 10.1038/nrg995.
7. Allen, M.A., Hillier, L.W., Waterston, R.H., and Blumenthal, T. (2011). A global analysis of *C. elegans* trans-splicing. *Genome Res.* *21*, 255-264. 10.1101/gr.113811.110.
8. Blumenthal, T., Evans, D., Link, C.D., Guffanti, A., Lawson, D., Thierry-Mieg, J., Thierry-Mieg, D., Chiu, W.L., Duke, K., Kiraly, M., et al. (2002). A global analysis of *Caenorhabditis elegans* operons. *Nature* *417*, 851-854. 10.1038/nature00831.
9. Spieth, J., Brooke, G., Kuersten, S., Lea, K., and Blumenthal, T. (1993). Operons in *C. elegans*: Polycistronic mRNA precursors are processed by trans-splicing of SL2 to downstream coding regions. *Cell* *73*, 521-532. 10.1016/0092-8674(93)90139-H.
10. Gu, W., Shirayama, M., Conte, D., Vasale, J., Batista, P.J., Claycomb, J.M., Moresco, J.J., Youngman, E.M., Keys, J., Stoltz, M.J., et al. (2009). Distinct Argonaute-Mediated 22G-RNA Pathways Direct Genome Surveillance in the *C. elegans* Germline. *Molecular Cell* *36*, 231-244. 10.1016/j.molcel.2009.09.020.
11. Gu, S.G., Pak, J., Guang, S., Maniar, J.M., Kennedy, S., and Fire, A. (2012). Amplification of siRNA in *Caenorhabditis elegans* generates a transgenerational sequence-targeted histone H3 lysine 9 methylation footprint. *Nat Genet* *44*, 157-164. 10.1038/ng.1039.
12. Priyadarshini, M., Ni, J.Z., Vargas-Velazquez, A.M., Gu, S.G., and Frøkjær-Jensen, C. (2022). Reprogramming the piRNA pathway for multiplexed and transgenerational gene silencing in *C. elegans*. *Nat Methods* *19*, 187-194. 10.1038/s41592-021-01369-z.
13. Gu, S.G., and Fire, A. (2010). Partitioning the *C. elegans* genome by nucleosome modification, occupancy, and positioning. *Chromosoma* *119*, 73-87. 10.1007/s00412-009-0235-3.
14. Liu, T., Rechtsteiner, A., Egelhofer, T.A., Vielle, A., Latorre, I., Cheung, M.-S., Ercan, S., Ikegami, K., Jensen, M., Kolasinska-Zwierz, P., et al. (2011). Broad chromosomal domains of histone modification patterns in *C. elegans*. *Genome Res* *21*, 227-236. 10.1101/gr.115519.110.
15. Ashe, A., Sapetschnig, A., Weick, E.-M., Mitchell, J., Bagijn, M.P., Cording, A.C., Doebley, A.-L., Goldstein, L.D., Lehrbach, N.J., Le Pen, J., et al. (2012). piRNAs Can Trigger a Multigenerational Epigenetic Memory in the Germline of *C. elegans*. *Cell* *150*, 88-99. 10.1016/j.cell.2012.06.018.
16. Schwartz-Orbach, L., Zhang, C., Sidoli, S., Amin, R., Kaur, D., Zhebrun, A., Ni, J., and Gu, S.G. (2020). *Caenorhabditis elegans* nuclear RNAi factor SET-32 deposits the transgenerational histone modification, H3K23me3. *eLife* *9*, e54309. 10.7554/eLife.54309.
17. Lee, H.-C., Gu, W., Shirayama, M., Youngman, E., Conte, D., and Mello, C.C. (2012). *C. elegans* piRNAs Mediate the Genome-wide Surveillance of Germline Transcripts. *Cell* *150*, 78-87. 10.1016/j.cell.2012.06.016.

Reviewer Reports on the First Revision:

Referees' comments:

Referee #1 (Remarks to the Author):

I commend the authors on their extensive efforts to address many in depth reviewer comments and queries. They have exceeded my expectations in the additional data and explanation they have provided, and have answered my queries exceptionally well. My only comment now is regarding the naming of the newly identified AGOs, WAGO-11, -12, 13. This may lead to some confusion, as NRDE-3 is WAGO-12 in *C.e.*, and WAGO-11 is a pseudogene. I recognize that the naming convention makes sense, given the repertoire of AGOs in *C.trop.*, but wanted to point this out for the authors to at least consider. Congratulations on this exciting work!

Referee #2 (Remarks to the Author):

The authors have conducted an extensive range of new experiments to address my concerns and significantly improved the manuscript. In particular, the conclusion of piRNAs in controlling parent-of-origin gene expression is strengthened. Several of their findings are of great interest to the piRNA field.

Minor comments:

1. Line 290, replace "Extended Data Fig. 5a, b" with the correct call-out: "Fig 3g, Extended Data Fig. 5a-d".
2. Line 453, change "whereas silencing..." to "whereas transcriptional silencing..." for clarity and avoiding confusion.

Referee #3 (Remarks to the Author):

In their revised manuscript, the authors carefully addressed the questions that I raised, and to do so, performed additional experiments. They also addressed most of the points raised by the other reviewers.

Overall, the data improved a lot. The manuscript now also provides more explanation and better summary figures, and should therefore be more accessible to the reader.

Nevertheless, I still have a few suggestions for the authors to consider:

-In the Title (and the manuscript text), please talk about parent-of-origin DEPENDENT gene expression: 'Genetic conflict drives the evolution of parent-of-origin dependent gene expression'.

-A major finding in this study is that the silenced state of *slow-1/grow-1* TA that is acquired upon paternal transmission is maintained trans-generationally across multiple generations. The authors should mention this in their Abstract (.....to not give the impression that this is a mechanism that is reset each generation, as is genomic imprinting in mammals).

-The new ChIP-seq data presented in Figure 3k are interesting and seem indeed to suggest that H3K9me3 levels at *slow-1* are low and comparable between the repressed and the active state. Unfortunately, the authors did not include experiments with control antisera, to assess background precipitation levels (us of non-specific antiserum), or to show that the chromatin at the locus is accessible to ChIP in both the lines (e.g., with an antiserum against a core histone). Admittedly, because of the limited material availability such experiments are difficult to perform. The authors could try to obtain these additional data, or, alternatively, they should clearly stipulate the limitations of their preliminary data on H3K9me3 in the text.

-Lines 81-82: As concerns mammalian genomic imprinting it is wrong to say that ‘...a detailed understanding of the underlying molecular mechanisms is lacking’. In fact, this is one of the best understood epigenetic mechanisms in mammals, both for the (DNA methylation) imprint establishment in male and female germ cells, and for the somatic maintenance of imprints during development. Please, correct this sentence.

Robert Feil

Referee #4 (Remarks to the Author):

Pliota et al. present an in-depth characterization of the *slow-1/grow-1* toxin-antidote selfish genetic element in the nematode *C. tropicalis*, which unusually has a parent-of-origin effect. The authors have conducted extensive mechanistic and genetic experiments to document how it is dependent on the piRNA pathway, rather than methylation as understood in mammalian systems. The other reviewers thoroughly assessed these aspects of the study, and my view is that it is solid work and includes some clever approaches to distinguish alternative molecular hypotheses. The writing is clear given the complexity of explaining epigenetic phenomena. I was asked to focus on the evolutionary aspect.

Lines 520-523: It is unclear to me what aspects of the results presented provide evidence for these statements and conclusions. I noted no explicitly evolutionary or population genetic analyses up to this point in the manuscript that would speak to how strong selection acts on the TA elements, nor what form of selection.

Lines 546-548: In referring to the present mechanism as a “major force driving the evolution of imprinting”, I can’t help but wonder how much the selfing mode of reproduction in *C. tropicalis* constrains or facilitates this speculation relative to obligatorily outbreeding systems like the insect,

plant, and rodent organisms alluded to earlier in the sentence. Given that the ancestral reproductive state in the lineage leading to *C. tropicalis* also was obligatorily outbreeding, it also raises the question whether selfers are predisposed to revealing TA systems in interpopulation crosses following their evolutionarily resolution within different subpopulations, whether they might be more prevalent in outbreeding taxa, etc. I realize that deep consideration of some of these issues lies beyond the scope of the present study, but if the authors elect to touch on broader evolutionary issues then it seems important to consider the implications in more than an offhand manner.

Minor comments:

Sometimes a "." is used to designate decimal numbers whereas other times a "," is used, please edit for consistency.

Author Rebuttals to First Revision:

Response to reviewer comments – Second revision

- Referee #1 (Remarks to the Author):

I commend the authors on their extensive efforts to address many in depth reviewer comments and queries. They have exceeded my expectations in the additional data and explanation they have provided, and have answered my queries exceptionally well.

We thank Reviewer #1 for their valuable comments throughout the review process and their extremely positive evaluation of our revised manuscript.

My only comment now is regarding the naming of the newly identified AGOs, WAGO-11, -12, 13. This may lead to some confusion, as NRDE-3 is WAGO-12 in *C.e.*, and WAGO-11 is a pseudogene. I recognize that the naming convention makes sense, given the repertoire of AGOs in *C.trop.*, but wanted to point this out for the authors to at least consider. Congratulations on this exciting work!

To avoid confusion with *C. elegans* Argonautes and following the reviewer's advice, we have renamed two *C. tropicalis* Argonautes: *C.tr*-WAGO-11 is now *Ctr*-WAGO-15 and *C.tr*-WAGO-12 is now *Ctr*-WAGO-14. These changes have been applied to the main text, main figures, and extended figures. In addition, we have included the FASTA sequences of all *C. tropicalis* Argonautes in the Supplementary Information section.

- Referee #2 (Remarks to the Author):

The authors have conducted an extensive range of new experiments to address my concerns and significantly improved the manuscript. In particular, the conclusion of piRNAs in controlling parent-of-origin gene expression is strengthened. Several of their findings are of great interest to the piRNA field.

We thank Reviewer #2 for their advice, and we are glad to hear that they find that our conclusions have been strengthened and that they anticipate great interest by the piRNA community.

Minor comments:

1. Line 290, replace “Extended Data Fig. 5a, b” with the correct call-out: “Fig 3g, Extended Data Fig. 5a-d”.

Thank you. We have corrected the call-out.

2. Line 453, change “whereas silencing...” to “whereas transcriptional silencing...” for clarity and avoiding confusion.

We agree. We have changed the phrase as suggested by the reviewer.

- Referee #3 (Remarks to the Author):

In their revised manuscript, the authors carefully addressed the questions that I raised, and to do so, performed additional experiments. They also addressed most of the points raised by the other reviewers.

Overall, the data improved a lot. The manuscript now also provides more explanation and better summary figures, and should therefore be more accessible to the reader.

We thank Reviewer #3 for their insightful comments, and we are pleased to hear that we have addressed all their major points, as well as those from the other two reviewers.

Nevertheless, I still have a few suggestions for the authors to consider:

-In the Title (and the manuscript text), please talk about parent-of-origin DEPENDENT gene expression: ‘Genetic conflict drives the evolution of parent-of-origin dependent gene expression’.

This is a very good suggestion, but unfortunately, the proposed title is much longer than the journal’s very strict limit (75 characters). Further, based on comments by Reviewer #4 and the Editor, we have decided to change the title to:

“Selfish conflict underlies RNA-mediated parent-of-origin effects”

In the main text, we have changed all instances of “parent-of-origin gene expression” to “parent-of-origin dependent gene expression”. However, we generally refer to this phenomenon as a “parent-of-origin effect on gene expression”, which is also consistent with the literature.

-A major finding in this study is that the silenced state of *slow1/grow-1* TA that is acquired upon paternal transmission is maintained trans-generationally across multiple generations. The authors should mention this in their Abstract (.....to not give the impression that this is a mechanism that is reset each generation, as is genomic imprinting in mammals).

We agree with the reviewer, and we have added the following sentence to the abstract:

“This parent-of-origin effect stems from transcriptional repression of the *slow-1* toxin by the piRNA host defense pathway. The repression requires PIWI (Argonaute) and SET-32 histone methyltransferase activities and is transgenerationally inherited via small RNAs.”

-The new ChIP-seq data presented in Figure 3k are interesting and seem indeed to suggest that H3K9me3 levels at *slow-1* are low and comparable between the repressed and the active state. Unfortunately, the authors did not include experiments with control antisera, to assess background precipitation levels (us of non-specific antiserum), or to show that the chromatin at the locus is accessible to ChIP in both the lines (e.g., with an antiserum against a core histone). Admittedly, because of the limited material availability such experiments are difficult to perform. The authors could try to obtain these additional data, or, alternatively, they should clearly stipulate the limitations of their preliminary data on H3K9me3 in the text.

Our H3K9me3 ChIP-seq experiment included a “chromatin input” control. That is, we treated control samples like an IP (but did not include the antibody) and then prepared a DNA library just like the IP sample. This input library controls for biases generated during chromatin purification and sonication, as well as DNA library preparation (Park et al, *Nat. Rev. Gen.* 2009; Meyer et al, *Nat. Rev. Gen.* 2014) Further, these control libraries were used for calling peaks using the MACS2 software. The reviewer is correct that we did not include a second kind of control, the “non-specific antiserum” (or mock-IP), which tests whether the antibody binds non-specifically the locus of interest. As pointed out by the reviewer, this control is expected to yield very low levels of DNA and it would be particularly challenging to generate in the context of our experimental paradigm, given the difficulties to purify sufficient amounts of chromatin in the first place.

We would like to point out that “chromatin input” controls, as the one we performed, are very standard in the field. For example, the vast majority of modENCODE and ENCODE ChIP-seq experiments exclusively use “chromatin input” as controls (Gerstein, et al. *Science* 2011; The ENCODE Consortium, *Nature* 2012; Xu et al. *NAR* 2021). We acknowledge that our ChIP-seq experiments are suggestive and thus, as recommended by the reviewer, we have pointed out very clearly the limitations of our approach in the main text of the revised manuscript and hope to address some of these challenging technical aspects in a follow up study.

See line 255:

“To test this, we performed H3K9me3 ChIP-seq in F₄ individuals following paternal inheritance of the TA, as well as in the parental NIL control. We did not observe any significant H3K9me3 enrichment in slow-1 even though we detected H3K9me3 enrichment in other loci (Fig. 3k; Supplementary Discussion). While we cannot rule out potential limitations of our assay such as dilution of the germline signal or unspecific binding of the antibody, this result suggests that H3K9me3 may not be required for the maintenance of silencing, in line with recent findings^{40–42}.”

-Lines 81-82: As concerns mammalian genomic imprinting it is wrong to say that ‘....a detailed understanding of the underlying molecular mechanisms is lacking’. In fact, this is one of the best understood epigenetic mechanisms in mammals, both for the (DNA methylation) imprint establishment in male and female germ cells, and for the somatic maintenance of imprints during development. Please, correct this sentence.

Thank you for bringing this point to our attention. We absolutely agree with the reviewer that imprinting is one the best understood epigenetic phenomena at the molecular level. In our initial manuscript we intended to convey a different point. That is, that the evolutionary origins of imprinting are not well understood at the molecular level. However, we acknowledge that this was not clearly articulated. We have revised the sentence to better convey this perspective:

- Original sentence

“The discovery of the first imprinted loci in mammals led to the hypothesis that imprinting evolved from host defense mechanisms that use DNA methylation to keep viruses and parasitic genes at bay. [...] However, a detailed understanding of the underlying molecular mechanisms and evolutionary forces behind is lacking”

- Revised version

“The discovery of the first imprinted loci in mammals led to the hypothesis that imprinting evolved from host defense mechanisms that use DNA methylation to keep viruses and parasitic genes at bay. [...] However, the evolutionary origins of imprinting are still poorly understood at the molecular level”

- Referee #4 (Remarks to the Author):

Pliota et al. present an in-depth characterization of the slow-1/grow-1 toxin-antidote selfish genetic element in the nematode *C. tropicalis*, which unusually has a parent-of-origin effect. The authors have conducted extensive mechanistic and genetic experiments to document how it is dependent on the piRNA pathway, rather than methylation as understood in mammalian systems. The other reviewers thoroughly assessed these aspects of the study, and my view is that it is solid work and includes some clever approaches to distinguish alternative molecular hypotheses. The writing is clear given the complexity of explaining epigenetic phenomena. I was asked to focus on the evolutionary aspect.

We thank reviewer #4 for their overall positive evaluation of our manuscript highlighting the soundness of our results, including some ingenious genetics.

Lines 520-523: It is unclear to me what aspects of the results presented provide evidence for these statements and conclusions. I noted no explicitly evolutionary or population genetic analyses up to this point in the manuscript that would speak to how strong selection acts on the TA elements, nor what form of selection.

The reviewer is referring to the following paragraph, which pertains to the discovery and characterization of a second and highly divergent TA, *slow-2/grow-2*:

“Collectively, our results strongly suggest that selfish TAs are under strong selection not only to escape their host defense system (piRNAs) but also to fight evolutionary related selfish haplotypes segregating in populations. The combined effect of these evolutionary forces is likely behind the extreme genetic divergence of this locus, which is far greater than the levels found between nematode species. These dynamics could explain hyperdivergent loci under balancing selection in nematode genomes”

We agree with the reviewer that this paragraph was speculative. Furthermore, since it is not related to the main findings of our manuscript, we have toned it down, significantly reduced its content, and moved the whole subsection to a Supplementary Note 3. This paragraph now reads:

*“We speculate that the high levels of genetic divergence between *slow-1* and *slow-2* could be explained in part by an evolutionary arms race between the two TAs originating from their antagonistic action and involving continues cycles of positive selection and counter-*

adaptation. However, a formal test of this model would require population genetic modeling and allele frequency measurements derived from wild populations”

Due to the space constraints in the main text, the subsection that contained this last paragraph has been replaced by following sentence in the main text (discussion section):

See line 389

“Remarkably, an evolutionary related but highly divergent TA, slow-2/grow-2, does not show a parent-of-origin effect, suggesting that this trait can quickly evolve in nature (Extended Data Fig. 8 and 9; Supplementary Note 3).

Lines 546-548: In referring to the present mechanism as a “major force driving the evolution of imprinting”, I can’t help but wonder how much the selfing mode of reproduction in *C. tropicalis* constrains or facilitates this speculation relative to obligatorily outbreeding systems like the insect, plant, and rodent organisms alluded to earlier in the sentence. Given that the ancestral reproductive state in the lineage leading to *C. tropicalis* also was obligatorily outbreeding, it also raises the question whether selfers are predisposed to revealing TA systems in interpopulation crosses following their evolutionarily resolution within different subpopulations, whether they might be more prevalent in outbreeding taxa, etc. I realize that deep consideration of some of these issues lies beyond the scope of the present study, but if the authors elect to touch on broader evolutionary issues then it seems important to consider the implications in more than an offhand manner.

The reviewer is referring to the following paragraph (discussion section):

“Because TAs and maternal-zygotic lethal factors are not only present in nematodes but also segregate in wild insect, plant, and mouse populations, parasitic conflict and co-option of sRNA-mediated defense systems may be a major force driving the evolution of imprinting”

We agree with the reviewer that the impact of the mode of reproduction on the evolution of TA systems (and parent-of-origin effects) is indeed a fascinating question, but we also feel exploring these questions is the beyond the scope of this study. Previous work by the Rockman lab (Noble et al. *eLife* 2021) and our own lab (Ben-David et al. *Current Biology*, 2021) suggests that the mode of reproduction indeed has a big impact on the evolutionary dynamics of TAs. Simulations show that TAs spread much more easily in obligatorily outbreeding populations.

[Redacted text]

Nonetheless, we have also toned down and modified the original paragraph to better reflect our findings:

“Because TAs and analogous maternal-zygotic lethal factors are not only present in nematodes but also segregate in wild insect, plant, and mouse populations, we propose that co-option of sRNA-mediated defense systems originating from selfish conflict might be a recurrent event facilitating the evolution of imprinting.”

Minor comments:

Sometimes a “.” is used to designate decimal numbers whereas other times a “;” is used, please edit for consistency.

Thank you, we found 7 instances of this typo in our manuscript that have now been corrected.